# Predicting discharge capacity of vegetated compound channels: uncertainty and identifiability of 1D process-based models

Adam Kiczko[1], Kaisa Västilä[2, 3], Adam Kozioł[1], Janusz Kubrak[1], Elżbieta Kubrak[1], and Marcin Krukowski[1]

[1]Warsaw University of Life Sciences – SGGW, Institute of Environmental Engineering
[2]Department of Built Environment, Aalto University School of Engineering, Espoo, Finland
[3]Freshwater Centre, Finnish Environment Institute, Helsinki, Finland

**Correspondence:** Adam Kiczko (adam_kiczko@sggw.edu.pl)

**Abstract.** Despite the development of advanced process-based methods for estimating the discharge capacity of vegetated river channels, most of the practical one-dimensional modeling is based on a relatively simple divided channel method (DCM) with the Manning flow resistance formula. This study is motivated by the need to improve the reliability of modeling in practical applications while acknowledging the limitations on the availability of data on vegetation properties and related parameters required by the process-based methods. We investigate whether the advanced methods can be applied to modeling vegetated compound channels by identifying the missing characteristics as parameters through the formulation of an inverse problem. Six models of channel discharge capacity are compared in respect of their uncertainty, using a probabilistic approach. The model with the lowest estimated uncertainty in explaining differences between computed and observed values is considered as the most favorable. Calculations were performed for flume and field settings varying in floodplain vegetation submergence, density, and flexibility, and in hydraulic conditions. The output uncertainty, estimated on the basis of a Bayes approach, was analyzed for a varying number of observation points, demonstrating the significance of the parameter equifinality. The results showed that very reliable predictions with low uncertainties can be obtained for process-based methods with a large number of parameters. The equifinality affects the parameter identification but not the uncertainty of a model. The best performance for sparse, emergent, rigid vegetation was obtained with the Mertens method and for dense, flexible vegetation with a simplified two-layer method while a generalized two-layer model with a description of the plant flexibility was the most universally applicable to different vegetative conditions. In many cases, the Manning-based DCM performed satisfactorily but could not be reliably extrapolated to higher flows.

## 1 Introduction

Compound channels consisting of a main channel and vegetated floodplains are commonly observed both in natural and engineered settings. For instance, vegetated compound (two-stage) channels have been recently proposed as an environmentally

preferable alternative to conventional dredging in flood and agricultural water management (e.g. Västilä and Järvelä, 2011). Such a nature-based solution (NBS) is expected to allow combining the technical needs, e.g. flow conveyance and channel bed stability, and the environmental requirements, e.g. improved water quality and biodiversity (Rowiński et al., 2018), but requires reliable predictions on the discharge capacity. Herein, the difficulty results from the complex cross-sectional geometry and the composite roughness resulting from parts of channel with highly different flow resistance. Floodplain vegetation is the main factor complicating the predictions particularly in small to medium-sized channels where up to 90 percent of the flow resistance can be caused by plants (e.g. Västilä et al., 2016).

With an increase of computing power, two- and even three-dimensional models are gaining popularity in flood assessments (Teng et al., 2017; Liu et al., 2019). In practice, one-dimensional models, on which the present study focuses, still play an important role, especially in tasks requiring long term or large spatial scale simulations (e.g. Yu et al., 2019; Chaudhary et al., 2019). In one-dimensional flow routing models the most widely used technique for predicting the discharge capacity of compound channels is the Divided Channel Method (DCM) with the Manning formula, defined in 1960 (Posey, 1967). In this approach flow is computed separately in channel zones with differing flow resistance, usually the main channel and floodplains. The momentum exchange between areas of the higher and lower stream velocity, the so called kinematic effect, is represented by rough imaginary walls at the interfaces (Sellin, 1964; Kubrak et al., 2019a, b). Despite the well-known limitations of the DCM (Myers, 1978; Fread, 1989; Soong and DePue, 1996; Pasche, 2007), the Manning formula is presently the basis for the majority of practical models for flood hazard assessments, design of hydraulic structures and water management (Shields et al., 2017).

To improve the reliability of practical discharge capacity estimation in vegetated channels, the key vegetation properties controlling the reach-scale flow resistance should be incorporated into the calculations (e.g. Yen, 2002; Luhar and Nepf, 2013). One of the most sophisticated model of the channel capacity can be attributed to Shiono and Knight (1991), who on the basis of a turbulent flow theory, derived equations for depth averaged velocities in the cross-section plane. Accompanied with an additional drag term, the method was successfully used to model flow in a channel with composite roughness consisting of vegetated and non-vegetated zones (e.g. Zhang et al., 2018; Abril and Knight, 2004; Zinke et al., 2011; Tang and Knight, 2008; Kalinowska et al., 2020). However, for a typical practical case, the Shiono and Knight (1991) model is too complex, requiring much of modelers efforts, especially in presence of efficient two-dimensional solutions.

Several approaches providing a physically-based characterization of vegetation and the flow-vegetation interactions are available for straightforward 1D discharge capacity assessments in small-to-medium-sized vegetated channels. In these models, vegetation can be represented as rigid or flexible, interacting with water stream as submerged and emergent (Shields et al., 2017). There are many methods explaining each of these types of vegetation and a comprehensive review can be found in Aberle and Järvelä (2013). Some of the most recognized methods include e.g. those developed by Pasche (1984) and simplified by Mertens (1989) to describe the flow in zones with unsubmerged (emergent) vegetation, by Arcement and Schneider (1989) who presented empirical relationships for Manning roughness coefficients and vegetation parameters; by Klopstra et al. (1996) who derived an process-based model for rigid, submerged vegetation; by Järvelä (2004) who provided a process-based approach for emergent rigid and flexible vegetation; by Baptist et al. (2007) who introduced a two-layer model for rigid vegetation; and

by Luhar and Nepf (2013) who developed a two-layer model for submerged vegetation. Despite the recent developments of these process-based methods, there is a lack of knowledge in whether the state-of-the-art methods with a significant number of parameters are reliable in common practical applications characterized by insufficient information on vegetative properties

and related model parameters.

An important drawback of vegetation models for hydraulic resistance, from the practical (modeler's) point of view, is that they require much more data than traditional methods. For example with the DCM, in terms of roughness, the river cross-section can be usually characterized using three values of the Manning coefficient, for the main channel and two floodplains. The vegetation models would require specific data on plant features, such as density, spacing, shape or species, and leaf area

indices. An exception may be channel design assignments, where it is possible to assume a future character of a plant cover after an intended intervention, necessary data on vegetation can be obtained through field surveys, which noticeable increase costs of a model application. A promising way for a more effective determination of vegetation features might be remote sensing and many studies were devoted to the use of these techniques in floodrouting. For example Casas et al. (2010); Forzieri et al. (2010); Abu-Aly et al. (2014); Wolski et al. (2018) investigated the use or airborne laser scanning for determining vegetation classes,

that corresponds to hydraulic features. The obtained values of plant properties are however affected by a strong uncertainty, resulting from classification itself, but also generalization and variation within a class, as demonstrated by Straatsma and Huthoff (2011). Forzieri et al. (2012) argued, that airborne laser scanning itself is not suitable to measure plant characteristics, without extensive field reference data. Therefore more recent attempts focused on application of Terrestrial Laser Scanning (e.g. Antonarakis et al., 2009; Jalonen and Järvelä, 2014; Jalonen et al., 2015; Kałuza et al., 2018). However still, the use of

the remote sensing data in vegetation models, requires extensive field measurements, to establish a link between obtained data and hydraulic properties.

Aforementioned Straatsma and Huthoff (2011) study showed, that even with field measurements of vegetation properties, generalization of acquired parameters is rather unavoidable, especially when dealing with larger areas. Values characterizing vegetation, obtained in the field, have to be attributed to a spatial unit, usually representing a vegetation class. On the one hand,

together with the nonlinear form of the vegetation resistance models, such a generalization introduces significant uncertainty. On the other, it weakens the link between measured values and model parameters, which reflect the lumped hydraulic effect instead of representing physical quantities. Such quantities are not measurable and depend on the structure of the flow model, adopted governing equations, simplifications of the flow dynamics. In still scarce studies where floodrouting is analyzed with the use of vegetation roughness models, some researchers tend considering plant properties as model parameters that should be

calibrated, i.e. identified in the respect of observations. So, to treat them similarly to Manning coefficients, which are usually obtained by the model calibration, where their values are adjusted, to ensure an agreement between computed and observed e.g. water levels, stream velocities or flow rates – by solving the inverse problem (e.g. Khatibi et al., 1997; Marcinkowski et al., 2018, 2019; Yu et al., 2019). The example is given by Dalledonne et al. (2019) who identified vegetation parameters describing e.g. stem diameters, their heights, drag coefficients and a leaf area index in the two-dimensional flow model. Berends et al.

(2019) directly addressed the problem of parameter identifiability of vegetation roughness models, also using two-dimensional

model. It seems, that when vegetation resistance methods become more popular in practical codes for floodrouting, this approach will become more common.

Performing model calibration using parameters of vegetation roughness models, rises at least four implications:

1. Is it possible to identify models for vegetation roughness on the basis of the inverse task? The problem arises from the larger number of parameters in vegetation roughness models, comparing to traditional approaches, based e.g. on the Manning formula. The problem was well demonstrated by Werner et al. (2005), who investigated the uncertainty and sensitivity of a hybrid two/one-dimensional model for a varying number of parameters used to describe a channel and floodplain roughness. Analyzing the parameter identification using a probabilistic approach, they showed, that with increasing number of parameters, the obtained parameter distributions become less specific, suggesting the same level of probability over a wide range of values. Moreover, the obtained parameter distributions were different from values suggested in literature. Although, the Werner et al. (2005) study did not account for vegetation roughness models, the same effect was observed in the case of these methods by Berends et al. (2019) and Kiczko et al. (2017). This leads to the second point.

2. Is it reasonable to apply process-based vegetation roughness models if the identification of their parameters results in values differing from the real values measured at the field (Werner et al., 2005; Kiczko et al., 2017; Berends et al., 2019)? Such a calibration procedure rises an impression of using process-based methods as data-driven, black-box models, common e.g. in rating curve assessments (Kiang et al., 2018). From this perspective, the process-based methods, with other than measured parameters act as functions with large number of parameters, comparing to traditional approaches like the Manning based DCM. The effect can be probably mitigated by applying constrains on the parameter values to ensure that they are within their physical bands. With additional information on channel vegetation, using e.g. remote sensing or land use maps, it might be possible to restrict their variability ranges further. The advantage of process-based approaches might come form the physical interpretability of their parameters. For instance, too large stem diameters of plants are easier to spot than too high values of Manning roughness coefficients. However, still there is a lack of evidence, if it is beneficial applying process-based models, instead of pure data-driven approaches.

3. The choice of the vegetation roughness model, e.g. for rigid or flexible vegetation, depends on the type of vegetation present in the channel. Is it then possible to choose an appropriate model without knowledge on the plant type? This issue should be considered in respect of the point 3, by analyzing if it is possible to choose an appropriate model structure by solving the inverse problem.

4. Are the process-based models beneficial compared to e.g. the DCM-based Manning approach when there is a need to extrapolate to higher flows? This is an issue well recognized in hydrology (Kuczera and Mroczkowski, 1998), that identification of simpler models is much more straightforward, but because process-based models incorporate casual interrelationships, they provide better basis for the extrapolation. It is of a special importance in flood assessments, where the calibrated models need to be extrapolated to higher flood flows.

The overall goal of the present paper is to investigate the implications of the use of 1D state-of-the-art process-based methods in discharge capacity estimation of small-to-medium-sized vegetated compound channels. These common practical applications are typically characterized by insufficient data on vegetative properties, so that models are identified in terms of the inverse problem. We compare the model identifiability, uncertainty, and physical interpretation of the parameters of discharge capacity methods characterized with different levels of parameterization. The following methods were investigated: Manning based DCM, Pasche (Pasche, 1984) and Mertens (1989) methods designed for emergent rigid vegetation, and three versions of the two-layer model proposed by Luhar and Nepf (2013) as modified by Västilä and Järvelä (2018), designed for flexible submerged or emergent vegetation. All models were applied to vegetation conditions differing in relative submergence (covering both submerged and emergent conditions) and density, as motivated by real cases where it is possible that e.g. a "rigid" vegetation model is applied for flexible vegetation because of a lack of information on the vegetation properties. Parameter identification was conditioned on water depths instead of discharges to make the problem more similar to practical cases, such as flood assessments, where a model outcome is usually the water level. It is out of the scope of the paper to provide a summary of all available methods.

## 2   Methods

This section provides an overall description of the applied methodology. In the subsection 2.2.2 Pasche (Pasche, 1984) and Mertens (1989) models for rigid emergent vegetation are presented. Flexible vegetation models based on the two-layer assumption of Luhar and Nepf (2013), generalized by Västilä and Järvelä (2018) are provided in subsections 2.2.3-2.2.4. Computations were performed for steady state conditions, by applying vegetation roughness models to find water levels in a channel cross section.

Two experimental data sets collected from vegetated compound channels were used: flume measurements with rigid vegetation (Koziol, 2010; Kozioł, 2013, section 2.3.1) and field measurements with natural mostly grassy vegetation at Ritobacken brook (Västilä et al., 2016, section 2.3.2). The process-based models of vegetation roughness were compared with the traditional DCM with Manning roughness coefficients. For the purpose of the identification task it was necessary to assume, that parameters are constant and for that reason, the experimental data was divided into sets, where vegetation features were constant as possible. Therefore, the model identification for the field data was performed separately for each season.

Similarly to Werner et al. (2005) and Berends et al. (2018), the parameter identification problem is defined in the probabilistic manner, on the basis of Bayesian estimation (section 2.1). The adapted assumption is that the methods can be compared in terms of assessed uncertainty: i.e., the more appropriate the method is, the lower is the uncertainty of its predictions. At this point it should be noted that with a such problem statement the goal is the model identification, rather than parameter identification (Mantovan and Todini, 2006), as without knowledge on true parameter values, only measures for model outputs are used in the calibration process. The model identifiability in a probabilistic manner is understood as the ability to determine the parameter distribution that explains the model uncertainty in relation to observations. An effort was made to ensure that uncertainty analysis is objective and repeatable, despite different assumptions on initial *a priori* parameter distributions for each method.

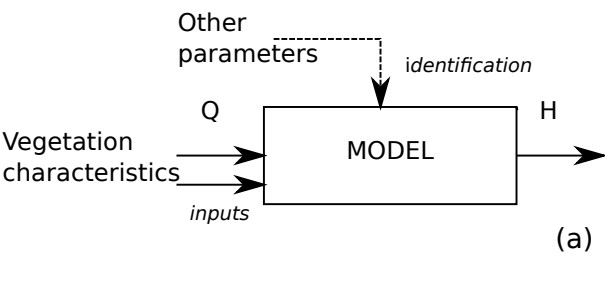

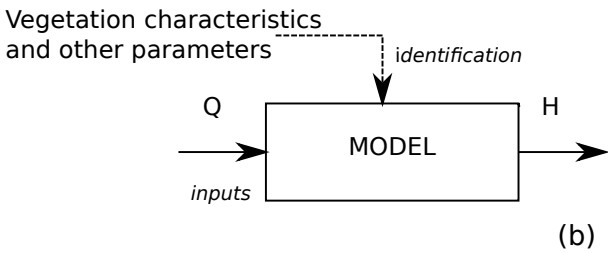

**Figure 1.** Two ways to define the parameter identification problem for process-based methods of channel discharge: (a) traditional approach, (b) adapted in the present study.

The identification was performed for a different number of observations, similarly to hydrological studies of Her and Chaubey (2015); Her and Seong (2018). For calibration the points of rating curves were used, the effect of different possible combinations of observations in identification task was also investigated, e.g. model was calibrated for a set of five lower flows, but also for a set of five higher and all intermediate sets. To address the issue of using simpler and more complex, process-based models for extrapolation of the rating curve, a special focus was made on predictions of maximum flows with a model identified using only lower flows.

## 2.1 Parameter identification and uncertainty analysis

River assessments using one-dimensional models with DCM, based on the Manning formula, are usually performed without detailed knowledge on vegetation properties. The Manning roughness coefficients are considered as model parameters, identified in the inverse problem, where their values are adjusted to ensure satisfactory fit between model outputs and observations, e.g. computed and measured water depths $H$ at given discharge $Q$. The vegetation roughness models provide a relationship between plant features and the water flow. Vegetation characteristics, that can be obtained by field measurements or e.g. design assumptions, are considered as a model input. In discharge calculations, the use of such models can be illustrated with Figure 1a, where vegetation properties are one of model inputs. It is still necessary, to specify remaining parameters, like roughness coefficients for bed or drag coefficients for plants. The present study investigates the approach given in Figure 1b, where also vegetation characteristics in vegetation roughness models are considered as model parameters, that have to be identified, without a knowledge on channel vegetation. This makes the application of vegetation roughness models, similar to the way

how Manning based approaches are used. From the practical point of view, the difference, apart the model structure, comes from the number of parameters that have to be identified.

In the probabilistic parameter identification approach, parameters are assumed to be random variables explaining the model uncertainty (Werner et al., 2005; Berends et al., 2019). The model identification is performed along with the uncertainty analysis and consists in a determination of parameter distributions, that translates using the model to probabilistic distributions of model outputs, here water depths $H$. The results of parameter identification and uncertainty estimation are usually presented in a form of confidence intervals for model outputs and parameter marginal distributions. The problem was defined on the basis of Bayes estimation using using Generalized Likelihood Uncertainty Estimation (GLUE) approach (Beven and Binley, 1992; Romanowicz and Beven, 2006). Parameters distributions, are obtained using the Bayes formula:

$$P(\theta/H) = \frac{L(H/\theta) P(\theta)}{\int L(H/\theta) P(\theta) d\theta} \tag{1}$$

where $\theta$ stands for parameters, $H$ water depths, $P(\theta)$ *a priori* parameter distribution, $P(\theta/H)$ *a posteriori* parameter distribution, $L(H/\theta)$ likelihood function. The equation is solved using Monte Carlo sampling of parameters within the adapted *a prior* distributions $P(\theta)$ and model simulations for given flow rates $Q$.

The choice of the likelihood function $L(H/\theta)$ depends on the assumptions of the character of model errors. In the present study it was assumed that models are unbiased and errors between computed and observed water levels $\zeta$ are independent and normally distributed $\zeta \sim \mathcal{N}(0, \sigma^2)$, where $\sigma^2$ is unknown variance. The relationship between observed water levels $\hat{H}$ and the computed $H$ for a given flow rate $Q$ and parameters $\theta$ can be given as follows:

$$\hat{H} = H(Q, \theta) + \zeta \tag{2}$$

The error $\zeta$ explains all discrepancies between the model and observations, so as well the measurement and model uncertainty. Therefore the performed uncertainty analysis accounts for the total uncertainty. When comparing different models for the same observation set, the measurement uncertainty is constant and differences results from the model uncertainty. For independent and normally distributed errors $\zeta$ the likelihood function is given by (Romanowicz et al., 1996; Romanowicz and Beven, 2006):

$$L(H/\theta) = \frac{1}{\sqrt{2\pi\sigma^2}} \exp\left[\frac{-\sum\limits_{i=1}^{m}\left(H_i - \hat{H}_i\right)^2}{2\sigma^2}\right] \tag{3}$$

with $m$ standing for the number of observation points $\hat{H}_i$ with discharges $Q_i$ used in the parameter identification. It should be noted, that with the likelihood function given with Equation 3 the selection of a so-called behavioral set, common in GLUE approaches, is not necessary.

The variance $\sigma^2$ is unknown and in GLUE approaches it is usually estimated using model residuals (Romanowicz and Beven, 2006; Stedinger et al., 2008). In the present study, the $\sigma^2$ is determined on the basis of observations, by ensuring that they appropriate share is enclosed in confidence intervals (Blasone et al., 2008) of modeled water depths $H$. The optimization

problem is defined in terms of scaling factor $\kappa$ for the variance of model residuals $\sigma_r^2$, used commonly in GLUE:

$$205 \quad 2\sigma^2 = \kappa\sigma_r^2 \tag{4}$$

The variance of model residuals $\sigma_r^2$ is calculated using the Monte Carlo sample Romanowicz and Beven (2006):

$$\sigma_r^2 = \text{var}\left(\frac{1}{m}\sum_{i=1}^{m}\left|H_i - \hat{H}_i\right|\right) \tag{5}$$

The purpose of Equation 4 is to provide an initial guess on $\sigma^2$. The $\kappa$ scaling factor is computed on the basis of minimization task:

$$210 \qquad \kappa = \arg\min_{\kappa}\left(\epsilon\kappa + \left|p - \frac{1}{m}\sum_{i=1}^{m}J\left(\hat{H}_i\right)\right|\right) \tag{6}$$

$$J\left(\hat{H}_i\right) = \begin{cases} 0 & \text{if} \quad \hat{H}_i \in [H_i^{q_L}, H_i^{q_u}] \\ 1 & \text{else} \end{cases} \tag{7}$$

where $H_i^{q_L}$, $H_i^{q_U}$ denote lower and upper quantile ($q_L$, $q_U$) of the calculated water levels from the *a posteriori* distribution (Equation 1), obtained with the likelihood function (Equation 3); $p$ stands for confidence interval, defined as: $p = q_U - q_L$. In the present study 95% confidence intervals ($p = 0.95$) were used, with $q_L = 0.025$ and $q_U = 0.975$. $\epsilon$ is a small number as a

215 penalty for too wide confidence intervals of water levels $H$. The minimum of the function given with Equation 6 should be the smallest value of $\kappa$ for which the last term in Equation 6 equals zero:

$$p - \frac{1}{m}\sum_{i=1}^{m}J\left(\hat{H}_i\right) \leq 0 \tag{8}$$

This is true when exactly $p \cdot m$ observations fall within the confidence intervals. For $p = 0.95$ and relatively small observation sets of $m \sim 10$ in the present study, minimum is found when all observations are enclosed by intervals. In such a case, the

220 sum term in Equation 8 is equal to 1 and the difference becomes negative. The procedure given with Equations 6-8 allows for determining the minimal value of $\sigma^2$ (Equations 2 and 3) sufficient to explain model uncertainty in respect of observations. It should be noted, that for a poor model and/or inappropriate variability ranges of *a priori* parameter distributions, such a solution might not exist. The term given with Equation 8 was therefore a criterion for the model identifiability. The model was considered identifiable, if the Equation 8 was fulfilled.

The assumption of *a priori* parameter distributions $P(\theta)$ have a significant effect on the *a posteriori* solution (Freni and Mannina, 2010; Tang et al., 2016). In the present study to obtain objective uncertainty estimates for a different methods and parameters it was decided to apply uninformative and relatively wide *a prior* distributions, assuming no knowledge on channel vegetation, maintaining however physically interpretable ranges (Table 1). The parameter ranges of uniform distribution were chosen to ensure that the high probability region is enclosed by the Monte Carlo sample. The span of this region links with

confidence intervals comprising 95% of the *a posteriori* distribution, so it was assumed that the sample should be noticeably larger. It was obtained by testing, if it is possible to make confidence intervals wider by increasing the $\kappa$ coefficient determined

**Table 1.** Parameter variability ranges (uniform $P(\theta)$ distribution) for Ritobacken and flume experiments, numerals in parameter symbols are used to distinguish properties on left (1) and right (1) channel side.

| Model | Parameter | $m_{mc}$ | Ritobacken data Min. Value | Max. Value | $m_{mc}$ | Flume data Min. Value | Max. Value |
|---|---|---|---|---|---|---|---|
| DCM | $n_1\ [\mathrm{m}^{-1/3}\mathrm{s}]$ | $2.5\cdot10^4$ | 0.012 | 0.15 | $2.5\cdot10^4$ | 0.012 | 0.06 |
|  | $n_2,\ n_3\ [\mathrm{m}^{-1/3}\mathrm{s}]$ |  | 0.012 | 0.15 |  | 0.012 | 0.12 |
| Pasche and Mertens | $d_p[\mathrm{m}]$ | $5\cdot10^4$ | 0.004 | 0.100 | $5\cdot10^4$ | 0.004 | 0.072 |
|  | $a_{x1},a_{x2}\ [\mathrm{m}]$ |  | 0.001 | 0.9 |  | 0.05 | 0.9 |
|  | $a_{z1},a_{z2}\ [\mathrm{m}]$ |  | 0.001 | 0.9 |  | 0.05 | 0.9 |
|  | $k_{ch}[\mathrm{m}]$ |  | $2.5\cdot10^{-5}$ | $4.5\cdot10^{-4}$ |  | $2.5\cdot10^{-5}$ | $4.5\cdot10^{-4}$ |
|  | $k_{fp1},k_{fp2}\ [\mathrm{m}]$ |  | 0.005 | 0.09 |  | 0.005 | 0.09 |
|  | $b_{iii}/B_{fp}\ [-]$ |  | 0.333 | 1 |  | 0.333 | 1 |
| GTLM | $C_{dx,F}\ [-]$ | $10^5$ | 0.09 | 0.2 | $5\cdot10^4$ | 0.001 | 1.5 |
|  | $C_{dx,S}\ [-]$ |  | 0.82 | 1.03 |  | 0.001 | 1.5 |
|  | $\chi_F\ [-]$ |  | -1.21 | -0.97 |  | -1.21 | -0.97 |
|  | $\chi_S\ [-]$ |  | -0.32 | -0.2 |  | -0.32 | -0.2 |
|  | $A_l/A_b\ [-]$ |  | 0 | 30 |  | 0 | 30 |
|  | $A_s/A_b\ [-]$ |  | 0 | 30 |  | 0 | 30 |
|  | $C^*\ [-]$ |  | 0.01 | 0.20 |  | 0.01 | 0.20 |
|  | $l_L/L_L,l_R/L_R\ [-]$ |  | 0 | 1 |  | 0 | 1 |
|  | $h_L,h_R\ [\mathrm{m}]$ |  | 0 | 2.15 |  | 0 | 0.3 |
| STLM | $C^*$ | $5\cdot10^4$ | 0.01 | 0.20 | $2.5\cdot10^4$ | 0.01 | 0.20 |
|  | $l_L/L_L,l_R/L_R\ [-]$ |  | 0 | 1 |  | 0 | 1 |
|  | $h_L,h_R\ [\mathrm{m}]$ |  | 0 | 2.15 |  | 0 | 2.15 |
| PTLM | $C_Da$ | $5\cdot10^4$ | 0.01 | 100 | $5\cdot10^4$ | 0.01 | 100 |
|  | $C^*$ |  | 0.01 | 0.20 |  | 0.01 | 0.20 |
|  | $h\ [\mathrm{m}]$ |  | 0 | 2.15 |  | 0 | 0.3 |

Note, in Flume experiments cross-section was symmetric and the same parameter values were used for following parameters:$l_L/L_L = l_R/L_R, h_L = h_R, a_{x1} = a_{x2}, a_{z1} = a_{z2}$

using Equations 6-8. This way it was possible to check, if confidence intervals are not directly affected by the span of the Monte Carlo sample. When confidence intervals were insensitive to increasing value of $\kappa$ it was necessary to extend ranges of a priori parameter distributions. It should be noted, that it was necessary only in the case of unsuitable models, where condition given by Equation 8 was usually not fulfilled.

It is acknowledged that the parameter identification and associated uncertainty depend on the size of the observation data set. To address this issue, the model identification (Equation 1) was performed for a varying number $m$ of observation points: $\hat{H}_1, \ldots, \hat{H}_m$ and corresponding flow rates $Q_1, \ldots, Q_m$ as the input. The $m$ included values form 1 to the total number of available observations $M$: $m = 1, \ldots, M$. The calculations included all possible combinations of observations with the given $m$ i.e. $\frac{M!}{m!(M-m)!}$. The number of all combinations is then $2^M - 1$, excluding the empty set ($m = 0$). Such an approach allows eliminating the effect of non-representative observation samples. The method was discussed previously by Kiczko et al. (2017).

Observation points not used for identification $M - m$ act as a verification set. In this analysis, both the proportion of verification points that falls within estimated confidence intervals and the width of confidence intervals are used as measures of model performance. The more narrow the confidence bands and the less observation points falling outside them, the better a model is. On the opposite, a less adequate model requires larger spread of the solution, to enclose observations, as it wrongly explains their variability. Because the different combinations of $m$ points resulted in multiple uncertainty estimates, the results were presented in terms of statistical moments, as a function of $m$. For a detailed description of results box-plots were used, where the median is given as a horizontal line within a box, that spans over 25% and 75% quantile, whiskers indicate the result extent, excluding extreme values given with cross marks.

As it was mentioned before, it should be noted that by applying the Bayesian concept, the objective is the model identification (see the comment on the purpose of the Bayesian identification of Mantovan and Todini, 2006). Parameter variability is used to describe the uncertainty, specifically the error $\zeta$ defined with the Equation 2. This comes from the form of the inverse problem, where likelihood measures depends only on measured model outputs, here water depths and it is possible that parameters that are different from real ones, but provide a good model fit, are considered as likely (Werner et al., 2005; Kiczko et al., 2017; Berends et al., 2019). To demonstrate this effect and to discus possible implications the obtained marginal *a posteriori* distributions of parameters $P(\theta/H)$ were compared with values obtained by direct measurements in analyzed case studies. A special focus was given on extrapolation capabilities of vegetation models with parameters determined on the basis of the inverse problem, assuming a lack of the knowledge on channel vegetation properties.

The Latin Hypercube Sampling (Budiman, 2017) was applied to improve performance of the Monte Carlo technique. The size of the Monte Carlo sample ($m_{mc}$, Table 1) was determined in each case by trial and error, to satisfy the convergence of the solution. As the criterion for the convergence the difference of estimated average water depth was used. The number of simulation was considered as sufficient, when difference in subsequent ensembles stabilized bellow $10^{-5}$ - $10^{-4}$ m.

## 2.2 Discharge capacity formulas

### 2.2.1 Divided Channel Method

In the DCM approach (Posey, 1967), the channel cross section is divided in flow zones of similar hydraulic conditions, typically the main channel and floodplain. The interactions between the zones of significantly different mean velocities are reproduced with a rough imaginary wall, applied to the zone with the higher velocity, i.e. the main channel. In the present study, the roughness of the interface was assumed to equal the roughness of the channel banks next to the interface. Parameters of the method

are the roughness coefficients for each flow zone. In the present study, DCM was based on the Manning formula, with the common approach of having separate Manning coefficients for the main channel ($n_c$), and left ($n_L$) and right floodplain ($n_R$). The parameter bands with $m_{mc}$ Monte Carlo sample sizes are provided in Table 1 separately for flume and field experiments. For flume data sets calculations were performed for a symmetric channel, which allowed to reduce the number of parameters, as the same values were used for the left and right floodplain.

### 2.2.2 Pasche and Mertens methods

A brief concept of the Pasche method is provided by Pasche (1984); Pasche and Rouvé (1985) and a detailed description of the algorithm used herein is provided in Kozioł et al. (2004). The model describes the discharge capacity of the compound cross section with rigid vegetation, derived for steady flow conditions. Similarly to DCM, the model divides the compound cross-section into regions of the main channel and floodplains, dominated by bottom and vegetation roughness, respectively. It accounts additionally for the transition region between these two main zones. As in the DCM, the interactions between the main channel and floodplains are modeled using an imaginary rough wall. For the resistance of the imaginary wall, bed and also vegetation stems the Darcy-Weisbach formula is used.

The Darcy-Weisbach friction coefficients are determined using a set of semi-empirical equations for each zone and the imaginary wall, including transitional regions. The method explains the extent of the transition region within the vegetated region, affected by the higher flow velocity of the unvegetated main channel. The flow in the main channel depends on the apparent resistance of the imaginary wall. There is no general expression for the span of the transition region in the main channel, and it has to be established for each case.

Velocities in the flow zones and transitional regions are interrelated by the apparent resistance. Equations describing these dependencies have an implicit form that requires iterative methods for solving, so that the Pasche method has a very complex numerical solution and it may be affected by a lack of convergence for infeasible parameter sets. Mertens (1989) attempted to improve the numerical efficiency of the Pasche concept by simplifying most of the demanding implicit formulas to less accurate but explicit ones, reducing the number of terms requiring iterative numerical solving.

In the Pasche and Mertens methods, a detailed parametrization of the channel, including plant properties, surface roughness and the extent of the interaction zone in the main channel, is used. Assuming that the modeler has only knowledge on the geometry of the cross-section, the following parameters have to be identified: $a_x$, $a_y$, longitudinal and horizontal spacing of plant stems; $d_p$ average diameter of the stems; $k_f$, $k_c$ roughness height of the floodplain and the main channel bed; $b_{III}/B_c$ ratio of the interaction region width in the main channel ($b_{III}$) to the main channel width ($B_c$). Assuming that the channel is symmetric, the total number of parameters is six. Modeling different properties of vegetation on left (subscript $L$) and right (subscript $R$) floodplains ($a_{x,L}:a_{x,R}$, $a_{z,L}:a_{z,R}$, $d_{p,L}:d_{P,R}$, $k_{f,L}:k_{f,R}$) increases the number of parameters up to ten.

### 2.2.3 Generalized and Simplified Two-Layer Model

In the present study, the two layer model of Luhar and Nepf (2013), generalized by Västilä and Järvelä (2018) for more complex cross-sections is considered as the state-of-art approach for submerged vegetation. This Generalized Two-Layer Model

(GTLM) is based on the momentum balance with drag coefficients at the interfaces between vegetated and unvegetated areas of the channel cross section. Generalization proposed to the original model (Luhar and Nepf, 2013) by Västilä and Järvelä (2018) consists in assuming a non-rectangular cross-section, so that the channel width is replaced by the wetted perimeter ($P$) and water depth by the hydraulic radius ($R$).

The channel discharge capacity is computed on the basis of equations for mean velocities in the unvegetated ($u_0$) and vegetated ($u_v$) parts of the cross section (Västilä and Järvelä, 2018):

$$\frac{u_0}{(gSR)^{1/2}} = \left[\frac{2P(1 - B_X)}{C^*(L_b + L_v)}\right]^{1/2} \tag{9}$$

$$\frac{u_v}{(gSR)^{1/2}} = \left[\frac{2PB_X + C^*L_v(u_0^*)^2}{C_D aPRB_X}\right]^{1/2} \tag{10}$$

where $g$ is the gravitational constant, $S$ energy slope, $u_0^* = \frac{u_0}{(gSR)^{1/2}}$ dimensionless velocity in unvegetated zone, $C^*$ the drag coefficient for shear stresses at the channel bed and at the interface between vegetated and unvegetated zones, $L_b$ and $L_v$ wetted lengths of the unvegetated channel margin and of the interface between vegetated and vegetated zones, respectively. $B_X$ denotes the vegetative blockage factor in the cross section, defined as the vegetated flow area divided by a total flow area. Physically, there might be different values of drag coefficients for bed and the interface of the vegetation zone. Following Luhar and Nepf (2013); Västilä and Järvelä (2018), it was herein assumed that the same value of $C^*$ can be used for both regions.

$C_d a$ is the vegetative drag per unit water volume, expressed conventionally as the product of a drag coefficient $C_d$ and the frontal projected plant area per unit water volume $a$, assuming that plants are rigid simple-shaped objects. To account for the presence of foliage and the flexibility of the plants inducing bending and streamlining, the vegetative drag per unit water volume can be parameterized as (Västilä and Järvelä, 2018)

$$C_D a = C_{D_X,F}\left(\frac{u_C}{u_{X,F}}\right)^{\chi_F}\frac{A_L}{A_B h} + C_{D\chi S}\left(\frac{u_C}{u_{XS}}\right)^{\chi_S}\frac{A_S}{A_B h} \tag{11}$$

where $u_C$ is a characteristic approach velocity, taken here as equal to the velocity in a vegetation layer: $u_C \approx u_v$. $A_S$ denotes total frontal projected areas of the plant stems and $A_L$ the total one sided leaf area per unit ground area $A_B$. $C_{D_X,S}$ and $C_{D_X,F}$ represent constant coefficients for the drag of stems and foliage, respectively. The effect of streamlining and reconfiguration on the drag is described using exponents $\chi_S$ and $\chi_F$, for stems and foliage, respectively. $u_{X,F}$ and $u_{X,S}$ are reference velocities needed for determining the drag and reconfiguration coefficients.

Equations 9 and 11 implicitly depend on each other and require numerical solving. In the conservative approach vegetation parameters have to be known (Figure 1 a). The blockage factor $B_X$ requires knowledge on the vegetation distribution and/or height in the cross section. $\frac{A_S}{A_B}$ and $\frac{A_L}{A_B}$ ratios characterizing the plant structure can be measured or typical values for a certain plant communities can be adopted. Drag coefficients $C_{D_X,S}$, $C_{D_X,F}$ and reconfiguration exponents $\chi_S$ and $\chi_F$, along with their reference velocities ($u_{X,F}$ and $u_{X,S}$), are factors specific for plant species or plant type and can be determined on the basis of laboratory measurements. Their values have been published for common plant species (Västilä and Järvelä, 2014; Jalonen and Järvelä, 2015; Västilä and Järvelä, 2018).

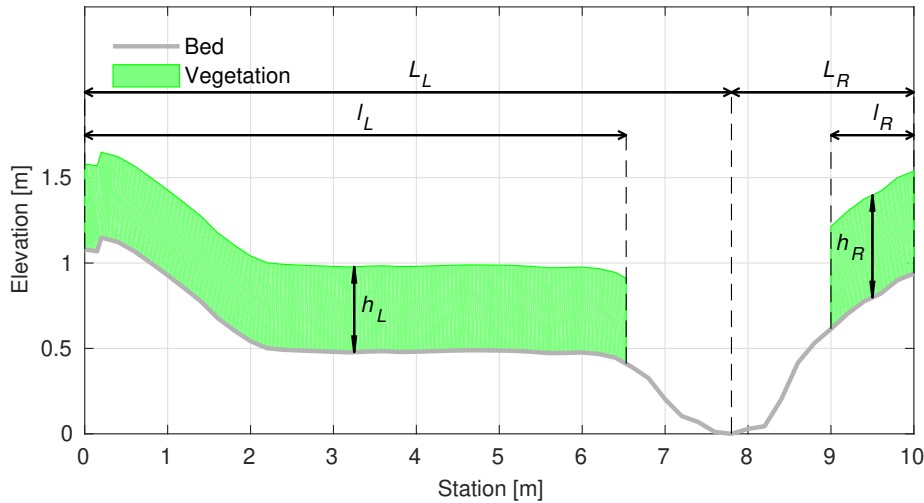

**Figure 2.** Parametrization of the blockage factor $B_X$, the cross section for Ritobacken Brook (Västilä and Järvelä, 2014)

For channel flows with dense vegetation for which over 80 percent of the discharge is conveyed in the unvegetated regions, the GTLM approach can be simplified by assuming that discharge in the vegetation layer is negligible with respect to the total

335 discharge: $u_v \approx 0 \ m/s$ (Luhar and Nepf, 2013; Västilä et al., 2016). The remaining Equation 9 does not require numerical solving. In the present study the above approach is referred as Simplified Two-Layer Model (STLM). It has to be noted, that with this approach, up to 20% of the discharge is neglected, depending on the density and cross-sectional blockage of vegetation. By neglecting the Equation 10, the STLM requires five and GTLM nine parameters.

Parameters of GTLM and STLM, resulting from Equation 9 are the drag coefficient for shear stresses $C^*$ and Blockage

Factor $B_X$. $B_X$ depends on the area occupied by the vegetation in the cross section. It changes with the water level and therefore should not be represented as a constant value but rather as the vegetaion share in the cross section area in the function of the depth. In the present study, to obtain a general parametrization, $B_X$ was described in terms of left-right extents $l_L/L_L$, $l_R/L_R$ and the height $h_L$,$h_R$ of vegetation. $L_L$, $L_R$ stand for the cross section width from the left and right bank, respectively, to the lowest elevation in the main channel. $l_L$ and $l_R$ denote vegetation extents, from banks towards the main channel (Figure 2).

$l_L/L_L$ is the vegetation extent on the left side, starting form the top of the left bank towards the channel middle point: 0 stands for clean bank, while 1 means that the vegetation cover extends over entire left side. The same applies for $l_R/L_R$, where it is assumed that vegetation zones starts from the top of the right bank. The vertical range of the vegetation in the cross section is obtained by adding $h_L$ or $h_R$ to the value of the ground elevation. The adopted parametrization for $B_X$ was verified with field estimates for Ritobacken Brook (Västilä and Järvelä, 2018) and allowed to obtain a fit with the linear correlation coefficient of

0.88.

It should be noted, that by parameterizing the Blockage Factor, the parameter identification task is much more complicated than in the conventional approaches. In the DCM the vegetation extent is equivalent to the division into main channel and

floodplains, which is known on the basis of the cross sectional geometry. Here, for GTLM and STLM it was considered as a part of the parameter identification problem.

### 2.2.4 Practical Two-Layer Model

Luhar and Nepf (2013) derived a formula for the Manning coefficient $n$ for shallow channels lined with vegetation, where the blockage factor can be approximated as $B_X \approx \frac{h}{H}$:

$$n \left( \frac{g^{1/2}}{KR^{1/6}} \right) = \frac{(gSR)^{\frac{1}{2}}}{U} = \left[ \left( \frac{2}{C^*} \right)^{\frac{1}{2}} \left( 1 - \frac{h}{R} \right)^{\frac{3}{2}} + \left( \frac{2}{C_D a h} \right)^{\frac{1}{2}} \left( \frac{h}{R} \right) \right]^{-1} \tag{12}$$

where $h$ stands for the vegetation height and $K = 1 \ m^{1/3}s^{-1}$ to ensure correct dimensions of the equation. In the presented form of the Equation (12), following Västilä and Järvelä (2018), the water depth $H$ was replaced with the hydraulic radius $R$.

Equation (12) has a convenient form to be easily applied in practical cases, where usually the Manning equation is used. In the present study, this approach is called the Practical Two-Layer Model (PTLM) as it requires less parameters influenced by vegetation. In the present study this approach is named Practical Two-Layer Model (PTLM) and applied as a three-parameter model, with the drag coefficient $C^*$, average vegetation height $h$ in the cross section and $C_D a$.

## 2.3 Case studies

The analyses were conducted for a flume data set (Koziol, 2010) and a field data set (Västilä et al., 2016) collected from vegetated compound channels, interpreted herein as 5 distinct case studies, as detailed below. To our knowledge, the field cases are one of the most thorough characterizations on the dependency between vegetation properties and discharge capacity in natural compound channels, including spatially-averaged values for vegetation height, blockage factor, and frontal area density in different seasons and flow conditions. The flume cases are representative of typical experimental arrangements where vegetation is simulated by rigid cylindrical elements at a uniform spacing.

### 2.3.1 Flume experiments

The experiments were conducted at the Warsaw University of Life Sciences (WULS-SGGW) using a physical model of a compound channel with rigid cylinders simulating vegetation. A detailed description of the dataset can be i.e. found in Kozioł and Kubrak (2015); Kozioł (2013); Kubrak et al. (2019a, b).

The modeled channel was straight, 16 m long with the slope of $s = 5 \cdot 10^{-4}$. The cross section was trapezoidal and wide for 2.10 m (Figure 3). The main channel bottom was made of smooth concrete with the estimated roughness height $k_s = 5 \cdot 10^{-5}$ m. Floodplain vegetation was simulated with rigid cylinders of a diameter $d_p = 0.008$ m and spacing $a_x = a_y = 0.1$m. There were two experimental variants of vegetation layout and floodplain roughness. In the first one (1) the floodplain bottom was made of the same smooth concrete as the main channel, with a single row of vegetation present also on channel bank (Figure 3a). In the second one (2), vegetation was constrained on the floodplain by removing the channel bank stems while floodplain surfaces were made rougher using a layer of terrazzo concrete of the grain size of 0.5 to 1 cm (Figure 3b).

Experiments were performed for steady and quasi-uniform flow conditions (Kubrak et al., 2019a, b). The water surface was kept parallel using a pressure gauge, measuring the differences in depths at cross sections located 4.8 and 12 m from the flume inflow and a weir localized at the outflow. Water discharge was measured using a circular weir and water levels were recorded in the middle of the channel.

The data set, used in the present study, consisted of discharge and water level observations (Appendix A1) within the range of: 0.037-0.060 $\text{m}^3/\text{s}$ (mean velocities: 0.2-0.4 m/s) and 0.2 - 0.3 m, respectively what includes only overbank flows. The number of observation point in the first variant was nine ($M = 9$) and in the second one ten ($M = 10$). The uncertainty calculations were performed for a symmetric channel, which allowed to reduce the number of parameters, as the same values were used for the left and right floodplain.

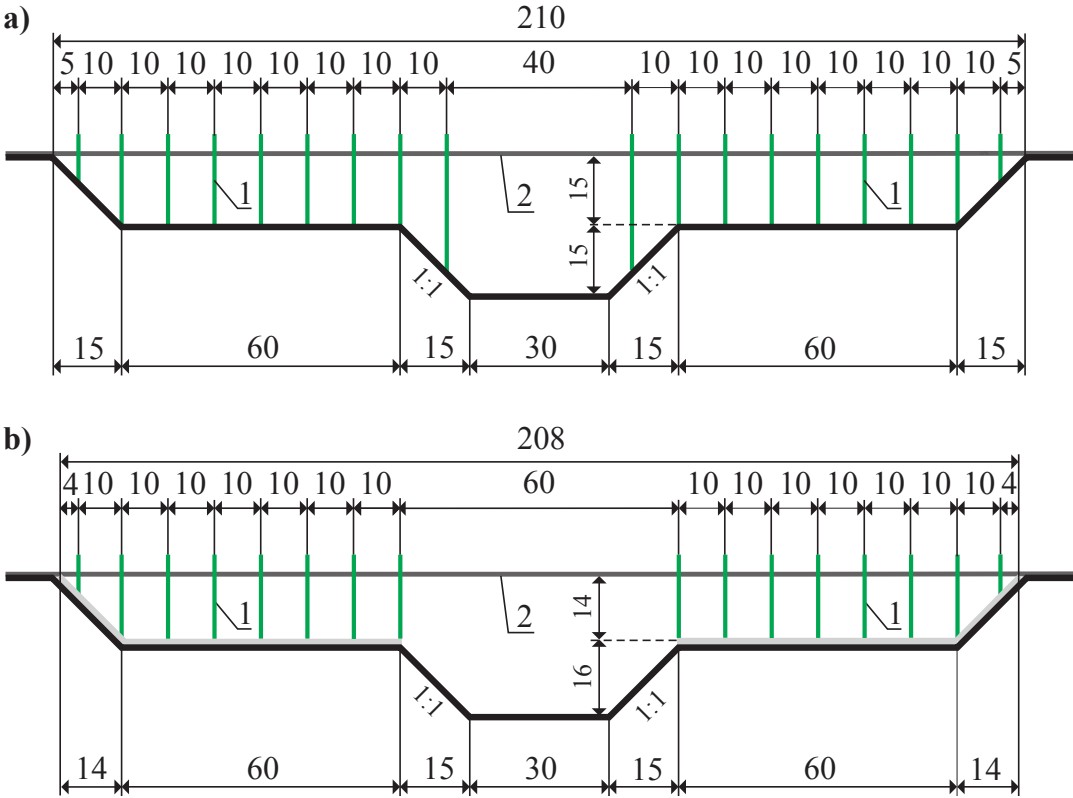

**Figure 3.** Laboratory channel cross section (dimensions in cm); 1 - rigid cylinders simulating vegetation; 2 - wooden strips supporting vegetation (Koziol, 2010); a) case 1; b) case 2.

### 2.3.2 Ritobacken field experiment

The field data with seasonally and annually varying vegetation was obtained from an 11 m wide compound channel, Ritobacken Brook (Finland, Figure 4), where the floodplain was excavated on one side of the existing channel in February 2010 (Västilä

et al. (2016)). Measurement series with vegetated floodplain flows (Appendix A2), were available for three seasons, with the number of observations given in brackets: Spring 2011 ($M = 6$), Autumn 2011 ($M = 12$) and Spring 2012 ($M = 11$). Vegetation consisted mainly of different grassy species, with both stems and foliage, while sparse woody vegetation covered 10% of the total wetted ground area.

The respective mean floodplain vegetation heights were $h = $ 9 cm, 47 cm and 24 cm while the vegetative blockage factor ranged at $B_X = 0.13 - 0.53$. The taller vegetation in Spring 2012 compared to Spring 2011 was explained by the ongoing succession phase after the floodplain excavation. Vegetation was submerged under all examined flows in Spring 2011 and under 42% and 64% of the flows in Autumn 2011 and Spring 2012, respectively.

The discharge capacity at different flow conditions was obtained from water level data recorded at 5-15 min intervals with pressure transducers at the upstream and downstream ends of a 190 m long test reach. The discharge was obtained from a rating curve determined for a culvert at the downstream end of the test reach. The stream is free flowing and there are no hydraulic structures affecting the flow or water levels at the investigated discharges. Flow conditions were gradually varied, and therefore the energy slope $S$ was used instead of the bed slope in determining the flow resistance.

At floodplain flows, discharge and floodplain water depth ranged at 0.19 - 1.59 m³/s and 0.10 - 0.67 m, respectively, with cross-sectional mean velocities of 0.11 - 0.30 m/s. The Manning coefficient of the narrow main channel as obtained from highest flows not inundating the floodplain was $n = 0.08 - 0.12$ m$^{-1/3}$s due to irregular main channel geometry, woody debris and some aquatic vegetation.

The calculations in the present study were performed for the channel geometry and water depths, averaged over 190 m of the stream reach.

### 2.3.3 Analysis of the numerical results

The numerical results were analyzed from four perspectives: (1) identifiability of the model for the given vegetation conditions; (2) width of estimated confidence intervals as a function of the number of the observation points; (3) representation of high flows with models identified for low overbank flows; (4) the physical interpretation of the obtained parameter values.

The obtained parameter distributions were compared with measured values, as in Berends et al. (2019), but using several vegetation roughness models. This way, it was possible to analyze the problem of parameter identifiability. In the second step, the applicability of models, which parameters differ from measured values, was discussed.

The obtained uncertainty estimates of computed water levels allowed to compare the efficiency of each model in explaining the rating curve. The same output was used to measure the selectivity of models, when applied for inappropriate case, e.g.modeling of the rigid vegetation with the model for flexible vegetation. It should be expected, that the solution for the model used for the inappropriate type of the vegetation, should be characterized with the relatively high uncertainty.

The obtained results were also compared with other studies on the vegetation model identification and uncertainty estimation, like already mentioned studies of Werner et al. (2005); Dalledonne et al. (2019); Berends et al. (2019), but also Warmink et al. (2013), who compare the uncertainty of a two-dimensional model for chosen methods of bed and vegetation resistance.

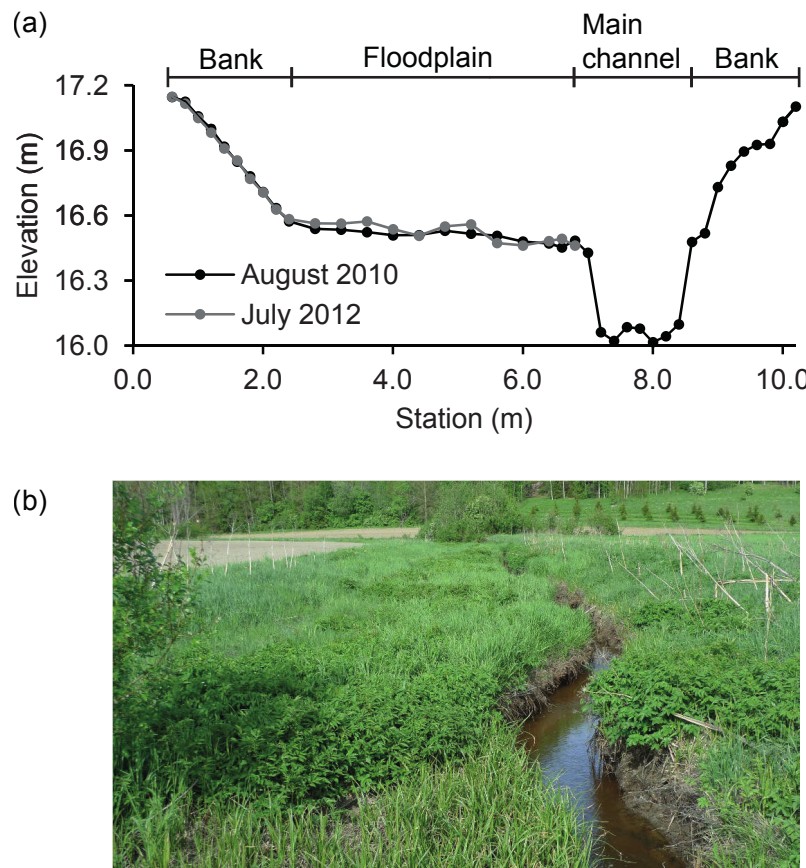

**Figure 4.** Ritobacken channel cross section (a) and a photography, Autumn 2011 (b)

## 3 Results

### 3.1 Computational output and general observations

The basic output of the computations which included Monte Carlo simulations using channel discharge models and parameter identification on the basis of Equations 1-7, were rating curves. They were derived with a different number of observation points $m$ for the parameter identification, for all possible combinations (see Section 2.1).

    Exemplary curves are presented to highlight some general observations (Figure 5). We show chosen solutions for $m = 5$ of observation points used in the parameter identification, for the two-layer approaches (GTLM, STLM, PTLM in Figure 5a-c)
developed for dense, submerged vegetation corresponding to the Ritobacken case study and for the Pasche, Mertens and Manning based DCM models for rigid emergent vegetation corresponding to the flume conditions (Figure 5d-f). In this example, chosen to provide a background for the analysis on extrapolation capabilities of models (Section 3.3), the parameters for discharge curves were identified at lower overbank flows, while the verification was conducted for highest flows. This represents

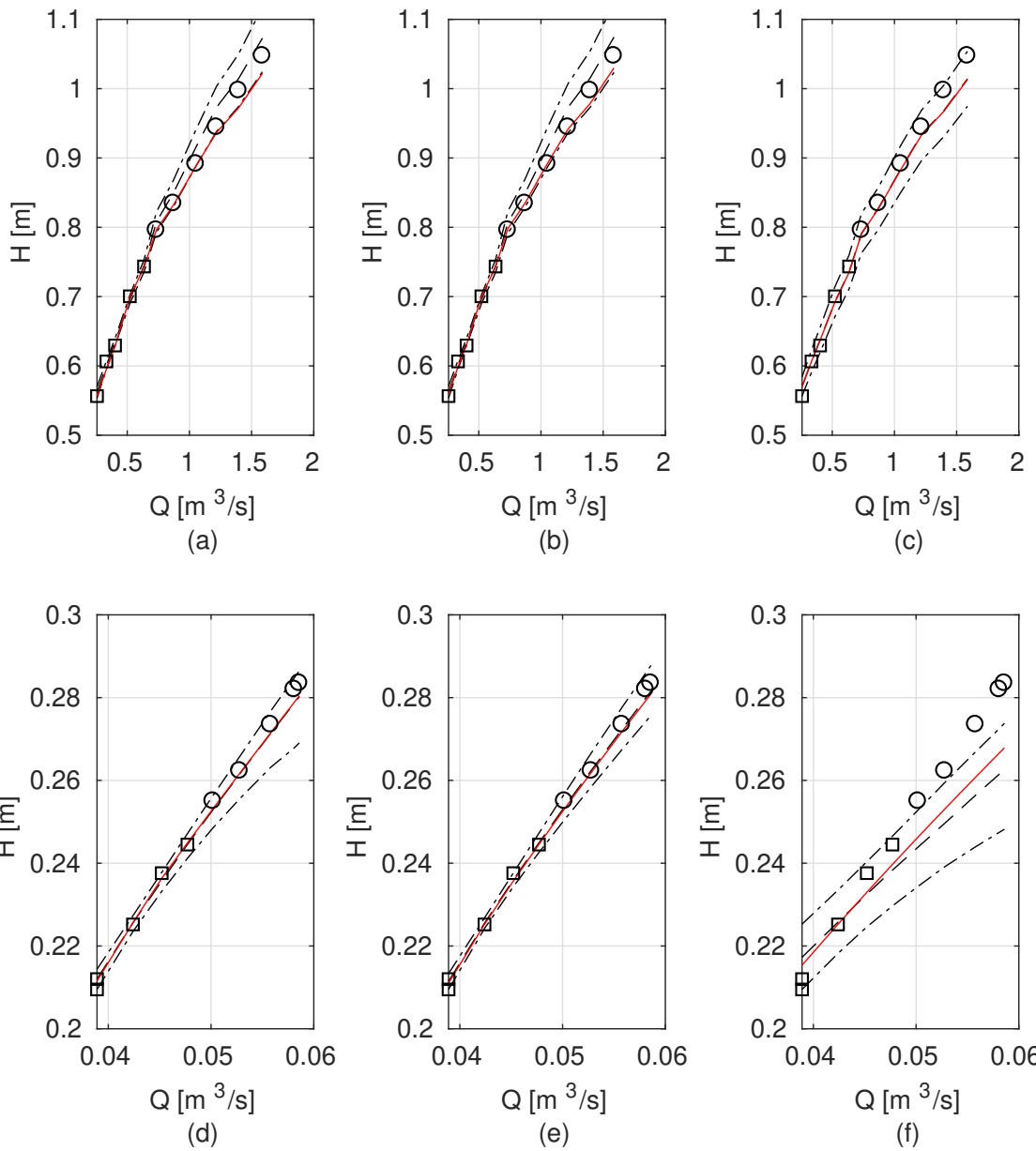

**Figure 5.** Exemplary rating curves for $m = 5$, Ritobacken case study (Spring 2012): (a) GTLM, (b) STLM, (c) PTLM; the flume data set, case 2: (d) Pasche, (e) Mertens, (f) DCM. Confidence intervals and the median of the probabilistic solution are given with dashed lines, red line denotes the best simulation in the Monte Carlo ensemble. Observation points used for parameter identification are marked with squares (□), while verification data points are marked with circles (○).

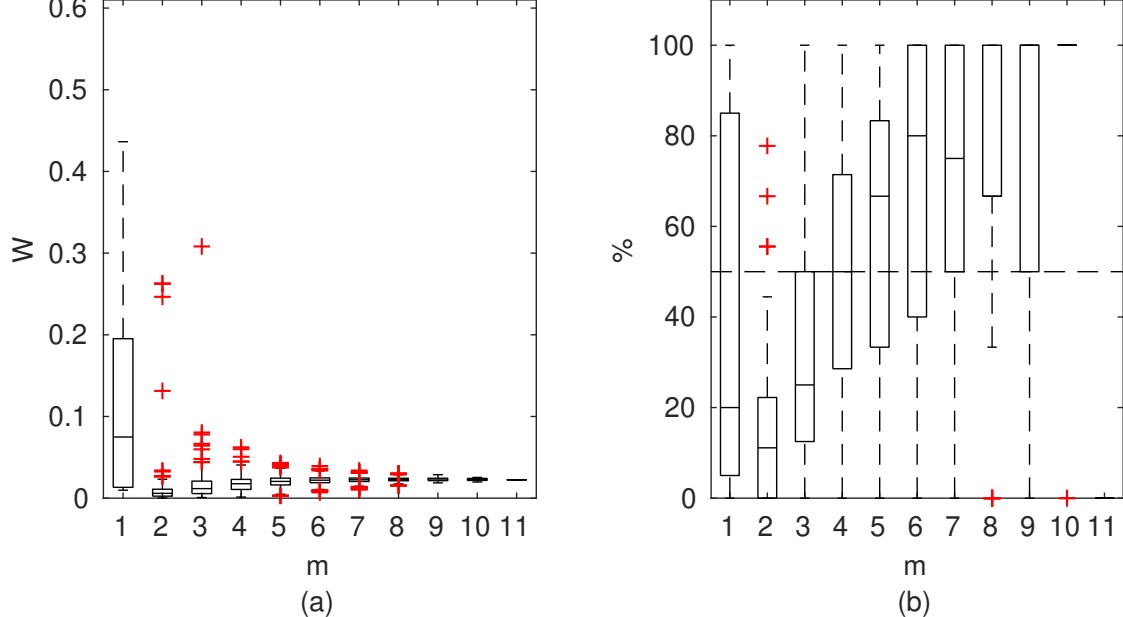

**Figure 6.** GTLM results for Ritobacken case study, Spring 2012: (a) Averaged relative confidence widths $W$ as a function of observation set size $m$ used for model identification; (b) Percentage of verification points enclosed by the confidence intervals (100% denotes all points within intervals, box spans over 25% and 75% quantile, median is given with horizontal line, whiskers indicate the result extent, cross marks are for extreme values)

the common practical way of using hydraulic models to assess flood hazard at flows higher than the ones the models were calibrated with. In terms of parameter identification results are considered as successful, as all $m$ observation points were enclosed by the confidence intervals. Except the DCM model in the flume case study (Figure 5f), all the remaining points, i.e. the verification set with $M - m$ points, given in Figure 5 as circles ($\circ$), are enclosed, indicating a good quality of the solutions. For the DCM (Figure 5f) the points used in the model identification are within confidence intervals (the condition given by Equation 8), but the verification points are outside despite the wide confidence intervals. The reason is that for the flume data with rigid vegetation, the Manning formula with constant values of roughness coefficients is unable to correctly reproduce the rating curve and fulfill the constraint given by Equation 8, which is only possible by extending the confidence intervals.

Along with the probabilistic solution, Figure 5 presents a deterministic solution obtained as a computed rating curve with the highest value of likelihood measure (Equation 3). The deterministic solution often deviates from the median of the probabilistic one, as in the case of the GTLM and STLM (Figure 5a-b).

On the basis of the rating curves computed for each combination of $m$ observation points, it is possible to analyze the estimated average widths of confidence intervals in a function of $m$ observation points used in the identification. The averaged

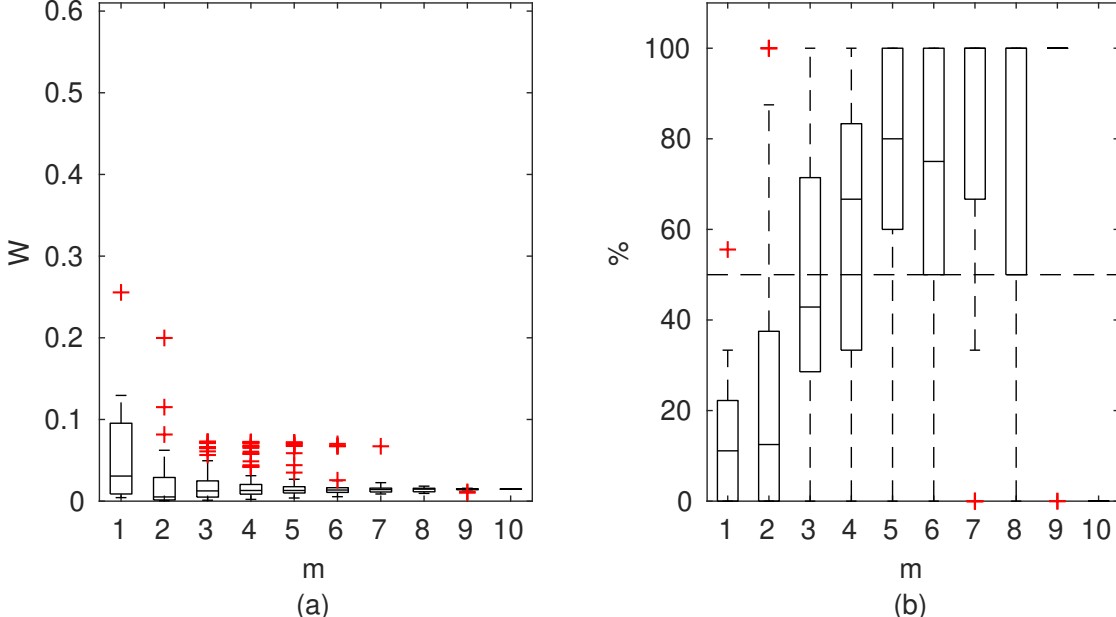

**Figure 7.** Pasche results for the flume data set, case 2: (a) Averaged relative confidence widths $W$ as a function of observation set size $m$ used for model identification; (b) Percentage of verification points enclosed by confidence intervals ($100\%$ denotes all points within intervals, box spans over 25% and 75% quantile, median is given with horizontal line, whiskers indicate the result extent, cross marks are for extreme values)

confidence widths were provided for a given $m$ in relative sizes as $W$:

$$W = \underset{m}{\mathrm{mean}}\left[ \frac{1}{m}\sum_{i=1}^{m} \frac{H_i^{q_L} - H_i^{q_U}}{\mathrm{median}\,(H)_i} \right] \tag{13}$$

where $H_i^{q_L}$ and $H_i^{q_U}$ stands for the estimates of lower and upper confidence intervals for calculated water level, normalized

for each $i$ point of the rating curve by the median of the probabilistic solution for the $i$-th point: $\mathrm{median}\,(H)_i$. From $m$ rating curve points a mean value is computed with the term $\frac{1}{m}\sum_{i=1}^{m}\frac{H_i^{q_L} - H_i^{q_U}}{\mathrm{median}(H)_i}$ for all possible combinations of $m$ observations in the full set of the size $M$. In the last step, mean values of confidence intervals widths were again averaged over sets where model was identified using $m$ observations.

      Chosen results on the influence of the number of observations used for identification on the widths of the confidence intervals

and the percentage of verification points included within the intervals are provided in Figures 6-8. In Figure 6 for GTLM applied for Ritobacken case study for Spring 2012 and also in the Figure 9 with the Pasche model used for the flume data set in case 1 it can be noticed that: (1) the relative confidence interval widths (Figures 6a, 7a) are high for a small $m$ as a result of the ill-posed inverse problem, i.e., the number of observations is insufficient for the unequivocal model identification; (2) with additional data points, the solution converges by reducing the span of intervals but also its variability due to different

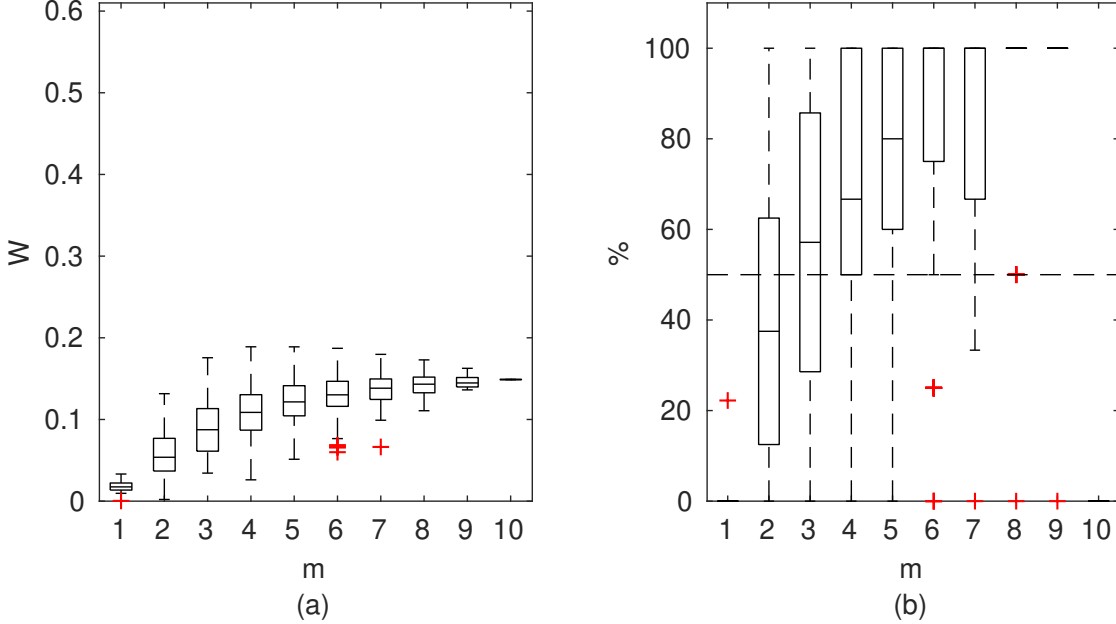

**Figure 8.** Manning based DCM results for the flume data set, case 2: (a) Averaged relative confidence widths $W$ as a function of observation set size $m$ used for model identification; (b) Percentage of verification points enclosed by confidence intervals (100% denotes all points within intervals, box spans over 25% and 75% quantile, median is given with horizontal line, whiskers indicate the result extent, cross marks are for extreme values)

combination of observation points; (3) the width of confidence intervals for the full data set $m = M$ in both cases is below 5%; (4) the confidence intervals estimated for a low number of observations ($m < 4$) have poor predictive performance, as most of the observations in the verification sets fall outside (Figures 6b, 7b); (5) in both cases for $m > 4$ more than 50% of the verification set is enclosed with the estimated confidence intervals. Figure 8 shows an example of a model with a poor performance, indicating the model's inadequacy to the given case. The confidence intervals are extending with $m$ (Figure 8a), which for $m > 4$ allows enclosing most of the verification set (Figure 8b).

## 3.2 Model identifiability

The model identifiability is understood here as the ability to determine the parameter *a posteriori* distribution that explains the model uncertainty in relation to observations (see Section 2.1). This is satisfied by meeting the constraint given in Equation 8, as for cases presented in Figure 5. The criterion of Equation 8 might be fulfilled even for a poor model by extending the parameter variability ranges (Table 1), specified with *a priori* distribution $P(\theta)$. The only limitation could be the physical meaning of the parameters.

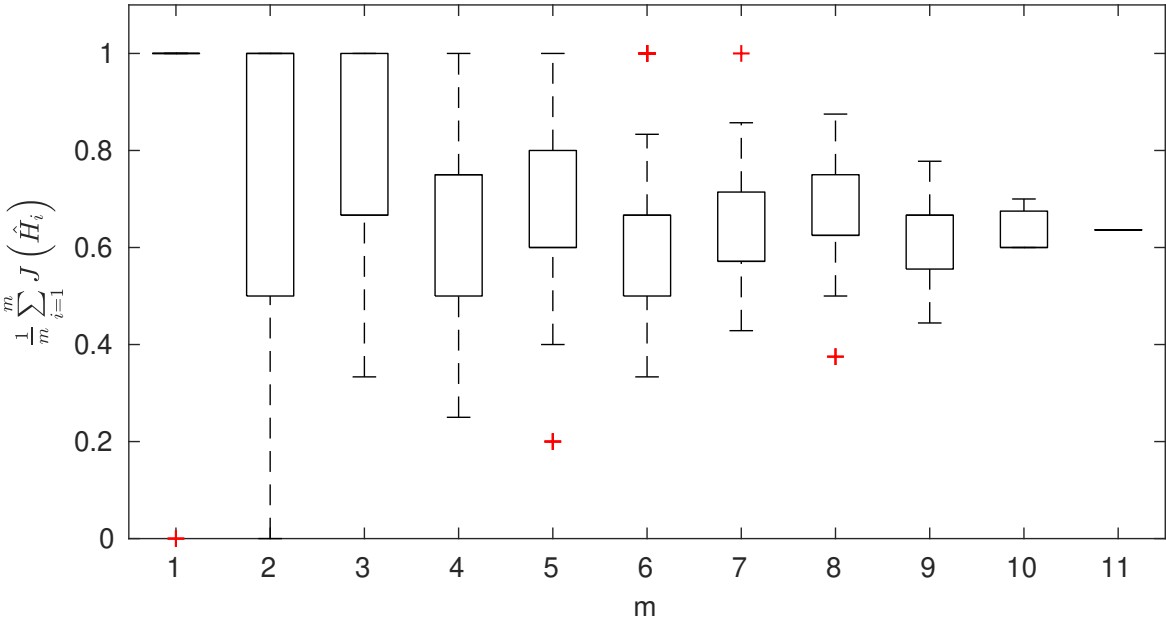

**Figure 9.** Portion of observation points within 95% confidence intervals for Pasche method in function of observation points used in parameter identification, presented in a form of box-plots; results for the unsuitable data set for the Pasche method of Ritobacken, Spring 2012.

Figure 9 shows exemplary results for a model that could not be identified for a given dataset. Values of $J$ (Equation 7) were computed for observation points used in the parameter identification and averaged in respect of their count $m$. This model was unable to correctly reproduce the rating curve over the whole Monte Carlo ensemble of parameters. The computed water levels did not follow the observed shape of the rating curve and as a result it was not possible to find such a solution of Equation 1 where identification data points would be enclosed by the confidence intervals (Equation 8). The constraint given with Equation 8 was fulfilled only for $m = 1$, but not for all points, as indicated with the single red cross in Figure 9. This indicates that not all observed water levels were covered by the Monte Carlo sample of computed water levels. With an increasing number of $m$, the number of observation points enclosed by the confidence intervals depends on the combination of observation points. Some beneficial effects allow to fulfill the constraint given with Equation 8, such as an extreme value of 1 for $m = 6$ whereas others enclose only a small share of observations. For $m = M = 11$, there is a single solution, in which about 60% observations were enclosed by confidence intervals. For an identifiable model, Figure 9 would consist of single horizontal lines between 0.95 and 1, indicating fulfillment of the constraint of Equation 8 for all simulations.

The Pasche and Mertens models applied to the Ritobacken case study were not identifiable even with relatively large variability ranges of the parameters (Figure 9). This is likely explained by the fact that these methods were developed for rigid emergent vegetation whereas the Ritobacken had mostly dense submerged flexible vegetation. The PTLM could be identified for the field site in Spring 2011 and Spring 2012 but not in Autumn 2011. This result is likely explained by the fact that the as-

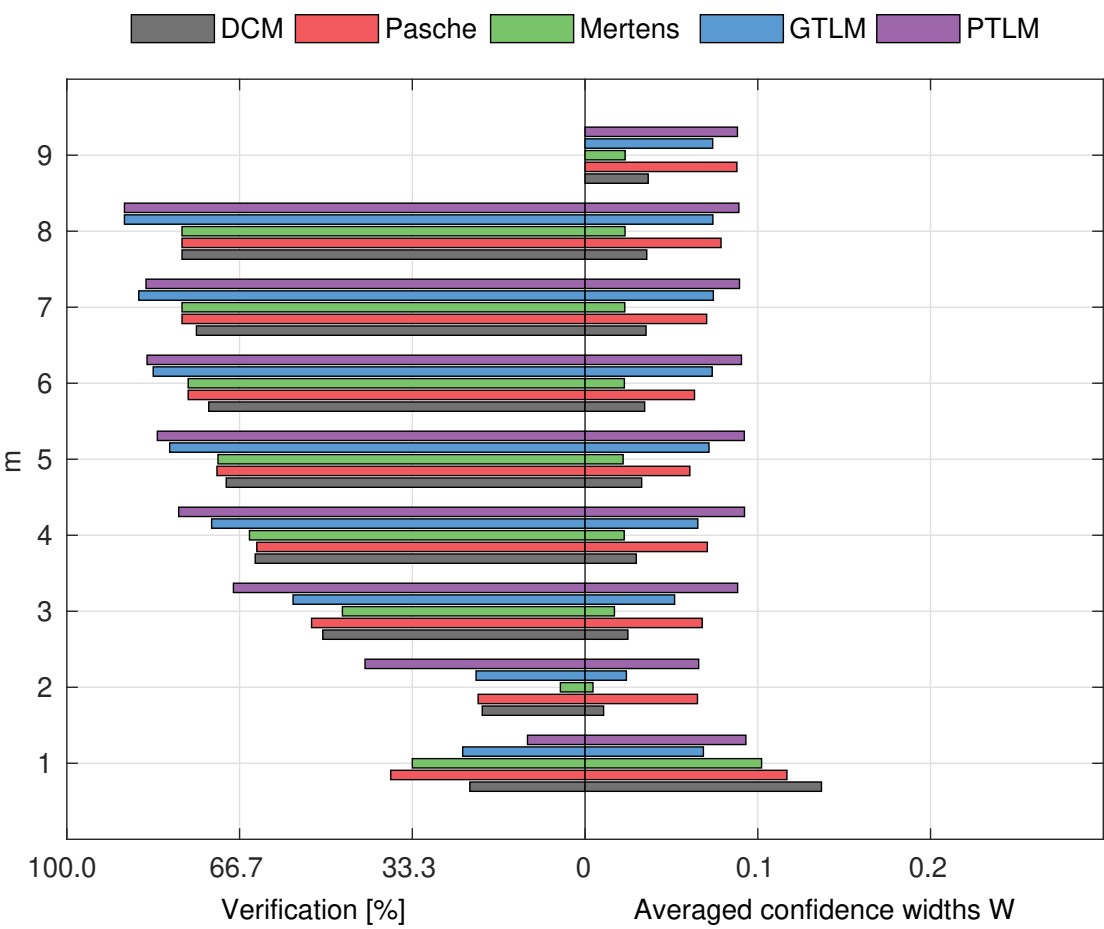

**Figure 10.** Percentage of verification set $(M - m)$ enclosed by confidence intervals and average width of confidence intervals for different number of data points for model identification $(m)$; flume data set, case 1.

sumption of $B_X \approx \frac{h}{R}$ noticeably overestimates $B_X$ in compound channels with unvegetated main channel and high floodplain vegetation, as in Autumn 2011 conditions.

By applying large parameters variability for the GTLM and PTLM models, it was possible to meet Equation 8 for the flume case study although these methods were not originally designed for such emergent vegetation. The STLM model failed for flume experiments, likely because the assumption that >80% of flow should be conveyed in the non-vegetated zones was not fulfilled. The rest of the models, including DCM for all cases, were identifiable.

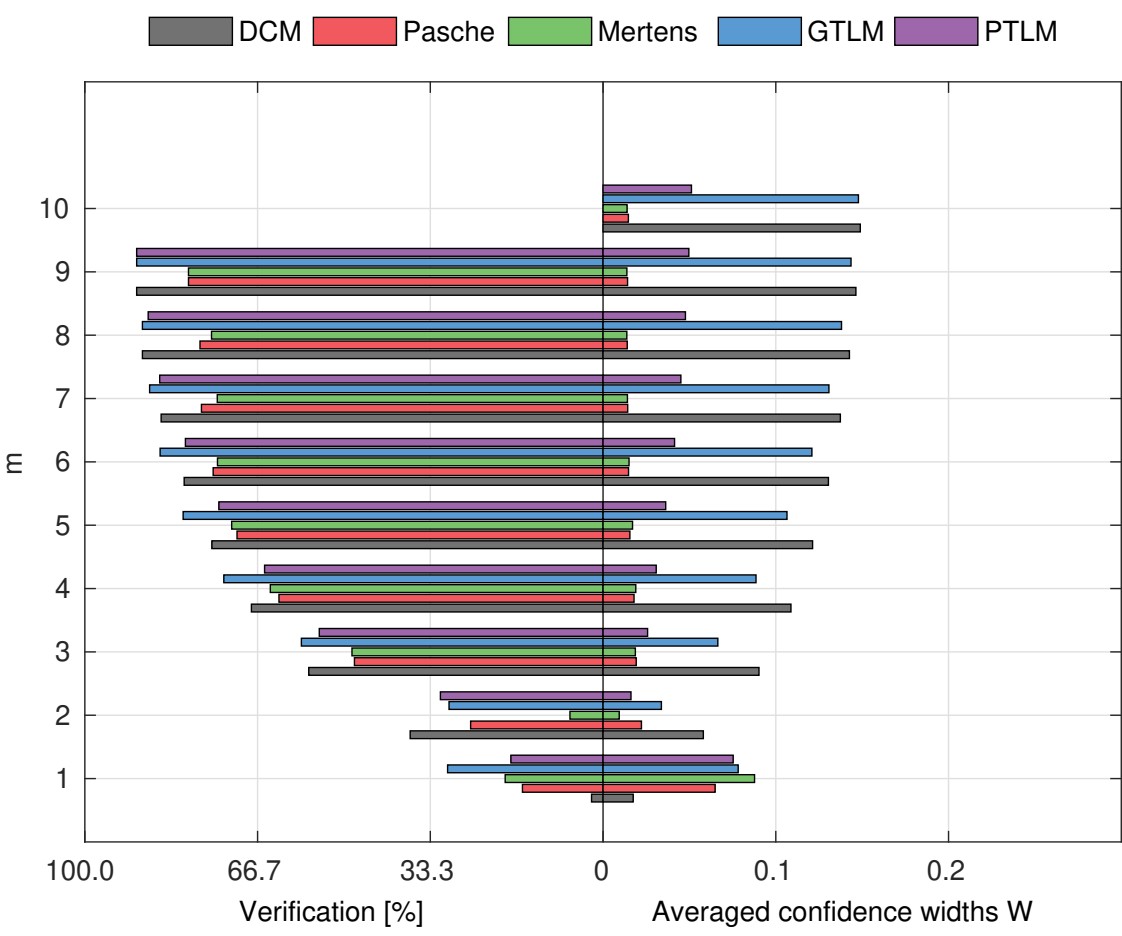

**Figure 11.** Percentage of verification set $(M - m)$ enclosed by confidence intervals and average width of confidence intervals for different number of data points for model identification $(m)$; flume data set, case 2.

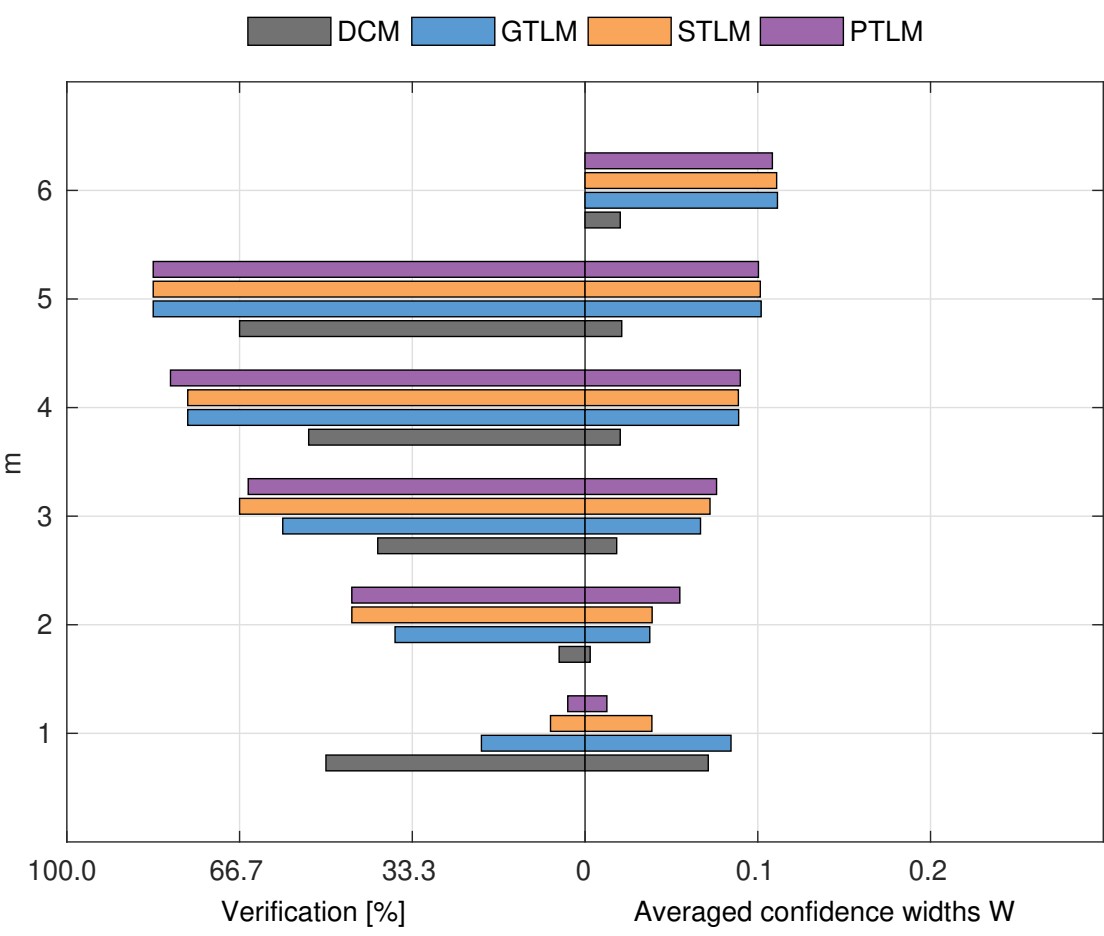

**Figure 12.** Percentage of verification set $(M - m)$ enclosed by confidence intervals and average width of confidence intervals for different number of data points for model identification $(m)$; results shown for the identifiable models for Ritobacken, Spring 2011.

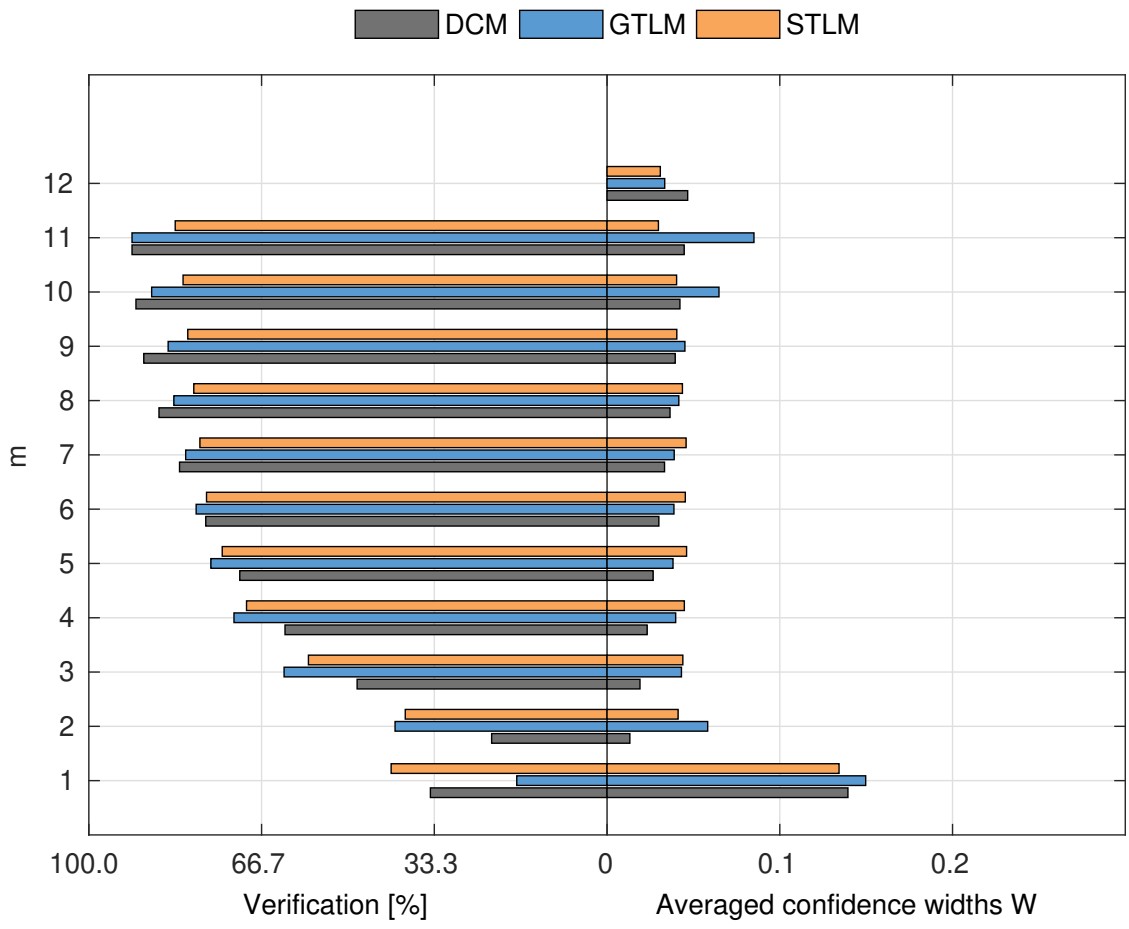

**Figure 13.** Percentage of verification set $(M - m)$ enclosed by confidence intervals and average width of confidence intervals for different number of data points for model identification $(m)$; results shown for the identifiable models for Ritobacken, Autumn 2011.

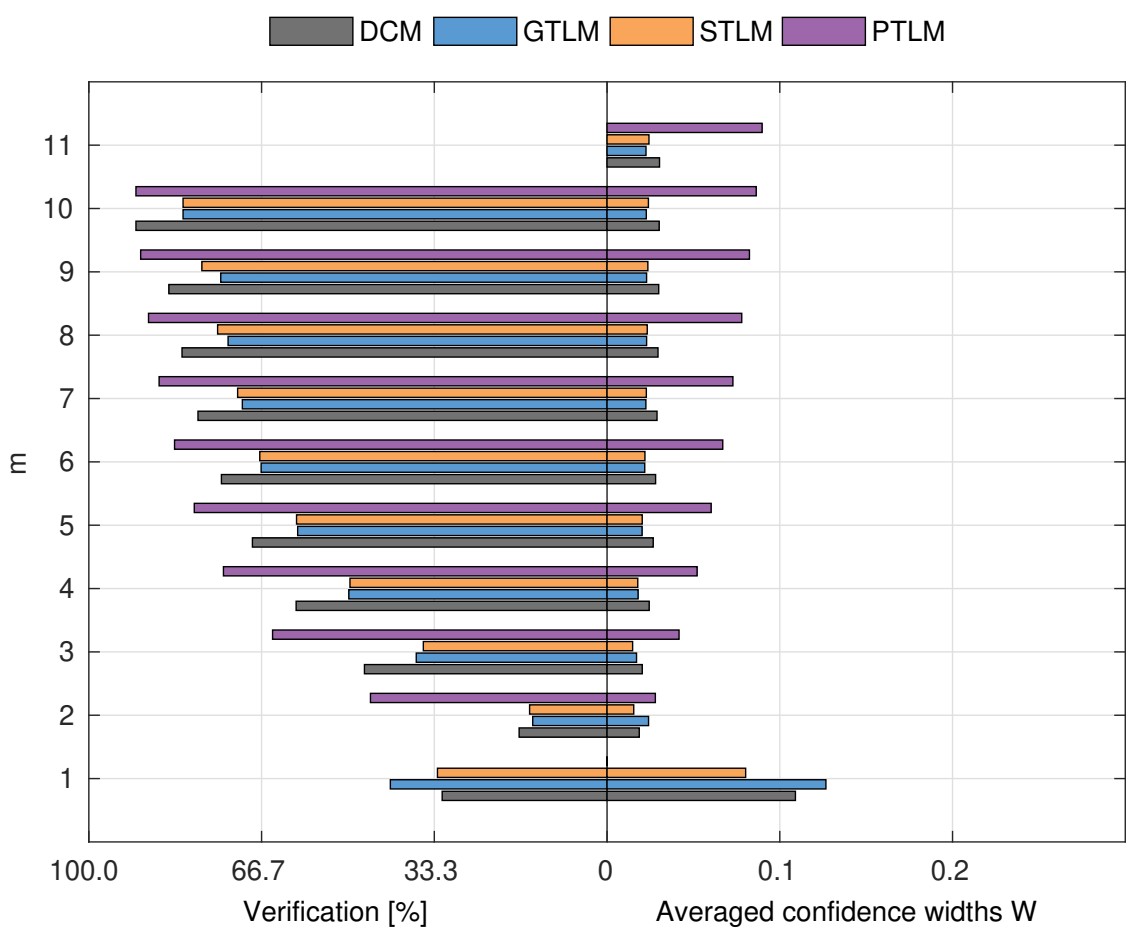

**Figure 14.** Percentage of verification set $(M - m)$ enclosed by confidence intervals and average width of confidence intervals for different number of data points for model identification $(m)$; results shown for the identifiable models for Ritobacken, Spring 2012.

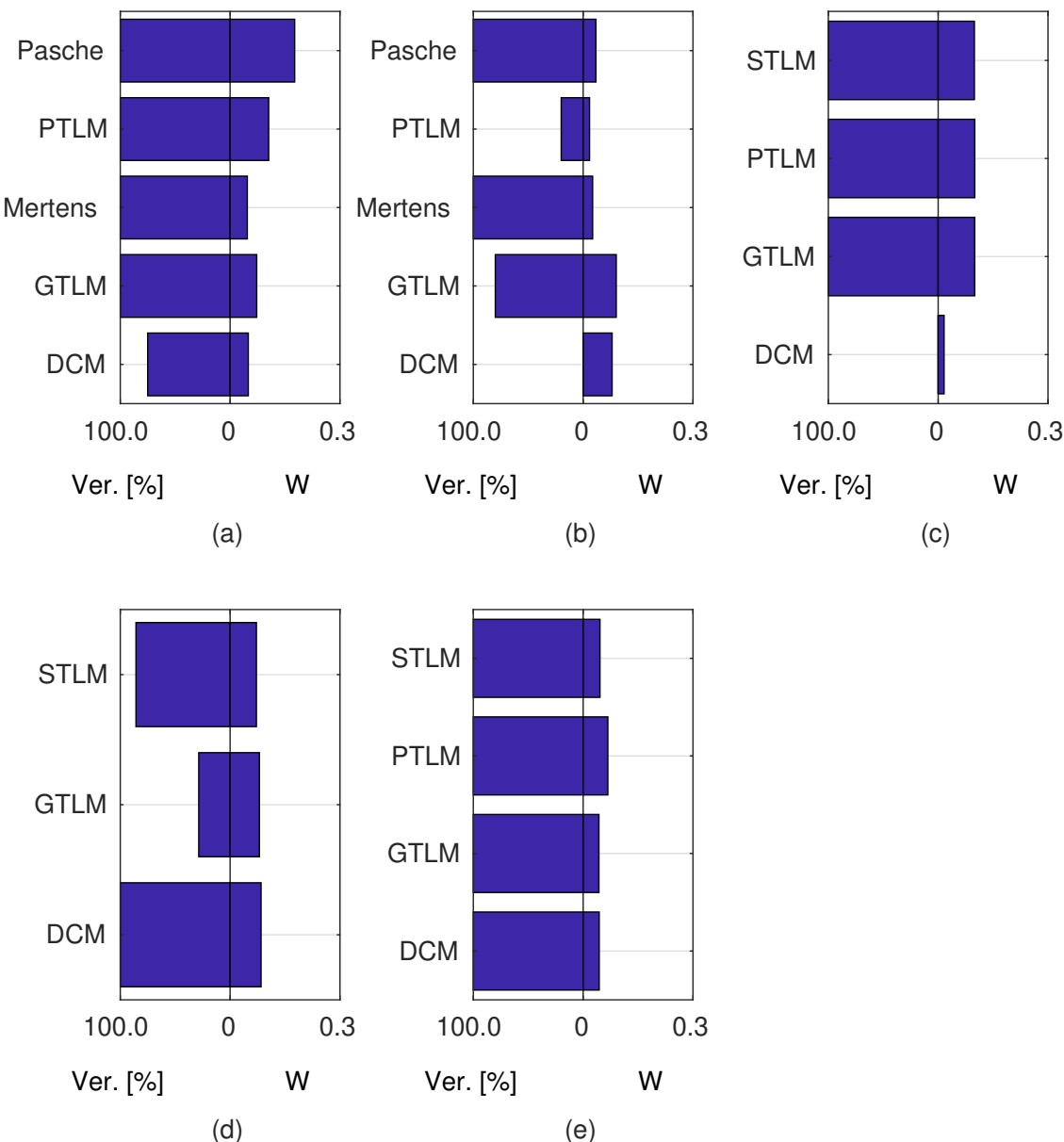

**Figure 15.** Percentage of verification points for higher flows enclosed within confidence intervals obtained with models identified for five ($m = 5$) lower flows (note, that only overbank flows were considered): (a) Flume experiment, case 1, ($M = 9$); (b) Flume experiment, case 2, ($M = 10$); (c) Ritobacken, Spring 2011, )$M = 6$); (d) Ritobacken, Autumn 2011, ($M = 12$); (e) Ritobacken, Spring 2012, ($M = 11$).

## 3.3 Widths of confidence intervals and quality of uncertainty estimation

To compare the performance of the applied identifiable discharge prediction methods, we show bar plots of average percentage of verification set points enclosed by confidence intervals and their relative widths as a function of observation points used in the model identification $m$ (Figures 10-14). The averaged values correspond to the mean values of the box-plots in Figures 6-8.

The values presented in Figures 10-14 are averaged over all uncertainty estimates at a given number of observations $m$. Therefore, for $m = M - 1$, where there was always only one verification point, the percentage for verification points can be

any value between $0 - 100\%$, not only 0 or 100%. An averaged ratio of verification points enclosed within confidence intervals, together with their relative width $W$, should be considered as a two criteria measure on how well the obtained model reproduces the discharge curve. Narrow confidence intervals indicate that the model uncertainty, estimated using $m$ observations, is small. The percentage of observations from the verification set enclosed within these intervals informs how the estimated uncertainty is representative for other data sets than these used for identification. The low percentage suggests that the model uncertainty

for the verification set is incorrectly predicted. Therefore, narrow confidence intervals for small $m$ numbers, enclosing small amount of observations should be considered as unsuccessful, as the uncertainty analysis appears to be too optimistic. On the other hand, for larger $m$, good ratios might be obtained with very wide confidence intervals, indicating a poor model. The best solution is that one, which has the narrowest confidence intervals with satisfactory percentage of verification set enclosed within it. We interpret the results by analyzing those both criteria together.

Widths of confidence intervals in a function of the number $m$ of observation points used in the model identification (Figures 10-14), allows for a qualitative analysis of the uncertainty, resulting from the insufficient data for calibration. Wide confidence intervals, and their spread for the small observation number $m = 1$ should be attributed to the ill-posed inverse problem. Additional data points allow to narrow confidence intervals and reduce their spread. The number of observations $m$ at which the widths of confidence intervals stabilizes, in some cases obtaining minimal values, suggests the point where the effect of ill-

posed inverse problem becomes less significant source of uncertainty for computed water levels. In these qualitative analyses, its effect cannot be excluded, but rather should be considered less important.

General investigations of discharge models in respect of obtaining confidence intervals were supplemented with the analysis on their extrapolation capabilities for higher flows. Figures 10-14 present averaged outcomes for models identified using all possible combinations of $m$ observations. This includes sets with only low or high but also mixed flow rates (note, that only

overbank flows are considered). In Figure 5 widths of confidence intervals and percentage of the enclosed verification set are presented for models identified only for the lowest $m = 5$ flow rates. The number of $m = 5$ observations used for the model identification was chosen arbitrary, following the impressions that this size is sufficient to minimize the uncertainty due to insufficient number of observation for the model identification (ill-posed inverse problem) and for all case studies with $m = 5$ a reasonable number $(M - m)$ of observations for verification was available.

### 3.3.1  Flume data set, case 1

For the flume data in the case 1 (Figure 10), with rigid-high vegetation in floodplains and also channel banks, the best results were obtained with the Mertens method. It is characterized with the narrowest confidence intervals $W$, having a good predictive performance. Confidence intervals for $m > 1$ were below 5% and for $m > 3$ they already enclosed more than 50% of the verification points. Almost similar performance was found for the DCM method, with slightly wider confidence intervals.

Surprisingly, both methods outperformed the Pasche model that is a very similar approach to the Mertens method, but with a much more detailed description of the vegetation induced resistance. Estimated confidence intervals widths were about three times larger than for the Mertens method and DCM, but enclosing a similar number of verification points. The reason could be the susceptibility of the Pasche method to numerical instabilities. Because of vegetation present on the channel banks, the floodplain region was extended above geometrical channel banks. This introduces discontinuity to the hydraulic radius in floodplains, as water levels slightly exceed geometrical banks. Probably, this might lead to numerical instability of implicit formulas used in the Pasche method, but not present in the Mertens method. GTLM and PTLM confidence intervals were similar to the Pasche, but enclosed even more observations than Mertens. However, confidence intervals for Mertens are almost three times narrower and this method should be considered as the most appropriate in this case.

Figure 15a, presents the results for models identified using the lowest $m = 5$ flow rates. The Mertens model with the smallest estimated uncertainty was capable explaining the rating curve for all verification points. Other models, except the DCM, allowed to enclosed whole verification set, but with much wider confidence intervals.

### 3.3.2  Flume data set, case 2

For the flume case 2 (Figure 11), both the Pasche and Mertens methods appear to be the most effective. Estimated widths of confidence intervals do not exceed 4-5% for $m > 1$ and fell bellow 1-2% for a sufficient number of observations ($m > 5$). The predictive skills of the identified models are high, with around 70% of the verification set enclosed by the confidence intervals at $m > 4$. GTLM has a similar uncertainty performance as DCM while PTLM provides noticeably much more narrow uncertainty estimates. For GTLM and DCM, the final confidence widths for $m = M$ are about 15% and for PTLM 5%. Because of their larger extent, the estimated intervals enclose slightly larger number of verification points than with the Pasche and Mertens methods. The DCM has three times wider confidence intervals than for flume case 1. The main difference between the flume cases 1 and 2 was the rough floodplain surface with the grain sizes of 0.5-1 cm for the case 2 compared to the smooth floodplain of case 1 indicating that the D was not able to perform reliably for the combination of rough surface and emergent vegetation.

Figure (11) highlights the specific dependency of DCM, GTLM and PTLM on $m$. For a small number of data points for a model identification at $m = 1$, confidence widths are high, because of the ill-posed inverse problem. With additional points, the effect is reduced, and for $m = 2$ the confidence interval widths are at their smallest but with poor predictive skills. With increasing $m$ the uncertainty estimates are corrected by additional data points. The same pattern is present but less noticeably for the Pasche and Mertens methods and for the other cases.

As in general output, Pasche and Mertens models provided the best results, when identified for $m = 5$ lower flows (Figure 15b). Their confidence intervals, more narrow for Mertens model, enclosed 100% of verification set. Performances of the Manning based DCM are here poor, as despite relatively wide confidence intervals, it appeared impossible to explain any of verification points. In Figure 5d-f rating curves for Pasche, Mertens and Manning based DCM were presented for this specific calibration case.

### 3.3.3  Ritobacken, Spring 2011 case

The Spring 2011 case study refers to flow conditions with poorly developed vegetation one year after the floodplain excavation. These conditions with low vegetation having a mean relative submergence (floodplain water depth divided by vegetation height) of 3.3 are reflected in the computational output (Figure 12), with process-based methods for vegetation resistance characterized with a relatively poor fit.

All three two layer models (GTLM, STLM and PTLM) have very similar performances, but with noticeably wider confidence intervals than the DCM, with $W$ of 12% to 3%. The percentage of enclosed verification points at $m > 2$ is better for two layer approaches, although difference is small (single observation point). The picture is different in the case of the Figure 15c presenting the extrapolation capabilities of the methods. Widths of confidence intervals of two-layer models are similar to averaged values at $m = 5$, given in Figure 12 and enclose all verification point (note, for Spring 2011 $M = 6$). DCM's narrow confidence intervals were unable to enclose the verification points.

### 3.3.4  Ritobacken, Autumn 2011 and Spring 2012 cases

Ritobacken Autumn 2011 and Spring 2012 case studies reflect the influence of seasonal differences of vegetation on the flow conditions. In Autumn 2011 vegetation was higher and denser than before and at the beginning of the growing season in Spring 2012. This can be seen in the performance of the applied discharge methods. For the fully vegetated conditions of Autumn 2011 (Figure 13), all the identified methods enclosed over 70% of the observations at $m > 5$ with $M = 12$. STLM has narrowest confidence intervals (4%) when all data was used for model identification. STLM had slightly lower percentage of enclosed verification points, comparing to DCM with also very narrow confidence intervals and GTLM with somewhat wider ones. For the Autumn 2011, it was not possible to identify the PTLM.

For the Spring 2012 (Figure 14), DCM, STLM and GTLM have almost equal confidence widths and ratios of enclosed verification points while PTLM has very wide confidence intervals. The overall measures are similar to those from Autumn 2011. The confidence widths for DCM, GTLM and STLM are about 3% and for $m > 5$ and more than 70% of points fall within confidence intervals. PTLM has slightly higher ratio of verification data enclosed, compared to the other methods, because of notably wider confidence intervals of 8-9%.

In the calibration case with the lowest $m = 5$ flow rates, for Autumn 2011 (Figure 15d), a high explanation of the rating curve was obtained with the STLM and Manning DCM. Poorer results for Autumn 2011 set were obtained for the GTLM, with low percentage of verification points enclosed. For Spring 2012 all two layer models (GTLM, PTLM and STLM) and also the Manning DCM allowed obtaining a very good explanation of the rating curve, when identified for the lowest $m = 5$ flow rates

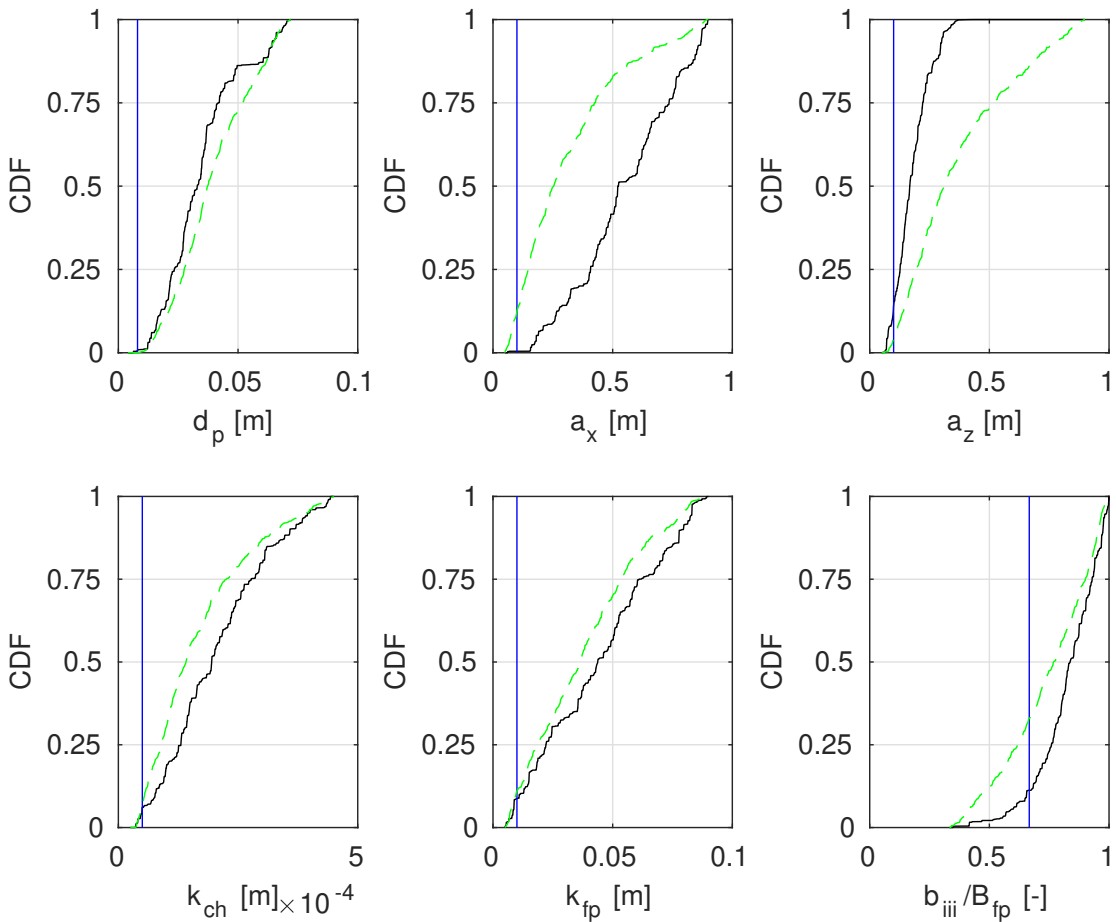

**Figure 16.** Marginal *a posteriori* distributions of Pasche (black lines) and Mertens (green lines) models parameters, identified using $m = M$ observation points for flume experiment, case 2; measured parameters values were provided with blue lines.

(Figure 15e). The rating curves of the GTLM, STLM and PTLM in this calibration case for Spring 2012 were presented in Figure 5a-c.

### 3.4   Physical interpretation of identified parameters

*A posteriori* parameters distributions $P\left(\theta/H\right)$ can be presented in a form of marginal Cumulative Distribution Functions (CDF). The CDF is plotted over the sampled parameter range, given in Table 1. The shape of the marignal CDF indicates the

likelihood of given parameter values. The linear dependency would mean that all values are equally likely in respect of the likelihood function (Equation 3). On the other hand, a strong CDF skewness characterizes regions of a high probability and

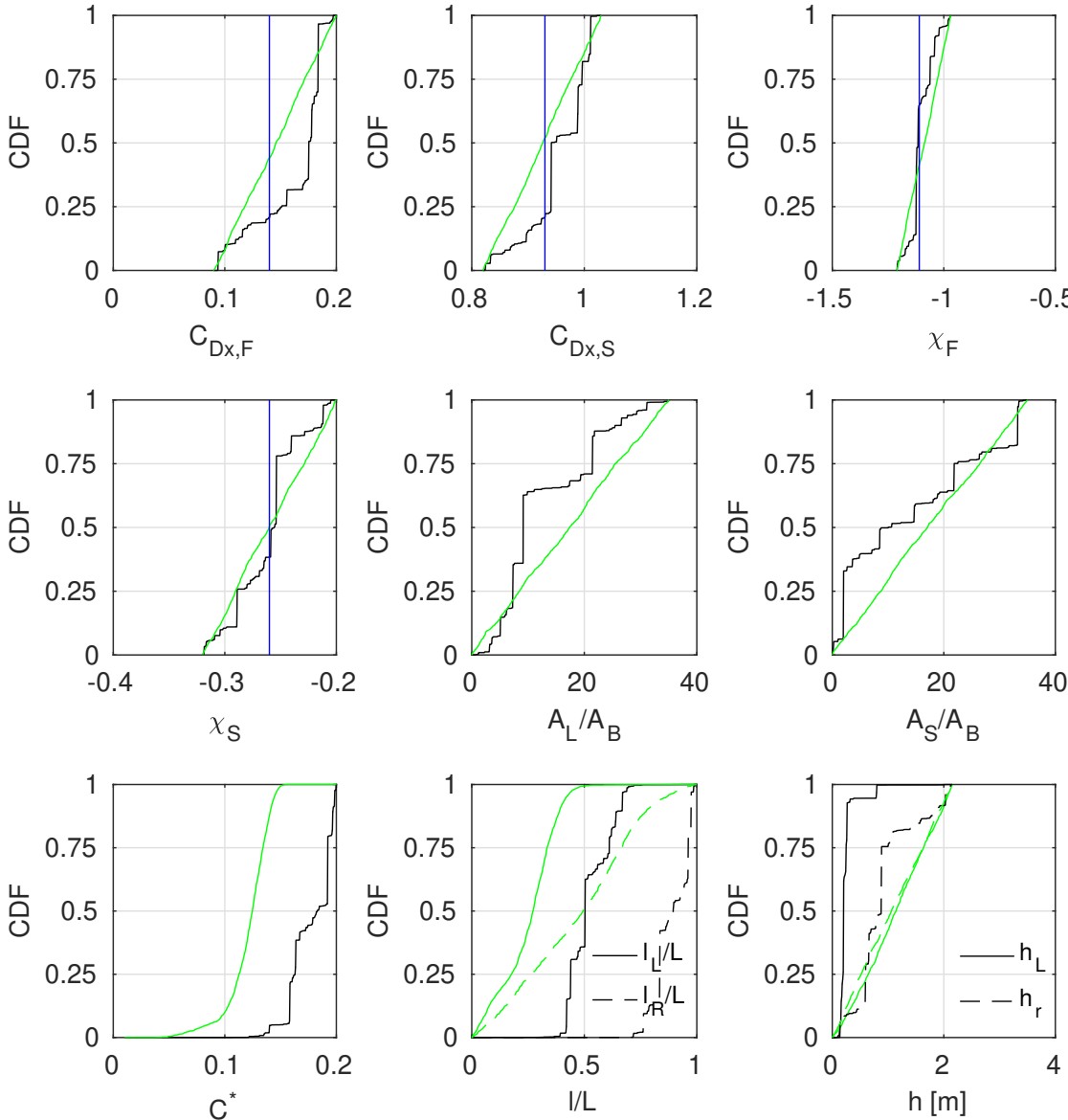

**Figure 17.** Marginal *a posteriori* distributions of GTLM model parameters, identified using $m = M$ observation points in the Ritobacken case study; black lines stand for Autumn 2011 set and green for Spring 2012; parameters values given by Västilä and Järvelä (2014) for woody vegetation were provided with blue vertical lines

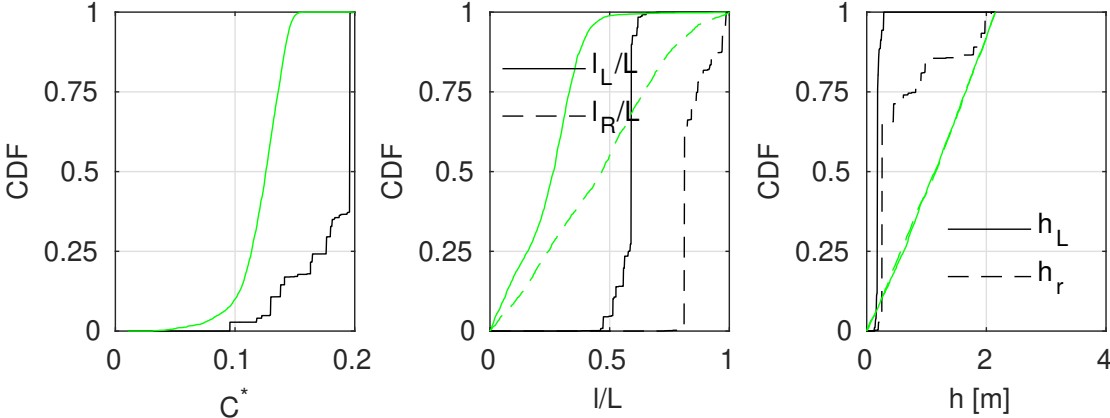

**Figure 18.** Marginal *a posteriori* distributions of STLM model parameters, identified using $m = M$ observation points in the Ritobacken case study; black lines stand for Autumn 2011 set and green lines for Spring 2012

larger model sensitivity on the parameter. The *a posteriori* marginal CDF of parameters were presented for four vegetation roughness models: Pasche, Mertens, GTLM and STLM. Parameters of Pasche and Mertens models (Figure 16), were given for the flume case 2, where both models explained the rating curve very well. GTLM and STLM parameter estimates (Figures 605 17-18) were compared for the Ritobacken Autumn 2011 and Spring 2012 sets, as both models were found here appropriate and additionally, it was possible to analyze the seasonal vegetative differences on parameter estimates (see Section 3.3.4). In all cases, solutions for all observation points $m = M$ were used.

In Figure 16 the CDF for Pasche parameters for the flume case 2 is given with black lines and green lines for Mertens. Measured values of parameters are provided with blue lines. The steep shape of the CDF for the Pasche $a_z$ indicates a strong 610 model sensitivity on the parameter and that the values above $\sim 0.3$ m are unlikely. For the Mertens model, a similar effect, but with smoother CDF is present for both $a_x$ and $a_z$. The differences in the case of these particular parameters comes from the more complex structure of the Pasche model, restricting values of $a_z$, due to lack of a numerical convergence for its implicit formulas. For both Models (Figure 16) $b_{III}/B_{fp}$ appears to be a sensitive parameter, while the response for remaining parameters is more uniform.

The strongest discrepancies between measured and identified values of parameters of Pasche and Mertens models (Figure 16) are present for the stem diameter $d_p$ and longitudinal stem spacing $a_x$. A median (at CDF 0.5) of the probabilistic solution for $d_p$ is close to 0.04 m, while the real diameter was 0.008 m. In the case of $a_x$ it is 0.6 m for Pasche and 0.25 m for Mertens to 0.1 m. This has a clear physical sense, as in terms of the model identification, small stem diameters $d_p$ at dense spacing with small $a_x$ were equivalent to larger $d_p$ and smaller $a_x$. This finding is supported, by much smaller discrepancies in other 620 parameters. It should be noted, that the measured parameter values provide a fit close to the best one in a deterministic sense (Kiczko et al., 2017).

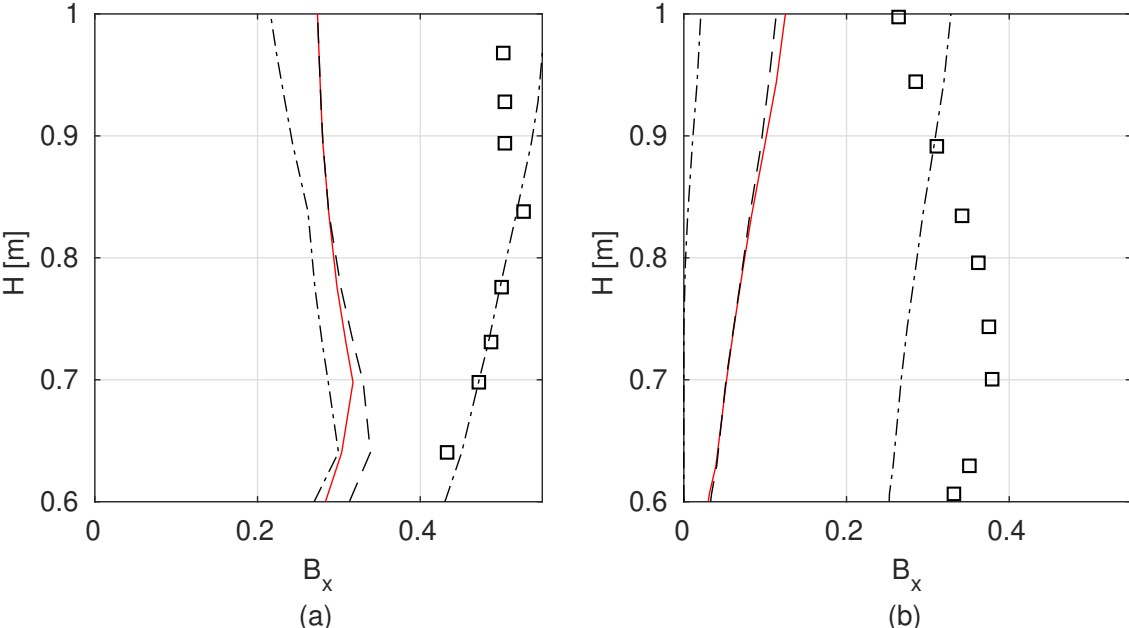

**Figure 19.** Blockage factor $B_X$ measured in the field and determined as an inverse solution of GTLM for Ritobacken Autumn 2011 (a) and Spring 2012 (b) case study; squares denote measured values, dashed lines – confidence intervals and median of a probabilistic solution, red line – the best simulation in the Monte Carlo ensemble.

In Figure 17 results for the GTLM model identified for the Ritobacken Autumn 2011 (black lines) and Spring 2012 (green lines) are provided. It can be seen that in both cases, the identified values of the parameterization for flexible vegetation (Equation 11) had a fairly narrow distribution for the reconfiguration ($\chi$) of the foliage, which fell close to the values observed for willows and other woody species (e.g. Västilä and Järvelä, 2018). In the case of remaining parameters it can be noticed, that for the Autumn 2011 set, the CDFs have a step-shape, clearly indicating more likely regions. For example, the most probable values of the steam reconfiguration coefficient $\chi_S$ for Autumn 2011, are very close to the observed ones. The same applies to $C_{Dx,S}$ and $C_{Dx,F}$. In all these cases, CDFs suggests also other highly probable regions, different from expected ones, e.g. for $\chi_S$ also values close to 0.3, were considered as very likely. The effect, also seen clearly for $A_S/A_B$, $A_L/A_B$, $C^*$, $C_{Dx,S}$, $h_L$ and $h_R$ is an example of parameter equifinality. Distributions obtained for the Spring 2012 set are much more uniform, without values that can be considered as highly probable.

Similar to the Pasche method, not all distributions follow the expected values. The CDF for $C^*$ in Autumn 2011 shows notably larger values than experimentally derived ($C^* \sim 0.034 - 0.08$, Västilä et al., 2016). For Spring 2012 $C^*$ values are much closer to the expected ones, but it is hard to find an explanation of the differences when Autumn 2011 case is considered, other than the effect of ill-posed inverse problem, where water depths are insufficient for identification of this parameter.

Wider ranges for the vegetation heights $h$, extents $l/L$ and frontal projected areas of stems $A_S/A_B$ and leafs $A_L/A_B$ in the Spring 2012 set, may be associated with lower vegetation roughness in that period (Västilä et al., 2016). The solution providing a good representation of water depths might be obtained for different combinations of these parameters, such as too small $h$ with too large $l/L$. Higher autumn flow resistance, resulting in a different shape of the rating curve, appeared to be more restrictive for these parameters.

Parameters of the STLM are given in Figure 18. As in this approach flow in the vegetation layer is neglected, it includes less parameters than GTLM; $l_L/L$, $l_R/L$, $h_L$, $h_R$ used for parametrization of the blockage factor $B_X$. The obtained CDFs are very similar to those for the GTLM (Figure 17). As previously, parameters of the Autumn 2011 are much better defined. Again a noticeable shift in $C^*$ can be observed for Autumn 2011. Such a good agreement between obtained parameters for GTLM and STLM, together with very similar uncertainty estimates (Figures 13-14) suggests that flow within vegetation layer was not significant for the shape of the discharge curve under the analyzed conditions. Otherwise, the shape of GTLM CDFs would be noticeably different as a result of interactions with parameters characterizing flow in vegetation layer.

Studies of Västilä and Järvelä (2018) provided estimates on the blockage factor $B_X$, which allow comparison to the results of model identification through calculating confidence intervals for modeled $B_X$ on the basis of identified parameters $l_L/L_L$, $l_R/L_L$, $h_L$ and $h_R$ for Autumn 2011 and Spring 2012 (Figure 19). The confidence intervals for the $B_X$ are wide and the observed values are shifted from the median of a probabilistic solution towards 0.9 quantile. The noticeable under-estimation of the $B_X$ by the model identification likely decreases the performance of GTLM for the field case, since under partly vegetated conditions the cross-sectional vegetative blockage has been found the most important property in determining the flow resistance (e.g. (Green, 2005), (Luhar and Nepf, 2013). A large spread of values for $B_X$ with very small variation of water levels for that solution (Figure 13) suggest a moderate model sensitivity on $B_X$, affected by interactions with other parameters.

## 4 Discussion

The present study is according to our knowledge the first one, where different discharge capacity methods were compared in respect of their uncertainty, estimated along with model parameters, using probabilistic formulation of the problem of the parameter identification. The noticeable focus was made to ensure that the uncertainty analysis was objective and repeatable. The novelty of the proposed approach includes the analysis of obtained confidence widths, together with the percentage of independent observations explained by them, with respect to the number of observations used in the model identification. The results confirm previous findings of (Kiczko and Mirosław-Świątek, 2018; Kiczko et al., 2018; Romanowicz and Kiczko, 2016), that for discharge formulas the probabilistic solution differs from the deterministic one. This is evident from Figure 5 for calculated rating curves. This obvious behavior of nonlinear models highlights the needs for such uncertainty analyses.

Our results show that the number of parameters seems not to be a factor precluding the identifiability of vegetation roughness models. It was possible to identify a model with more than ten parameters (i.e. GTLM accompanied with a parameterization of complex reconfiguring vegetation), almost as well as three-parameter ones (DCM). In the most cases, the ill-posed inverse problem appears affecting the uncertainty estimates only when the number of observation points was very small, independent

of the number of parameters. Widths of confidence intervals stabilized close to the final extent at about three-four observation points ($m > 3$, Figures 10-14). The process-based methods have more parameters, than the required number of observations, necessary for the identification. This suggests the ill-posed problem, but might be explained with a low model sensitivity to groups of parameters, seen in the marginal CDF of the *a posteriori* parameter distributions (Figures 16-18) and in the result the model fit depends on only several parameters. The observations are however different for the field case with the most developed vegetation, Ritobacken in Autumn 2011, where the uncertainty estimated for the GTLM, with the largest number of parameters, falls below levels obtained for the DCM only for the full set of observations used for the model identification. In this case the GTLM was found very sensitive on parameters characterizing flow in the vegetation layer (Section 3.4) and a noticeably larger number of observations was necessary to restrict variability of parameters.

Our findings indicated that the performance of a model depends on its adequacy for the given vegetative and flow conditions. For emergent sparse rigid vegetation, the most reliable method was the Mertens model with mostly explicit formulas. Because of a simpler numerical form than in the Pasche method, the Mertens method was less vulnerable to numerical instabilities, which probably affected the outcomes of the Pasche uncertainty estimation. In the case of dense mostly grassy vegetation typically observed on natural floodplains (Figure 4), the most reliable performance with respect to uncertainty estimates was obtained with the simplified two-layer approach (STLM), which neglects the flow in the vegetation layer (Figures 12-14). The full two layer model (GTLM) also provided a reasonable representation of the rating curve for flexible vegetation, although with higher estimated uncertainty, probably because of a larger number of parameters. For all cases, except Ritobacken Spring 2011 with the least developed vegetation, the best performing process-based method produced narrower confidence intervals than the DCM, when the models were identified with all observation points. Further, for the field conditions, the predictions of the validation dataset were notably better with the process-based models compared to DCM when the number of data points used for model identification was low (2-4) while the confidence intervals were reasonable for practical applications.

An important aspect when comparing the different methods is their general applicability for different channel conditions. Despite the larger number of parameters, the process-based methods were less generally applicable than the Manning based DCM approach, which could be identified and thus applied in all cases. Pasche and Mertens methods were only applicable for the sparse rigid emergent flume vegetation, for which they were derived. By contrast, the two-layer approaches GTML and PTML, although it was possible to identify them, had a less favorable performance when applied to the flume vegetation (Figure 3). Further, our findings appeared to confirm that the the STLM is strict about the assumption that less than 20% of the flow is conveyed within vegetation (Section 3.2). The STLM could not be identified for the flume conditions with sparse vegetation likely resulting in substantial flow on the floodplain. The results for the DCM with constant values of the Manning coefficient were quite good except for flume case 2, indicating that the process-based methods are expected to perform better and more reliably than DCM when several important sources of flow resistance, such as rough floodplain surface and sparse emergent vegetation are present. These methodological findings suggest that it could be possible to choose an appropriate method on the basis of its goodness-of-fit measures and uncertainty estimates.

For practical channel design or flood inundation estimation cases, the capability to extend the model calibrated with observations at low flows to high flows is crucial. Of the six models, none provided good extrapolation results under all tested cases.

GTLM was the most reliable model as it performed reasonably in four of five cases, and thus across a wide range of vegetative

conditions (Figure 15). The GTLM parameterized at low flows successfully predicted the more rapid increase in discharge at water levels exceeding vegetation height (Figure 5a), except the Autumn 2011 set. For instance, the DCM was in two of the five cases unable to reliably predict the water levels at higher discharges when optimized based on observations at lower discharges (Figure 15). The overestimation of channel flows (Figure 5f) is a known feature of the DCM with constant Manning coefficients, as it does not account for the momentum transfer between the main channel and floodplains (Myers, 1978).

The GTLM was in this paper amended with a vegetation parameterization (Equation 11) that describes the influence of the plant streamlining and reconfiguration on flow resistance. Although Equation 11 has been developed for woody vegetation, it was applicable to the predominantly grassed vegetation at the field site. Field surveys indicated that much of the plants consisted of a main stem and more flexible leaves, conceptually similar in behaviour to foliated woody vegetation. Equation 11 describes the drag from stem and leaves and allows to set different values for the flexibility-induced reconfiguration for the stem and foliage. By setting the reconfiguration parameters to 0, the model can be used for rigid vegetation, which might explain the applicability of the model in flume cases with rigid vegetation.

Further justification of the wide applicability of the two layer modelling concept is not straightforward with the obtained results. Shields et al. (2017) suggested that two-layer models based on the Luhar and Nepf (2013) concept allow for a better representation of the transition from the submerged to emergent flows, in which case the cross-sectional vegetative blockage and the bulk flow resistance typically start to decrease. Obtained CDF of *a posteriori* parameters distributions for STLM and GTLM suggest that this effect might be important. For the Autumn 2011 case, with well developed vegetation, the most probable solution included moderated vegetation heights and larger extents ($h_L$ and $h_R$, Figure 17), which ensures that transition from submerged to emerged vegetation is present. On the other hand, this effect was not observed for other cases.

Put together, our various analyses show the advantages of the more complex process-based methods over the Manning-based DCM. The results agree with Dalledonne et al. (2019), who obtained the narrowest uncertainty estimates for the more complex models. Besides being applicable to flood water level estimation, the described process-based models allow predicting the influence of different channel management scenarios on water levels. The methods are expected to be helpful in planning common practical management measures for vegetated compound channels, such as cutting of the floodplain and bank vegetation as well as maintaining the channel through dredging the main channel or lowering the floodplain. Improved reliability of the discharge capacity estimates may help in decreasing unnecessary, environmentally disruptive management actions, and allow to plan more sustainable alternatives, such as partial cutting.

We found that the differences between the one-dimensional methods were notably larger than for the study of Dalledonne et al. (2019) focusing on a two dimensional model. Further, the Warmink et al. (2013) study did not consider the choice of the flow resistance parametrization method as crucial. The presently investigated flume and field cases had a notable portion of the cross-section covered by the floodplain vegetation, with Manning's $n$ ranging at 0.017-0.150 $\mathrm{m}^{-1/3}$. Thus, our results indicate that the choice of the resistance formula is important for cases where vegetative resistance dominates. On the other hand, one-dimensional models may be more sensitive to uncertainty related to the identification of the resistance parameters than are two-dimensional models.

The most important issue is the physical interpretation of parameters obtained by the model identification. As expected, on the basis of previous studies of Werner et al. (2005); Berends et al. (2019) the obtained values, showed in a form of CDF of marginal *a posteriori* distributions in Figures 16-18 differs from measured ones. This results from the parameter equifinality. One of the reasons might be insufficient observation sets used in model identification. The likelihood function, conditioned only on water levels is not capable to restrict variability of parameters in more complex vegetation roughness models. It can be seen in the shape of the marginal CDF of parameters, presented in Figures 16-18, suggesting small sensitivity of the model on given parameters, except only the Ritobecken Autumn 2011 case. Their variability can be probably reduced by additional data sources, as discussed in hydrological studies of Her and Chaubey (2015); Her and Seong (2018). For channel flows it could be velocity measurements, used e.g by Berends et al. (2019) for model identification. It should be however noted, that in practical assignments on a flood hazard, such data is rarely available. The other reason of parameter equifinality and therefore discrepancies with measured values of parameters are parameter interactions. The shift in a given parameter is compensated by others, e.g. the large stem diameter $d_p$, observed for Pasche and Mertens models, comes along with too large spacing of plants $a_x$ and $a_z$. Such an effect is probably present in all process-based models, identified in terms of an inverse problem.

The inability to specify parameters of process-based methods by model identification is an argument against such an approach, already signalized by Werner et al. (2005). Moreover, with parameters different form real values, the use of these complex models, rises an impression of black-box modeling, as the identification goal is only to obtain a satisfactory fit and uncertainty estimate. With outcomes of the present study, it is hard to address this problem directly, as it would require comparing process based methods with a pure data-based model. However, the overall impression is that, the application of models with numerous parameters seems to be inseparably connected with the problem of the equifinality. A similar behavior is known e.g. for the Shiono-Knight model by Knight et al. (2007). For vegetation-roughness models, it will apply not only in the cases, where parameters are identified purely in terms of the inverse task, but also when available measurements of vegetation properties are uncertain and have to be generalized over larger areas (Straatsma and Huthoff, 2011). In such cases it will always be necessary to find values characterizing rather hydraulic conditions than true vegetation features. The difference is that even with a very uncertain data, the identification problem will be limited to relatively narrow parameter ranges.

The parameter identification is expected to result in more physically realistic values if at least some of the required vegetation properties or the channel bed roughness can be directly measured and used as the input. For instance, the vegetation extents of the two-layer models (Figure 2) are straightforward to obtain at the field, or vegetation can be assumed to cover all channel perimeter above the bankfull level. Typical heights of grassy floodplain vegetation in a given geographical area can be obtained through remote sensing coupled with information on channel geometry, and these values may be extrapolated to other sites where such information is not available.

Process-based models introduce however physical constrains, providing, as mentioned before, better basis for extrapolation, than purely data driven approaches and in this study better than a simpler model. In most of analyzed here cases, vegetation roughness models, when applied for vegetation conditions they were originally developed for, provided better predictions of higher flow than the Manning based DCM (Figure 15). Also some advantages of using the process-based models, even without knowledge on parameters, might be their clear physical interpretation, comparing for example with Manning coefficients.

Nonphysical stem diameters are more obvious to large values of the Manning coefficient. A modeler aware of parameter interactions can decide, if e.g. given discrepancies in vegetation characteristics are important in an analyzed case.

Discharge formulas analyzed in the study are usually only a part of the one-dimensional model. The uncertainty of such models depends also on additional elements, like spatial variability of resistance and simplification of the channel geometry. It should be also noted, that the investigated cases had a fairly regular cross-section and homogeneous vegetation. Therefore, care should be taken when attempting to generalize the presented findings to all one-dimensional approaches. In complex real-world cases, it might be beneficial to include several discharge formulas through an ensemble approach, which is also used in other fields, such as climate modeling.

## 5 Conclusions

This study investigated the application of advanced process-based methods for the discharge capacity estimation of vegetated compound channels in practical cases with limited information on the vegetation properties. We compared five process-based methods with a physically-based vegetation characterization to the conventional Manning-based divided channel method (DCM), focusing on their uncertainty. The developed probabilistic approach and the used data covering a range of conditions on floodplain vegetation submergence, density, flexibility, and flow hydraulics, allowed to draw the following conclusions:

1. The calculations showed that it is possible to identify process-based models with a large number of parameters on the basis of the inverse problem with narrower or similar uncertainty bands compared to the Manning-based DCM.

2. The uncertainty related to the ill-posed inverse problem, resulting from the insufficient number of observations, is in the most cases noticeable only when a small number ($< 3 - 4$) of observations is used in the model identification. However, in the cases where the shape of the rating curve is more sensitive to model parameters, the results suggest that methods with more parameters have wider uncertainty bands when identified with small number of observations.

3. The model identification resulted in some parameters differing from their measured physical values, raising doubts on the physical interpretation of obtained models.

4. Despite unrealistic values of parameters, the process-based models for vegetation roughness revealed good extrapolation capabilities to high floodplain flows, when identified using only low floodplain flows.

5. Uncertainty estimates clearly indicate the applicability of a given model to the analyzed case. Unsuitable models, e.g. those developed for non-submerged vegetation but applied to submerged vegetation, have relatively wide uncertainty estimates or lack a probabilistic solution. Therefore, the results showed that it is possible to choose an appropriate model, without a prior knowledge of vegetation properties in the channel, by comparing obtained uncertainty widths.

6. The best results in terms of the lowest uncertainty estimates were obtained with the Mertens method for the emergent, rigid vegetation case. For the dense flexible vegetation, the simplified two-layer method (STLM) neglecting the flow in the vegetation layer, had the most reliable performance across different seasons, functioning under submerged and

emergent conditions. The generalized two-layer model (GTLM), of the process-based approaches, amended with a vegetation parameterization describing the flexibility and reconfiguration of the plants was the most universally applicable to different vegetative conditions.

7. In most cases, the Manning-based DCM had also satisfactory performance, but results suggests it had poorer capabilities for extrapolation to high floodplain flows when calibrated with only low floodplain flows, in comparison to process-based models.

8. An open issue is the generalizability of the obtained results to spatially distributed one-dimensional models.

9. The proposed approach with the novelty of comparing different models in terms of their uncertainty along with the quality of the uncertainty estimation might be useful in other similar studies.

*Author contributions.*  Adam Kiczko: manuscript text, implementation of discharge formulas in Matlab, numerical experiments; Kaisa Västilä: manuscript text, methodology for two layer models, Ritobacken field experiment, analysis of numerical outputs; Adam Kozioł, flume experiments and together with Janusz Kubrak: Pasche and Martens methodology including computation algorithm; Elżbieta Kubrak and Marcin Krukowski: flume experiments, measurement data analysis, improving article text.

*Competing interests.*  No competing interests are present.

*Acknowledgements.*  The research was partly supported by National Science Centre (Poland), Program Miniatura 1, project no. 2017/01/X/ST10/00987, Maa- ja vesitekniikan tuki ry (Grant No 33271), Maj and Tor Nessling Foundation (Grant No 201800045), and by the National Centre for Research and Development (Grant No 347837/11/NCBR/2017, "Technical innovations and system of monitoring, forecasting and planning of irrigation and drainage for precise water management on the scale of drainage/irrigation system"). We acknowledge Academy of Finland (Grant No 133113), Maa- ja vesitekniikan tuki ry and the Finnish Ministry of Agriculture and Forestry for funding the collection of the original field data.

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

 # Appendix A: Measurement data used in computations

## A1 Flume experiments

**Table A1.** Measured water depth $H$ and flow rate $Q$ for quasi-uniform flow conditions in flume experiments with constant slope $s = 5 \cdot 10^{-4}$ (Koziol, 2010; Kozioł, 2013; Kozioł and Kubrak, 2015)

| No. | Case 1 | | Case 2 | |
|---|---|---|---|---|
| | H (m) | Q (m$^3$/s) | H (m) | Q (m$^3$/s) |
| 1 | 0.170 | 0.018 | 0.209 | 0.039 |
| 2 | 0.177 | 0.019 | 0.212 | 0.039 |
| 3 | 0.183 | 0.021 | 0.225 | 0.042 |
| 4 | 0.195 | 0.023 | 0.238 | 0.045 |
| 5 | 0.211 | 0.026 | 0.244 | 0.048 |
| 6 | 0.225 | 0.030 | 0.255 | 0.050 |
| 7 | 0.243 | 0.035 | 0.262 | 0.053 |
| 8 | 0.270 | 0.041 | 0.274 | 0.056 |
| 9 | 0.289 | 0.046 | 0.282 | 0.058 |
| 10 | | | 0.284 | 0.059 |

## A2 Ritobacken field experiment

**Table A2.** Cross-section for the Ritobacken brook. Original data collected by Västilä et al. (2016)

| Station (m) | 0.20 | 0.35 | 0.40 | 0.60 | 0.80 | 1.20 | 2.00 | 2.20 | 2.40 | 3.40 | 5.00 | 6.40 | 6.60 | 7.00 |
|---|---|---|---|---|---|---|---|---|---|---|---|---|---|---|
| Elevation (m) | 1.08 | 1.07 | 1.15 | 1.12 | 1.07 | 0.93 | 0.61 | 0.54 | 0.50 | 0.48 | 0.49 | 0.47 | 0.45 | 0.33 |

| Station (m) | 7.20 | 7.40 | 7.60 | 7.80 | 8.00 | 8.40 | 8.60 | 8.80 | 9.00 | 9.60 | 9.80 | 10.00 | 10.20 |
|---|---|---|---|---|---|---|---|---|---|---|---|---|---|
| Elevation (m) | 0.20 | 0.10 | 0.07 | 0.01 | 0.00 | 0.04 | 0.20 | 0.41 | 0.53 | 0.78 | 0.82 | 0.90 | 0.94 |

Obtained form field surveys 2010-2012 for 190 m river reach and averaged to obtain a single cross-section; number of measurement points were reduced using the algorithm of Recursive Douglas-Peucker Polyline Simplification (Schwanghart, 2010), with the tolerance of 0.01 m

**Table A3.** Data for the Ritobacken case study, used in calculations: water depth $H$, flow rate $Q$, energy grade slope $S$, inundated vegetation height $h_{v,inud.}$ and Blockage factor $B_X$. Water depths $H$ were obtained by averaging upstream and downstream depths. Original data collected by Västilä et al. (2016)

| Case | No. | H (m) | Q (m³/s) | S (−) | $h_{v,inud.}$ (m) | $B_X$ (−) |
|------|-----|-------|----------|-------|-------------------|-----------|
| Spring 2011 | 1 | 0.611 | 0.349 | $9.0 \cdot 10^{-4}$ | 0.073 | 0.189 |
| | 2 | 0.647 | 0.440 | $8.0 \cdot 10^{-4}$ | 0.081 | 0.197 |
| | 3 | 0.694 | 0.565 | $7.0 \cdot 10^{-4}$ | 0.086 | 0.185 |
| | 4 | 0.738 | 0.709 | $7.0 \cdot 10^{-4}$ | 0.086 | 0.166 |
| | 5 | 0.785 | 0.844 | $6.0 \cdot 10^{-4}$ | 0.086 | 0.148 |
| | 6 | 0.841 | 1.022 | $6.0 \cdot 10^{-4}$ | 0.086 | 0.130 |
| Autumn 2011 | 1 | 0.583 | 0.184 | $1.6 \cdot 10^{-3}$ | 0.147 | 0.369 |
| | 2 | 0.640 | 0.244 | $1.6 \cdot 10^{-3}$ | 0.204 | 0.433 |
| | 3 | 0.698 | 0.316 | $1.7 \cdot 10^{-3}$ | 0.257 | 0.472 |
| | 4 | 0.731 | 0.366 | $1.7 \cdot 10^{-3}$ | 0.288 | 0.487 |
| | 5 | 0.776 | 0.459 | $1.8 \cdot 10^{-3}$ | 0.326 | 0.500 |
| | 6 | 0.838 | 0.565 | $1.7 \cdot 10^{-3}$ | 0.374 | 0.527 |
| | 7 | 0.894 | 0.684 | $1.6 \cdot 10^{-3}$ | 0.414 | 0.504 |
| | 8 | 0.928 | 0.788 | $1.7 \cdot 10^{-3}$ | 0.438 | 0.504 |
| | 9 | 0.968 | 0.901 | $1.7 \cdot 10^{-3}$ | 0.467 | 0.502 |
| | 10 | 1.021 | 1.053 | $1.7 \cdot 10^{-3}$ | 0.505 | 0.500 |
| | 11 | 1.071 | 1.218 | $1.7 \cdot 10^{-3}$ | 0.535 | 0.478 |
| | 12 | 1.114 | 1.396 | $1.7 \cdot 10^{-3}$ | 0.552 | 0.476 |
| Spring 2012 | 1 | 0.556 | 0.257 | $1.5 \cdot 10^{-3}$ | 0.096 | 0.271 |
| | 2 | 0.606 | 0.333 | $1.5 \cdot 10^{-3}$ | 0.135 | 0.332 |
| | 3 | 0.629 | 0.402 | $1.5 \cdot 10^{-3}$ | 0.153 | 0.351 |
| | 4 | 0.700 | 0.521 | $1.4 \cdot 10^{-3}$ | 0.201 | 0.379 |
| | 5 | 0.743 | 0.635 | $1.4 \cdot 10^{-3}$ | 0.218 | 0.375 |
| | 6 | 0.796 | 0.735 | $1.2 \cdot 10^{-3}$ | 0.233 | 0.362 |
| | 7 | 0.834 | 0.872 | $1.3 \cdot 10^{-3}$ | 0.236 | 0.342 |
| | 8 | 0.891 | 1.053 | $1.3 \cdot 10^{-3}$ | 0.236 | 0.311 |
| | 9 | 0.944 | 1.218 | $1.3 \cdot 10^{-3}$ | 0.236 | 0.285 |
| | 10 | 0.997 | 1.396 | $1.4 \cdot 10^{-3}$ | 0.236 | 0.264 |
| | 11 | 1.047 | 1.587 | $1.4 \cdot 10^{-3}$ | 0.236 | 0.246 |

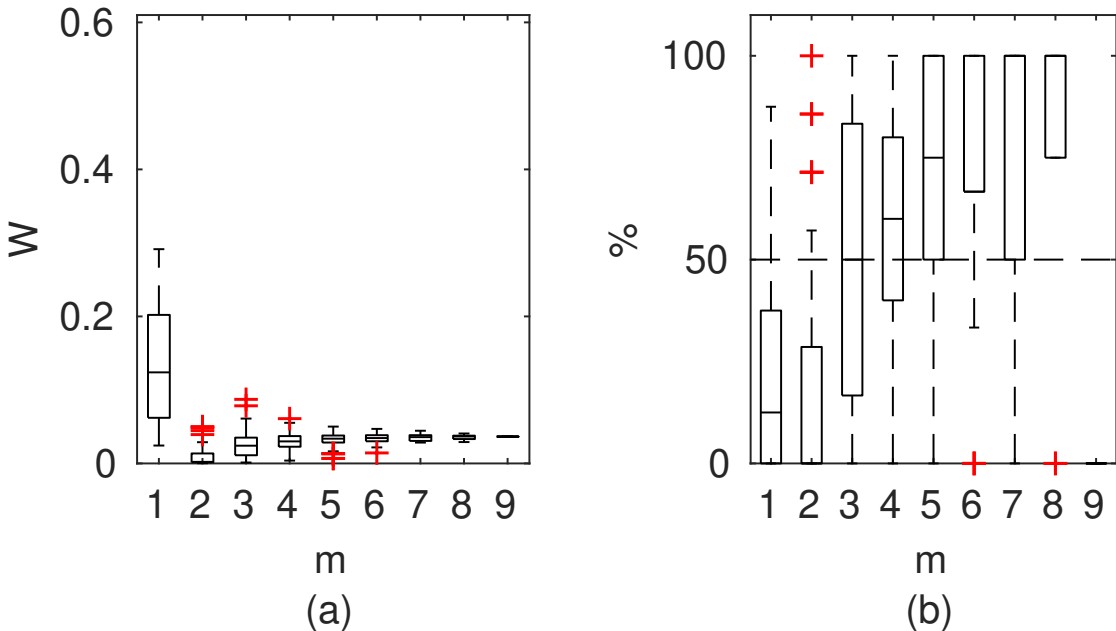

**Figure B1.** DCM Manning results for the flume case 1, (a) Averaged relative confidence widths $W$ as a function of observation set size $m$ used for model identification; (b) Percentage of verification points enclosed by the confidence intervals (100% denotes all points within intervals, box spans over 25% and 75% quantile, median is given with horizontal line, whiskers indicate the result extent, cross marks are for extreme values)

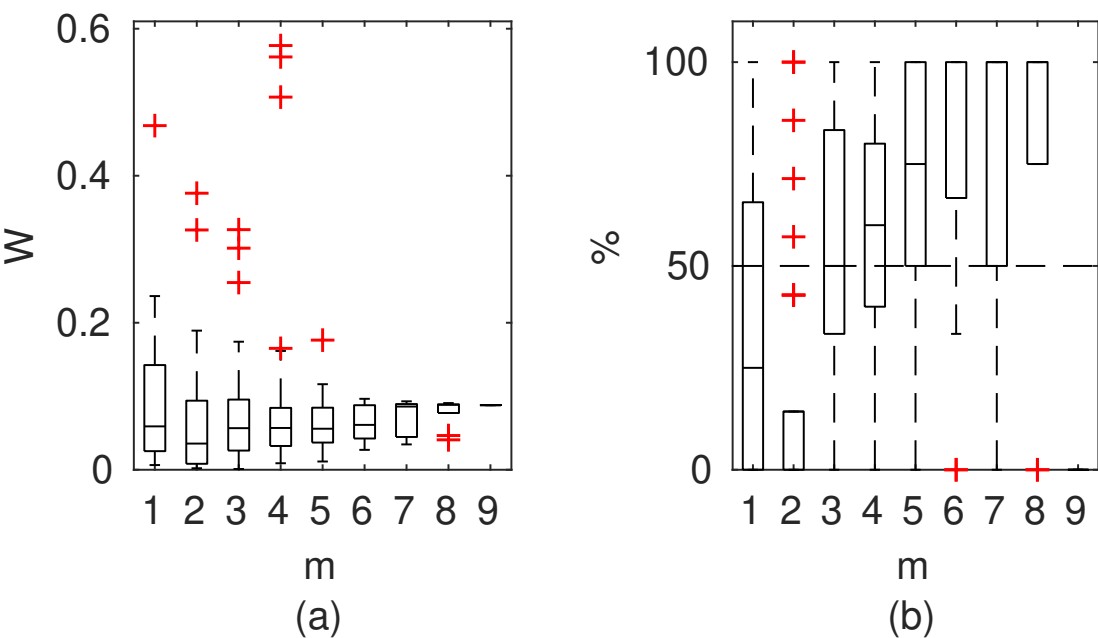

**Figure B2.** Pasche results for the flume case 1, (a) Averaged relative confidence widths $W$ as a function of observation set size $m$ used for model identification; (b) Percentage of verification points enclosed by the confidence intervals ($100\%$ denotes all points within intervals, box spans over 25% and 75% quantile, median is given with horizontal line, whiskers indicate the result extent, cross marks are for extreme values)

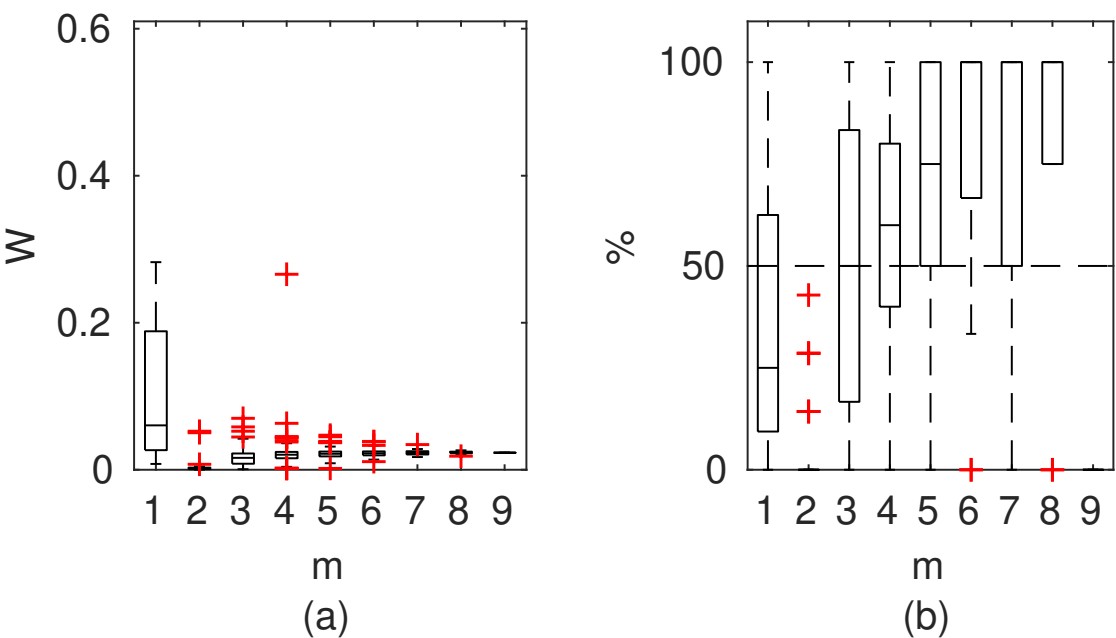

**Figure B3.** Mertens results for the flume case 1, (a) Averaged relative confidence widths $W$ as a function of observation set size $m$ used for model identification; (b) Percentage of verification points enclosed by the confidence intervals (100% denotes all points within intervals, box spans over 25% and 75% quantile, median is given with horizontal line, whiskers indicate the result extent, cross marks are for extreme values)

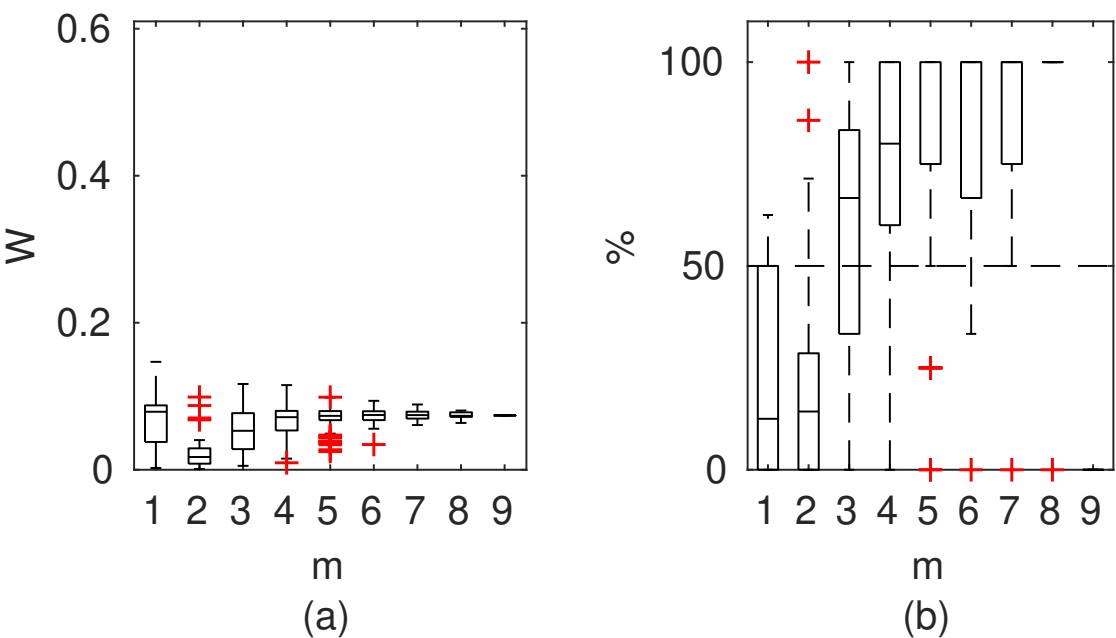

**Figure B4.** GTLM results for the flume case 1, (a) Averaged relative confidence widths $W$ as a function of observation set size $m$ used for model identification; (b) Percentage of verification points enclosed by the confidence intervals (100% denotes all points within intervals, box spans over 25% and 75% quantile, median is given with horizontal line, whiskers indicate the result extent, cross marks are for extreme values)

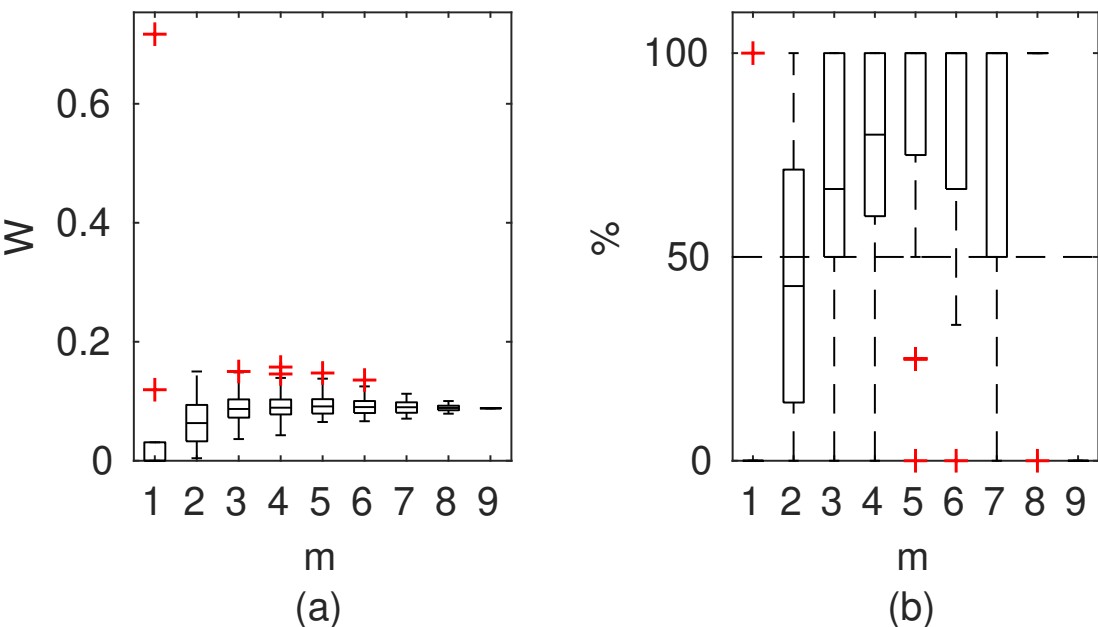

**Figure B5.** PTLM results for the flume case 1, (a) Averaged relative confidence widths $W$ as a function of observation set size $m$ used for model identification; (b) Percentage of verification points enclosed by the confidence intervals ($100\%$ denotes all points within intervals, box spans over $25\%$ and $75\%$ quantile, median is given with horizontal line, whiskers indicate the result extent, cross marks are for extreme values)

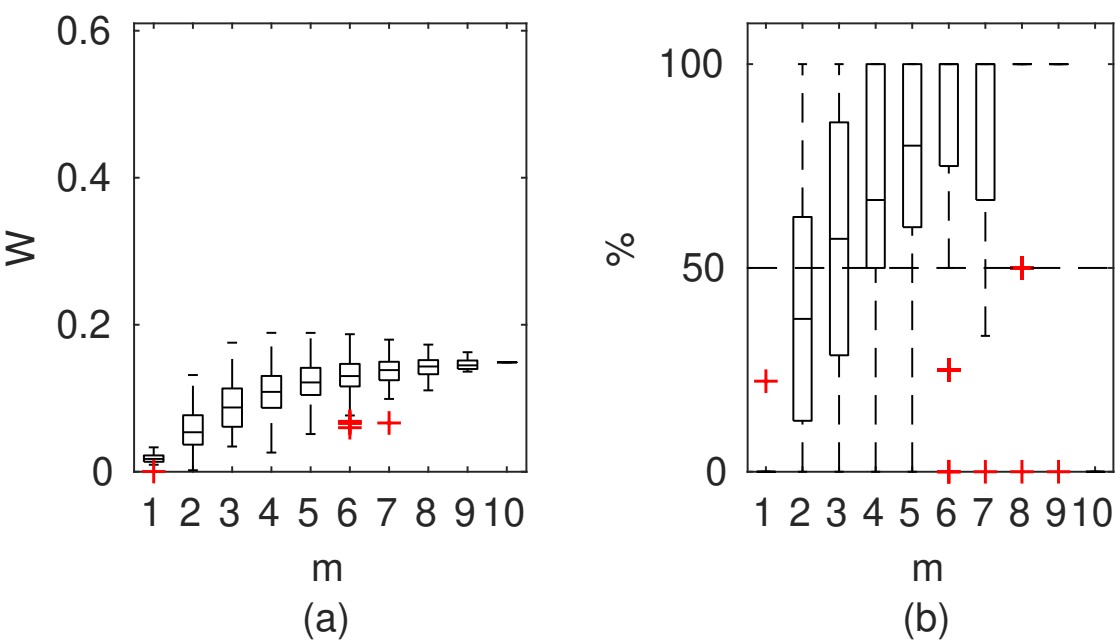

**Figure B6.** DCM Manning results for the flume case 1, (a) Averaged relative confidence widths $W$ as a function of observation set size $m$ used for model identification; (b) Percentage of verification points enclosed by the confidence intervals (100% denotes all points within intervals, box spans over 25% and 75% quantile, median is given with horizontal line, whiskers indicate the result extent, cross marks are for extreme values)

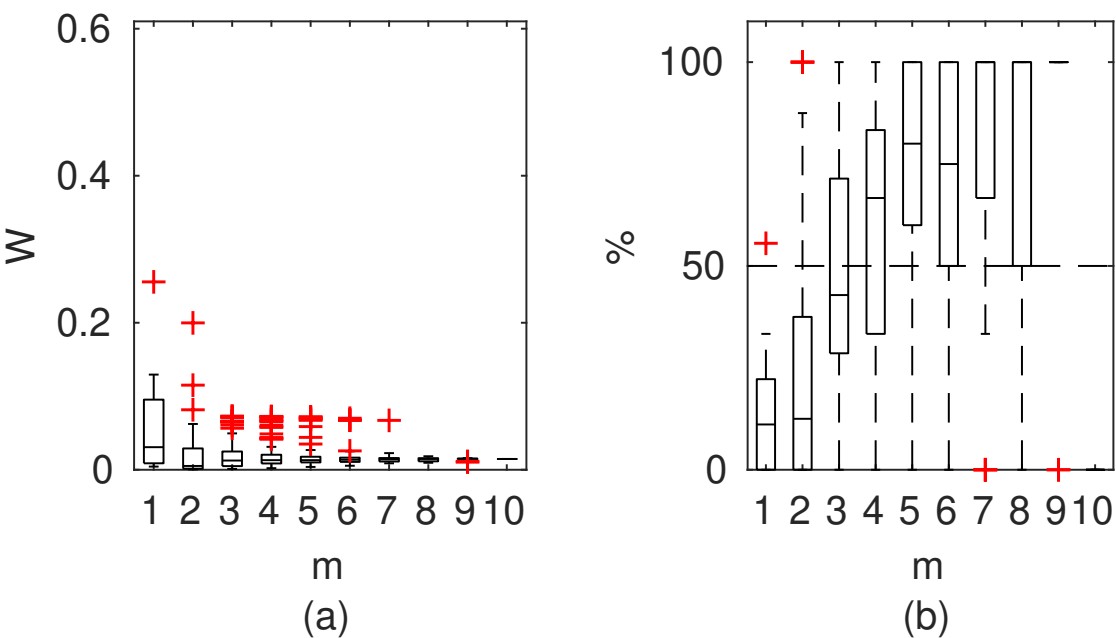

**Figure B7.** Pasche results for the flume case 1, (a) Averaged relative confidence widths $W$ as a function of observation set size $m$ used for model identification; (b) Percentage of verification points enclosed by the confidence intervals ($100\%$ denotes all points within intervals, box spans over 25% and 75% quantile, median is given with horizontal line, whiskers indicate the result extent, cross marks are for extreme values)

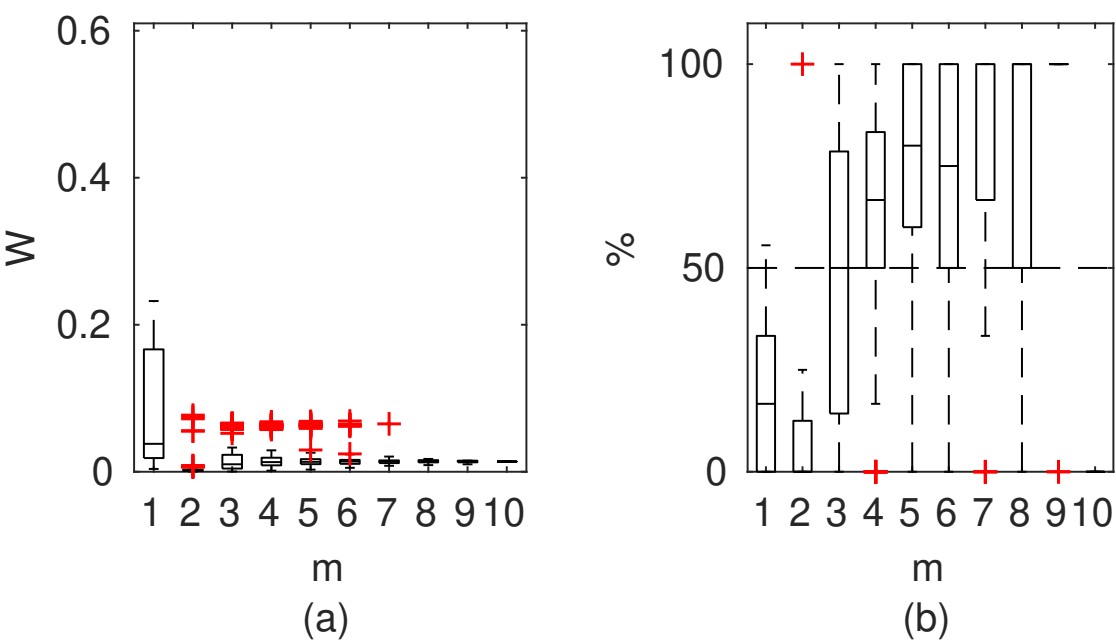

**Figure B8.** Mertens results for the flume case 1, (a) Averaged relative confidence widths $W$ as a function of observation set size $m$ used for model identification; (b) Percentage of verification points enclosed by the confidence intervals ($100\%$ denotes all points within intervals, box spans over 25% and 75% quantile, median is given with horizontal line, whiskers indicate the result extent, cross marks are for extreme values)

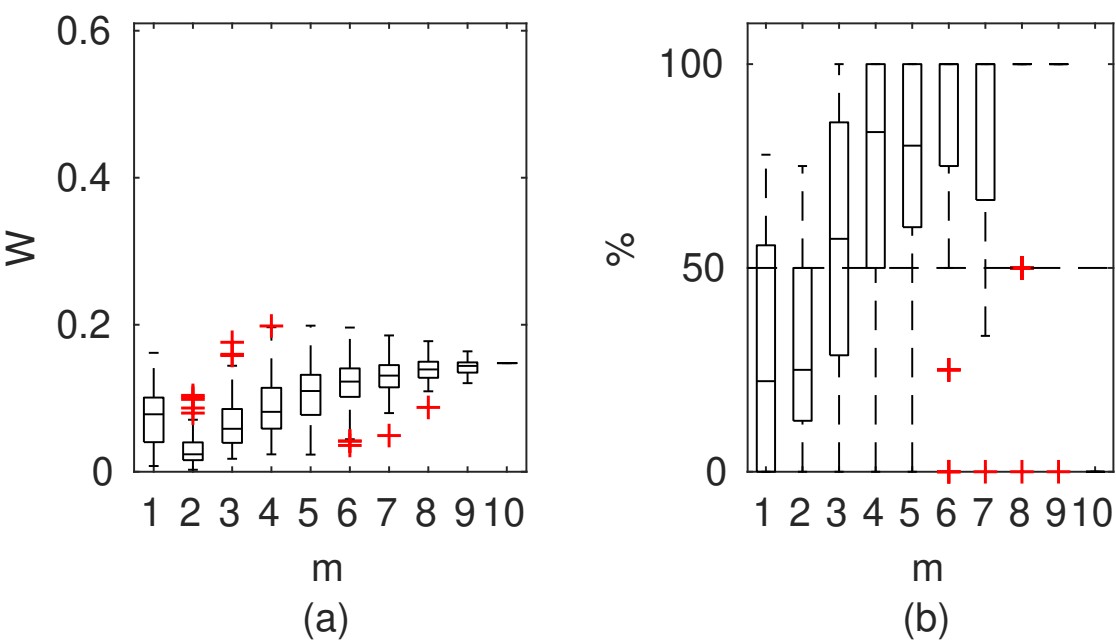

**Figure B9.** GTLM results for the flume case 1, (a) Averaged relative confidence widths $W$ as a function of observation set size $m$ used for model identification; (b) Percentage of verification points enclosed by the confidence intervals ($100\%$ denotes all points within intervals, box spans over 25% and 75% quantile, median is given with horizontal line, whiskers indicate the result extent, cross marks are for extreme values)

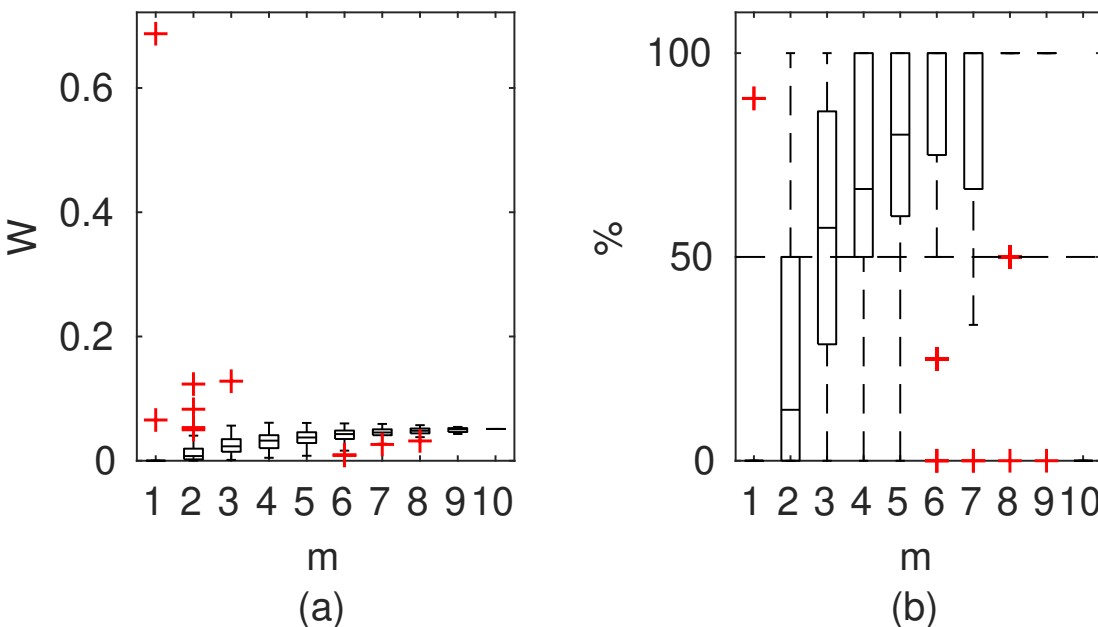

**Figure B10.** PTLM results for the flume case 1, (a) Averaged relative confidence widths $W$ as a function of observation set size $m$ used for model identification; (b) Percentage of verification points enclosed by the confidence intervals (100% denotes all points within intervals, box spans over 25% and 75% quantile, median is given with horizontal line, whiskers indicate the result extent, cross marks are for extreme values)

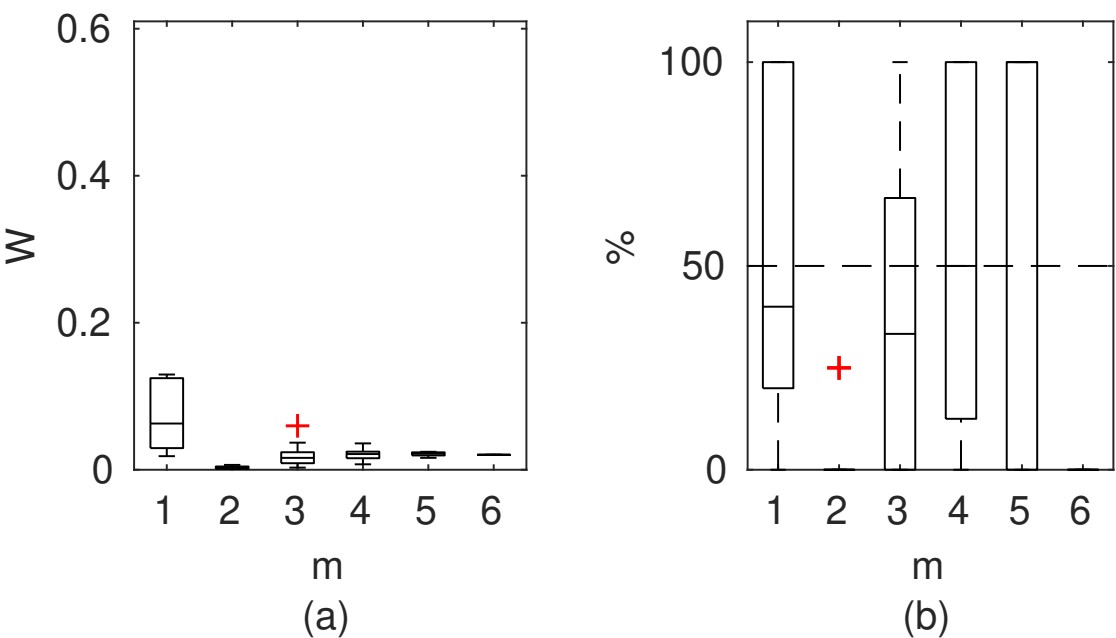

**Figure B11.** Manning DCM results for Ritobacken case study, Spring 2011, (a) Averaged relative confidence widths $W$ as a function of observation set size $m$ used for model identification; (b) Percentage of verification points enclosed by the confidence intervals (100% denotes all points within intervals, box spans over 25% and 75% quantile, median is given with horizontal line, whiskers indicate the result extent, cross marks are for extreme values)

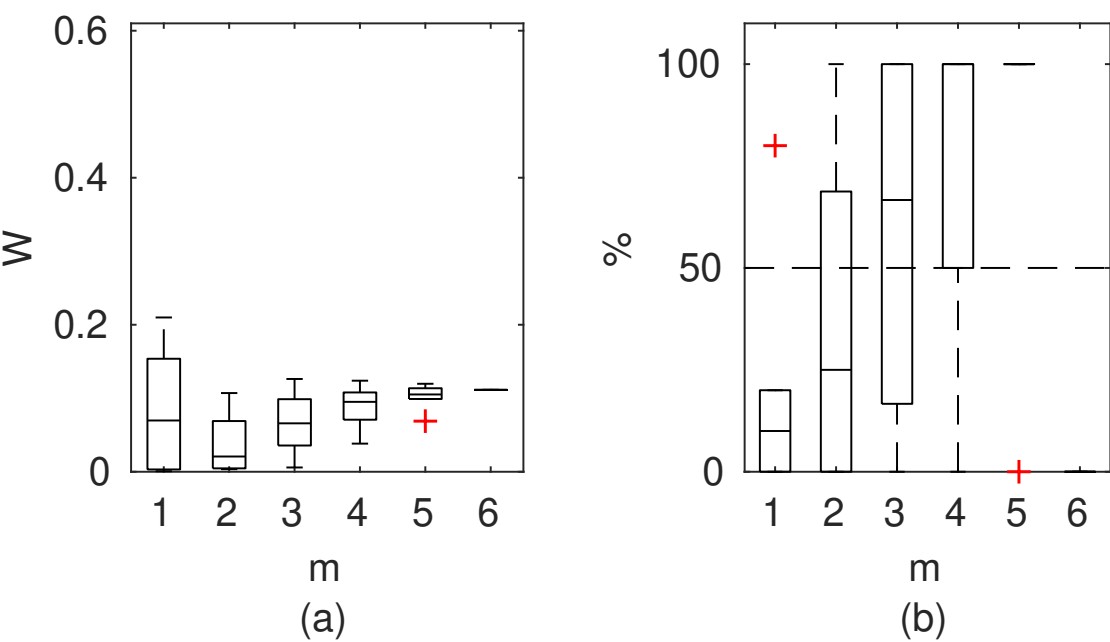

**Figure B12.** GTLM results for Ritobacken case study, Spring 2011, (a) Averaged relative confidence widths $W$ as a function of observation set size $m$ used for model identification; (b) Percentage of verification points enclosed by the confidence intervals (100% denotes all points within intervals, box spans over 25% and 75% quantile, median is given with horizontal line, whiskers indicate the result extent, cross marks are for extreme values)

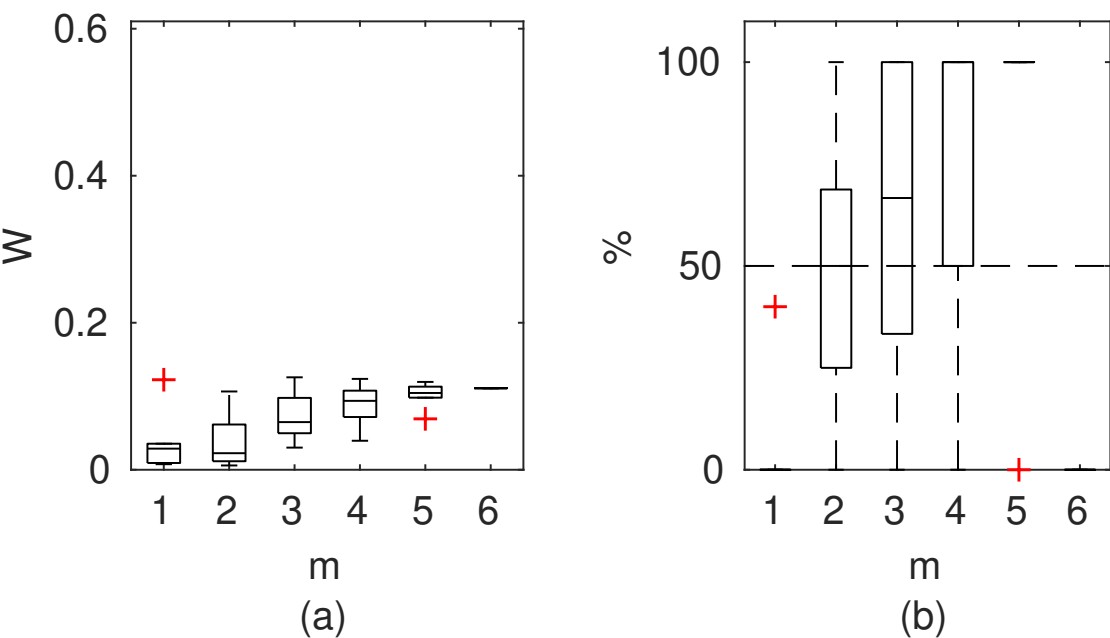

**Figure B13.** STLM results for Ritobacken case study, Spring 2011, (a) Averaged relative confidence widths $W$ as a function of observation set size $m$ used for model identification; (b) Percentage of verification points enclosed by the confidence intervals (100% denotes all points within intervals, box spans over 25% and 75% quantile, median is given with horizontal line, whiskers indicate the result extent, cross marks are for extreme values)

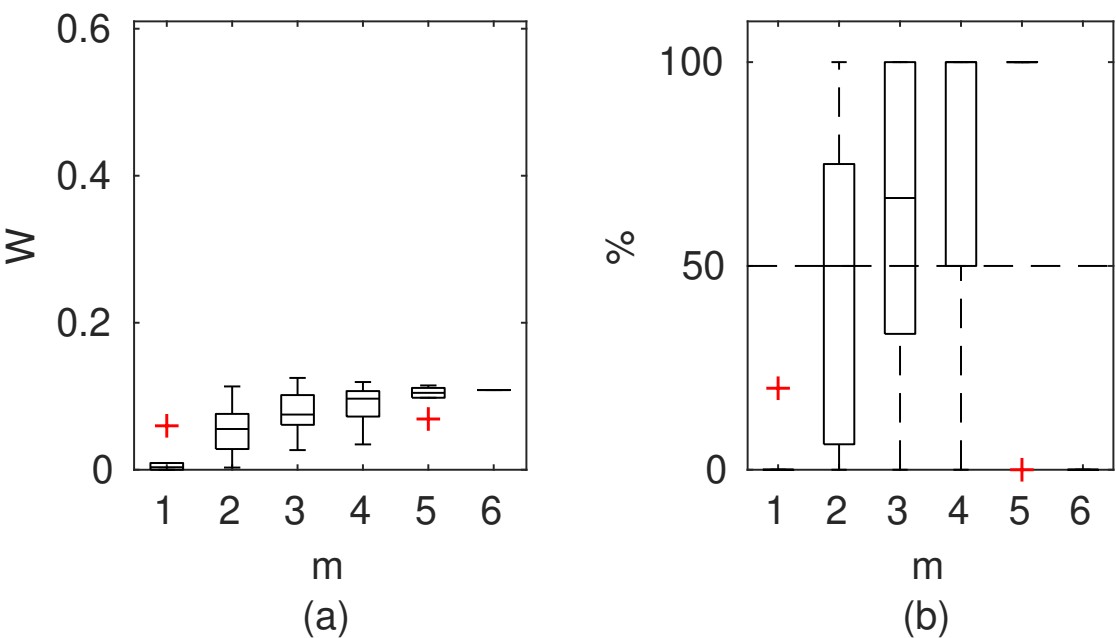

**Figure B14.** PTLM results for Ritobacken case study, Spring 2011, (a) Averaged relative confidence widths $W$ as a function of observation set size $m$ used for model identification; (b) Percentage of verification points enclosed by the confidence intervals (100% denotes all points within intervals, box spans over 25% and 75% quantile, median is given with horizontal line, whiskers indicate the result extent, cross marks are for extreme values)

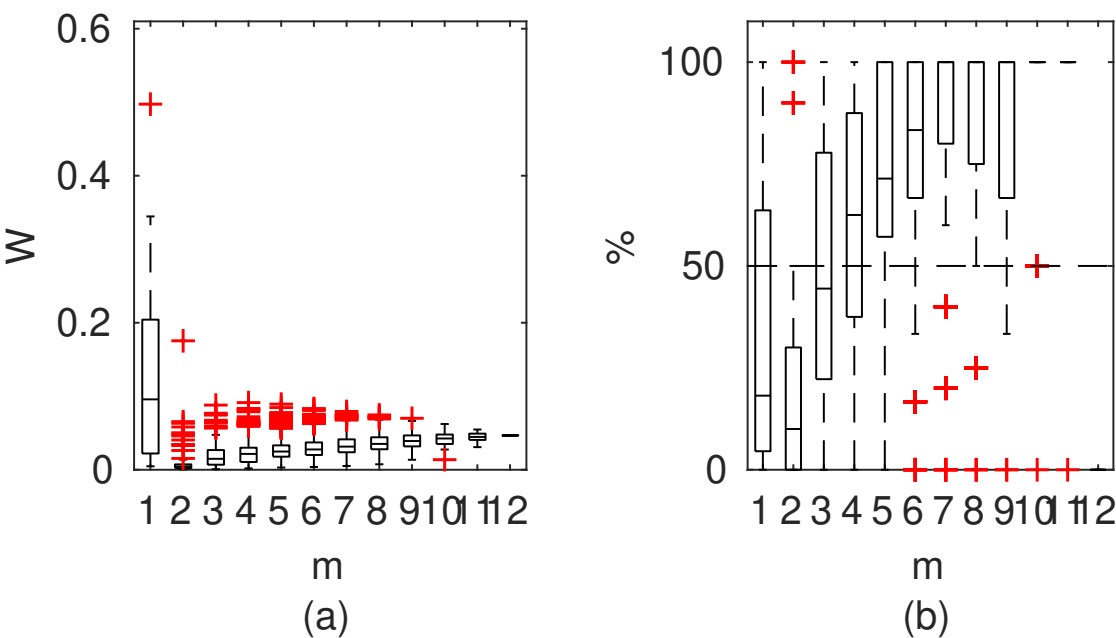

**Figure B15.** Manning DCM results for Ritobacken case study, Autumn 2011, (a) Averaged relative confidence widths $W$ as a function of observation set size $m$ used for model identification; (b) Percentage of verification points enclosed by the confidence intervals ($100\%$ denotes all points within intervals, box spans over $25\%$ and $75\%$ quantile, median is given with horizontal line, whiskers indicate the result extent, cross marks are for extreme values)

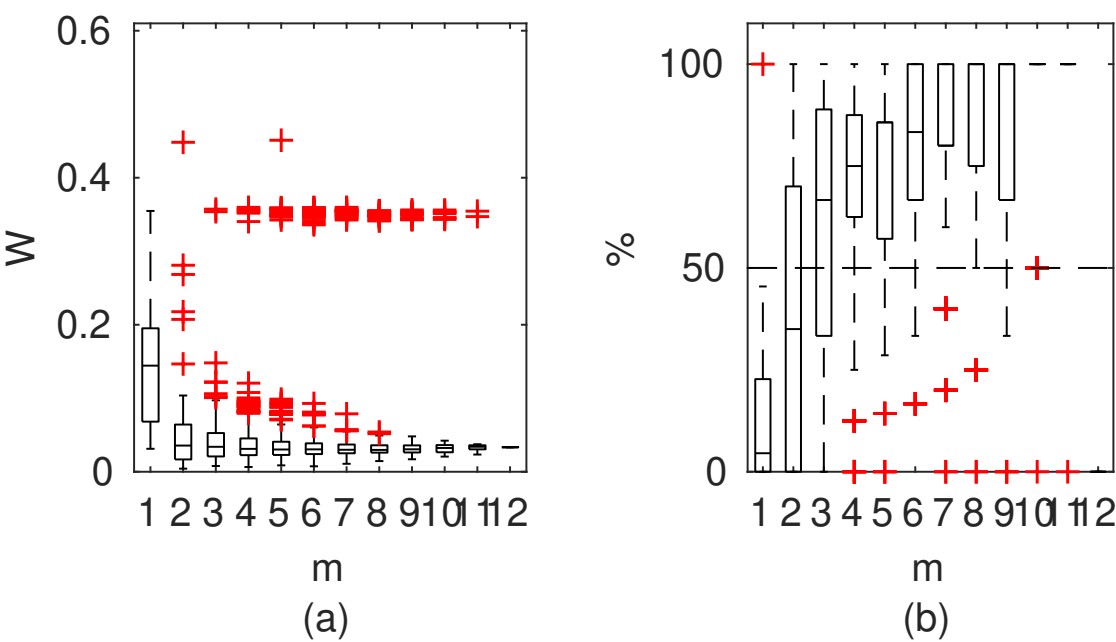

**Figure B16.** GTLM results for Ritobacken case study, Autumn 2011, (a) Averaged relative confidence widths $W$ as a function of observation set size $m$ used for model identification; (b) Percentage of verification points enclosed by the confidence intervals (100% denotes all points within intervals, box spans over 25% and 75% quantile, median is given with horizontal line, whiskers indicate the result extent, cross marks are for extreme values)

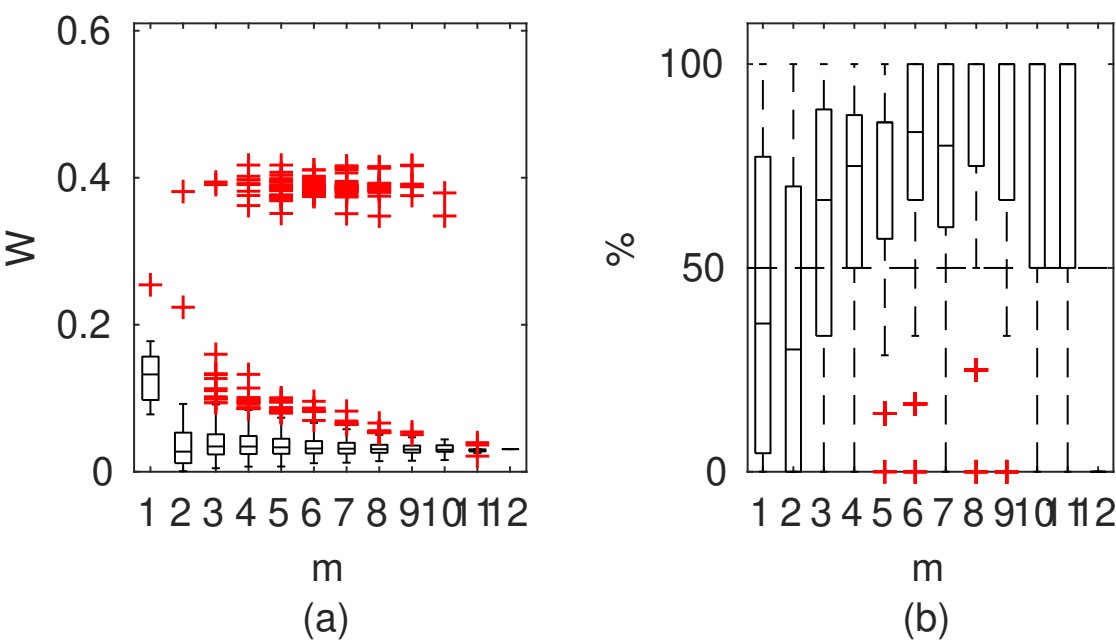

**Figure B17.** STLM results for Ritobacken case study, Autumn 2011, (a) Averaged relative confidence widths $W$ as a function of observation set size $m$ used for model identification; (b) Percentage of verification points enclosed by the confidence intervals (100% denotes all points within intervals, box spans over 25% and 75% quantile, median is given with horizontal line, whiskers indicate the result extent, cross marks are for extreme values)

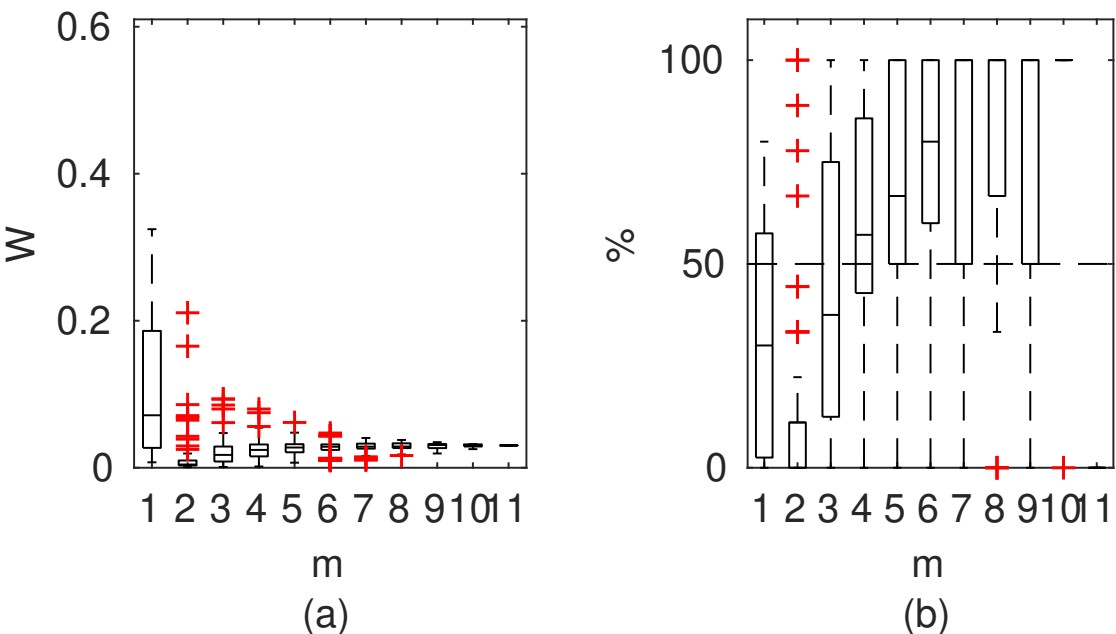

**Figure B18.** Manning DCM results for Ritobacken case study, Spring 2012, (a) Averaged relative confidence widths $W$ as a function of observation set size $m$ used for model identification; (b) Percentage of verification points enclosed by the confidence intervals (100% denotes all points within intervals, box spans over 25% and 75% quantile, median is given with horizontal line, whiskers indicate the result extent, cross marks are for extreme values)

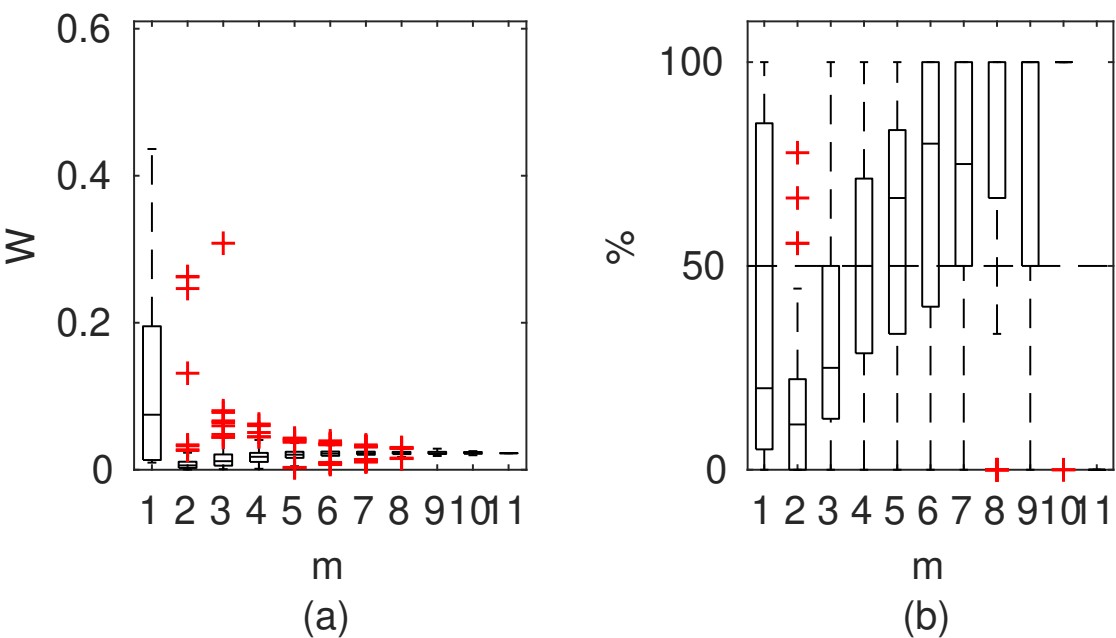

**Figure B19.** GTLM results for Ritobacken case study, Spring 2012, (a) Averaged relative confidence widths $W$ as a function of observation set size $m$ used for model identification; (b) Percentage of verification points enclosed by the confidence intervals (100% denotes all points within intervals, box spans over 25% and 75% quantile, median is given with horizontal line, whiskers indicate the result extent, cross marks are for extreme values)

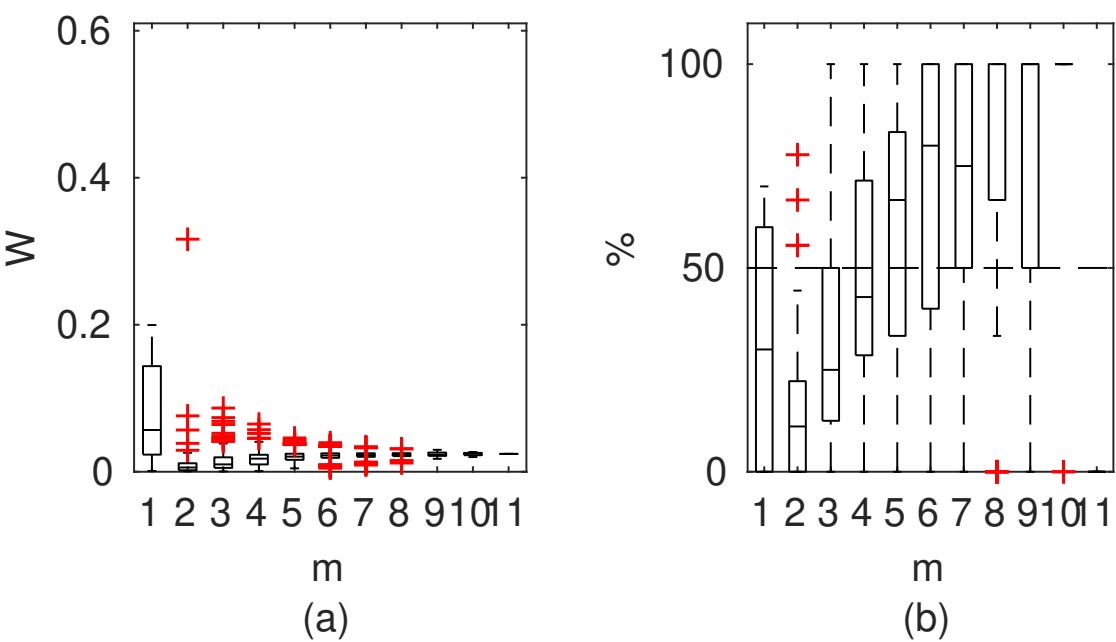

**Figure B20.** STLM results for Ritobacken case study, Spring 2012, (a) Averaged relative confidence widths $W$ as a function of observation set size $m$ used for model identification; (b) Percentage of verification points enclosed by the confidence intervals (100% denotes all points within intervals, box spans over 25% and 75% quantile, median is given with horizontal line, whiskers indicate the result extent, cross marks are for extreme values)

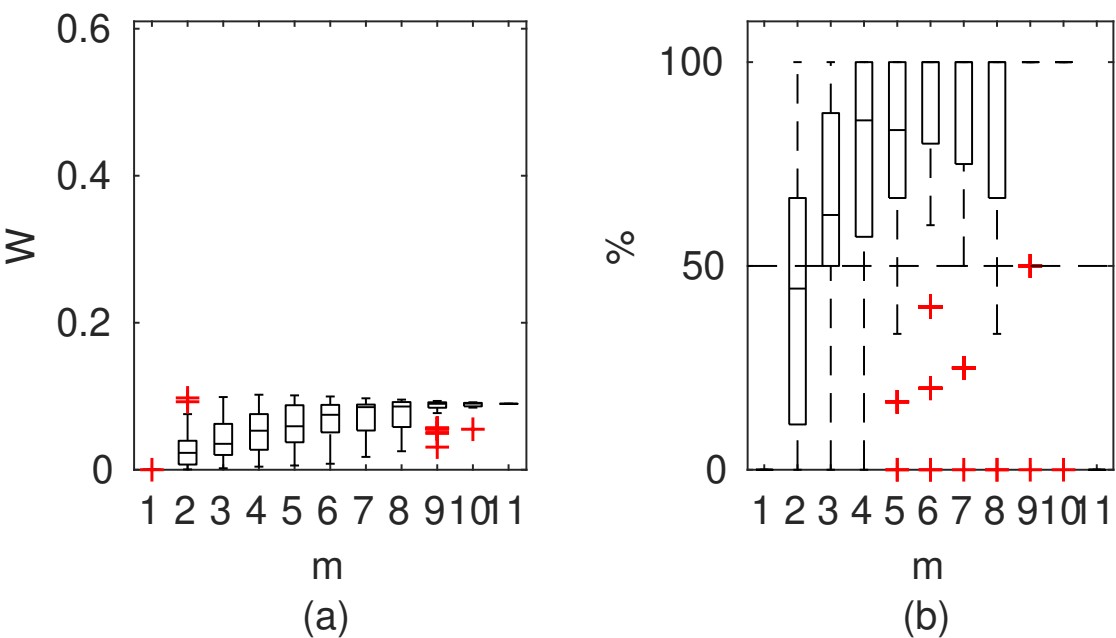

**Figure B21.** PTLM results for Ritobacken case study, Spring 2012, (a) Averaged relative confidence widths $W$ as a function of observation set size $m$ used for model identification; (b) Percentage of verification points enclosed by the confidence intervals (100% denotes all points within intervals, box spans over 25% and 75% quantile, median is given with horizontal line, whiskers indicate the result extent, cross marks are for extreme values)