# Peer review of "Predicting discharge capacity of vegetated compound channels: uncertainty and identifiability of 1D process-based models"

_Hydrology and Earth System Sciences, 2019_

## Referee Comment (RC1) · Anonymous Referee #1 · 7 Feb 2020

The paper addresses an important and highly relevant topic and is of great importance for the professional community to improve the capabilities to accurately model the complex flow in vegetated compound channels. However, I identified various issues that should be addressed and clarified by the authors. In general, the paper has been written with great care, but there are many passages that should be improved stylistically and where the language should be improved (see my detailed comments below; I recommend that the language is cross-checked by a native speaker). I also found that the terminology should be better introduced and defined to help the reader to better understand the complex content of the manuscript. For example, it became not really clear to me what is meant by, for example, identification data points, verification data

points, computation points, observation points etc. Overall, the manuscript focuses on a statistical analysis of different approaches regarding uncertainties associated with input parameters. This is highly relevant, but the manuscript is mostly written from a statistical (probabilistic) point of view. However, in practice the tested approaches are typically used by hydraulic or environmental engineers, and it would be nice to outline the chosen approach more generally at the beginning of the manuscript, so that the significance of the results becomes clearer for the target-audience. In this context, I am not an expert in statistics and this fact triggered many questions (see below). In my opinion, this manuscript could make a real impact if it would be written in a way that practitioners will better understand what has been done. Also, more specific statements related to the parameter variation would be helpful. I acknowledge that this is partly addressed in the discussion, but it becomes not really obvious from the preceding sections. To summarize, this is an interesting manuscript. However, the presentation of the material should be improved. As I have many specific comments, I am afraid that I have no other choice than to recommend returning the manuscript to the authors for major revisions. I hope the authors will find my comments useful.

Specific comments

Please note that the number of comments decreases towards the end of the manuscript - this is due to the fact that various issues have already been highlighted at the beginning.

Title: I am not sure that I understand what is meant by "identifiability" – I guess a "statistical" meaning is meant, but that becomes not clear. P2, L22: What is meant by stability? P2, L25: I find this a bit confusing, as there is some redundancy with the sentence before. This could be formulated better. P2, L27: What exactly is meant by "regions"? I assume the different channel parts are meant (but one could also think about different geographical regions). P2, L33: Is this approach really simple? I would delete the latter word. P2, L36: Please check language. P2, L41: Which relationship is meant here? P2, L42: Check language P2, L43: Check language P2, L44: Why

"should"? P2, L47: The approach was published in 1991, and the only reference for its successful use is from 2020? I doubt that it took 30 years that it was successfully used... P2, L48: Please check language P2, L49: Why referring specifically to Pasche, and not to Pasche & Rouvé? P2, L53: Please check language P2, L55: Check how reference is included into the text. P3, L57: I doubt that all of the cited approaches have been developed to parametrize the two-layer approach. Some of them deal more with the parameterization of vegetation properties. P3, L58: Please check the sentence ("... for in...."). P3, L60: Please improve "Methods like Pasche and Rouvé (1985)..." - Pasche and Rouvé are the names of the authors who developed the method (a similar comment can be given regarding the reference to Västilä and Järvelä in the same line). P3, L62: Is this really true? Remote sensing methods have significantly developed, and dependent on regions and countries, such information may be available... P3, L65: In my opinion, this is not an argument for simpler methods. Such a non-physical based black-box will not help to better understand the problems at hand and requires manual and arbitrary calibration. P3, L67: Why disused? What is meant by this? P3, L68: Fr typically defines the Froude number – but I guess here the word "For" is meant? I stop here giving particular comments on the language, as I already have provided many such comments showing the need to improve the paper. P3, L69: I do not understand this statement – "when bathymetric data do not account for the true complexity of the river geometry" – what is meant by "true complexity of the river geometry"? P3, L70: This example is not really related to the topic of the paper... P3, L72-75: This is difficult to understand – please improve. P3, L77: Which answer? In other words, what is the question? P3, L81: "...comparing to the Manning...." – this part of the sentence remains unclear to me. P3, L86: I do not understand what is meant by "parameterized in a sense of their distributions" P4, L93: Improve stylistically (...Järvelä Järvelä...) P4, L95: Which study? This one? P4, L99: "The overall goal of this paper is to compare the uncertainty, parameter identifiability and physical interpretation of the parameters of discharge capacity methods characterized with different levels of parameterization". This sentence is very difficult to understand. P5, Figure 1: Please improve the caption

and the description of the figure; I find it difficult to understand (note also that not all parameters have been defined) P5, L120 – 125: Please improve – this could be explained more clearly in my opinion. P9, L242: "...are plant species..."? I am not sure that I understand what is meant. P9, L247: This depends on the level of submergence – otherwise 20% of the discharge may be neglected... P9, L251: This depends – for the typically used rigid cylinder analogy, Bx will basically be constant. P9, L253: What exactly is l_l and l_r? In this context, a sketch would be helpful. P10, L264: The model of Luhar anf Nepf was already mentioned before - it may be a good idea to restructure the manuscript and to present this approach earlier? Also, this is not the "original formula" (which should be stated more clearly), as the hydraulic radius is used while Luhar and Nepf used the water depth. P10, L269: Check writing style (..."formula 9...", "...three parameter one...") etc). P10, L280: Remove the full stop after "experiments" P10, L290: Strictly speaking, uniform conditions are impossible to achieve by this setup - I would prefer if the terminology "quasi uniform" is used. The flume slope (or the slope range) should also be given - I could not find this information in Koziol (2010) as this paper seems to be in Polish language. P10, L291: How were water levels recorded and what was the spacing between the measurements? P11, L300: What kind of vegetation? P13, Ö320: What is constrain 5? Is equation 5 meant? P13, L328/330: Please use another notation for the number of observation points - before, n was used to define the Manning coefficient, and this is confusing (see also my comment below; L336). In this context, what exactly is meant by observation point? It could also be the number of points Also, it should be mentioned that only floodplain flows were investigated (for convenience of the reader). P13, L334: Not necessarily - in some (or many) cases there exist data for high flows that can be used for calibration. P13, L336: I am partly confused here, n is the number of observation points which could also be the number of readings taken for the water depth measurement. Please be more specific. P13, L346: Please improve cross-references - e.g., here it should be Figure 4a. Figure 4d-f: Why are the lowest two points characterized by the same discharge? P15, L349: I am not sure that I can follow - this could be explained better

(what is exactly is meant by computation points). P16, Table 2 (also Table 1): Could you comment on the used parameter bands in the text (why were these bands chosen?)? P16, L353: What exactly are identification data points? P16, L355: But the Pasche model results are not included in Fig. 5? This is confusing. P16, L356: (5a, 6a) - does this refer to the Figures? Please improve throughout the manuscript. P17, Figure 5: I am not sure that I understand what is meant by "Ratio". P17, L365: This definition could have been given earlier. P17, L369: Here, equation is used, at other places formula is used and at some places numbers are just given. Please improve - this is confusing. Also, the used model could be specified more precisely (the same applies to "other unspecified models"). P17., L370: Now n is the ensemble count - this is confusing. P18, L374: I am not a statistical expert (although I have some knowledge regarding statistics), but this is a bit confusing.... P18, L378: Why is 1 an extreme value? P18, L380: This fact could be explained in some more detail when outlining the approach. I am getting a bit lost here... P18, L383: Vegetation characteristics of Ritobacken have not been defined; what is meant by a "flexible approach"? P19, L396: I have trouble understanding this - a more general outline of the procedure would be helpful (this should be provided earlier, not here in the presentation of the results). P26, L416: I am not sure that I understand what is meant here. I stop giving more comments here on chapter 3 as I have problems to understand what exactly was done - the procedure could be outlined in some more detail P31, L492-498: It would be good to explain all his earlier in some more detail. P31, L508: Isn't this rather obvious? By the way, what about errors in the measurements - how would they affect this analysis? P33, L578: Numerical experiments are mentioned - but I doubt that detailed numerical simulations were carried out (no statements are given in the manuscript); this again shows the need to formulate statements more precisely.

---

## Referee Comment (RC2) · Anonymous Referee #2 · 22 Feb 2020

[referee-annotated manuscript omitted]

---

## Referee Comment (RC3) · Anonymous Referee #3 · 24 Feb 2020

**General comments**

The authors present a study of vegetated flow in both a lab and a field experiment. This topic is highly relevant. Field validation of the various proposed vegetation models is needed - especially in the smaller scale systems such as described by the authors. I highly appreciate the authors' contribution to this discussion by presenting these two interesting case studies. In line with the authors' stated desire to present an objective and transparent study, I do hope they publish the data as well, attached to a future version of this manuscript.

The authors' approach to compare three different conveyance models using proba-

bilistic parameter estimation is commendable. I do not share the authors' statement of complete novelty of this approach (p31, l493), and refer to the work of e.g. Reitan & Overleir (10.1029/2010wr009504) and Coz et al. (10.1016/j.jhydrol.2013.11.016), although I'm sure more may be found.

In general, the authors' presentation of the current state-of-the-art in literature is poor, with the notable exception of the introduction on vegetation models in the first half of the introduction. The chosen methodological approach has some well-known weaknesses, well discussed in scientific literature, but which the authors do not sufficiently discuss. For example, the conclusions drawn by the authors on parameter identifiability (conclusions nr. 1 and nr. 2) are contrary to what should be expected. These claims, if upheld, must be supported by better evidence and be placed in light of other, more recent literature.

Overall, I feel the authors focus too much on (sometimes trivial) details of the analysis, and not enough on the practical interpretation of their results. The widths of the confidence bands is just not very interesting, given the use of the informal GLUE approach. If succesfully applied, the confidence bands should describe the variance present in the dataset. I think the check whether the error statistics held up for higher discharges (the verification set) is more interesting, as this shows which model is better suited to present the data, and I'm happy to see the GTLM model performs so well in this regard.

However, this does raise fundamental questions as to the application of the physics-based models. The author's approach treats all models essentially as black-boxes, and calibrated them on a limited number of data points. This seems to defeat the purpose of physics-based models, and one would ask how the authors foresee application of the best tested model?

In conclusion, I believe the manuscript needs revision before publication can be considered. I do hope the authors find my remarks helpful, and I would welcome a revised manuscript for reconsideration.

[Figure]

**Specific comments**

P1-L10: 'quasi-Bayes'. The only time this term is mentioned is in the abstract. Perhaps a definition could be included in the manuscript.

P3-L58 the authors use 'process-based' models and 'physically-based' models seemingly interchangeably. I recommend choosing either 'process-based' or 'physics-based' (not 'physically-based', which is admittedly used throughout literature)

P3-L58 I do not share the authors' broad assertion that physics-based approaches are unpopular in practical applications.

P3-L62-64 The authors skip over many other possibilities by jumping from the detailed parameters of the Vastila and Jarvela models to the Manning coefficient. A (physics-based) model with fewer parameters would be an option. Numerous studies can be found in literature where two-dimensional models use spatially distributed information on vegetation, often based on remote sensing techniques.

P3-L72 This is a very valid point, and in my view the most important objective of this study

P3-L74 "Any method can ... a parameter calibration". I reject this statement. Do the authors like cake only if all its ingredients can be individually tasted?

P3-L77 'predictive uncertainty' is a technical term used differently by different authors. The authors should define their use of the term.

P3-L79 'As one of the first works, this paper' What paper do the authors refer to? (If they mean their manuscript, see my general remark on novelty).

P3-L82 "most of the previous studies": the literature review by the authors cites dated literature. To give confidence in this statement, the authors might provide a review of more recent literature.

P3-L89-90 I don't think this is a valid contrast. Morphologic and hydraulic modelling

are very different challenges and many existing vegetation models are not suited for morphological modelling.

P4-L95 The authors compare 'explicit' and 'implicit' uncertainty analysis. I'm not sure this terminology is commonplace, and in any case requires explanation.

P4-L116-L120 I'm unsure why the authors use the terms 'minor values' (text), 'minor parameters' (figure), or 'conservative approach'. To me this sounds derogative, as if to discredit this approach in favour of their proposed alternative, although I readily assume the authors do not intend this. For instance, I do not see why surface roughness is in any way 'minor'. In fact, if this parameter is used for calibration it is very likely the most sensitive parameter, so by all accounts should be labelled 'major'. Nor do I see why the first approach should be labelled 'conservative' (what is conserved? Do the authors mean 'traditional'?).
Second (and this point was raised by another referee as well), the 'conservative approach' is surely preferable - if reasonable estimates of the uncalibrated parameters are available - over treating vegetation models as black boxes.

P5 L134-135: For its merits in popularizing uncertainty analysis, the GLUE method is (in)famous for the liberal use of the likelihood measure, which does not agree with Bayes' theorem. The authors choose to use a so-called 'informal' likelihood measure with a scaling factor that controls the uncertainty. The authors then force the model uncertainty to include at least the right amount of data points through equation 5. This approach, inspired by Bayes theorem, has the known disadvantage that predictive and model uncertainty are lumped (the authors approximate the total uncertainty), that parameter uncertainty tends to be overestimated, and that the choice for a likelihood measure is arbitrary (i.e. not following from the error model, as is the case in a proper Bayesian approach). A defense by the authors on their choice for an informal over a formal approach would be appreciated for readers unfamiliar with this distinction. Also, given that the authors use an informal method, I'm interested whether a behavourial threshold was used (the authors mention the need for this, but not whether it was used.

I assume none was used given the scaling factore.)

P7-L194 Please elaborate which resistance (all, only the bed, only the imaginary wall?)

P11-L297 Is the Ritobacken Brook free flowing? Can uniform flow be reasonably assumed?

P13 In general, I suggest adding the first section of chapter 3 to the method section, as new methodology is introduced here.

P13-L315 By 'trial and error' choosing the sampling size of Monte Carlo, do the authors mean increasing the sample until convergence is observed? What convergence criteria is used? Which sampling method is used?

P13-L315 "In a similar way"; are a priori distributions chosen by trial and error? In principle a priori distributions are either informative, based on prior knowledge, or uninformative. Here an uninformative uniform prior is chosen, but I have to learn this from the captions in table 1. It would be helpful if this is explicitly added to the text as well. I would also appreciate a brief exposition of the choice for an uninformative prior. Given the models are physics-based, and the authors have a pretty good estimation of their likely values, it seems more logical to use informed priors.

P17-L365 Here the authors define "model identifiability" as (I paraphrase): "it is identifiable if it is fittable". The authors then admit that their approach would allow even poor models to fit well, while "the only limitation could be the physical meaning of the parameters". It is unclear whether the authors did indeed let themselves be restrained by the physical meaning of those parameters. One may remark here that minor changes to their chosen approach (i.e. a formal Bayesian approach and informed priors) would be expected to alleviate some of these problems.

At this point it is also good to remark on a different, perhaps more fundamental point. The authors go into depth into 'model identifiability' but the reader was led to believe

that 'parameter identifiability' was the objective of the study. Yet most results and almost all figures focus on the question 'will the model fit' - which is not a very interesting point to stress given the objective of the study. The only figures that support 'parameter identifiability' are 14 and 15, but those are currenty insufficient to support the claims made in the conclusions; it would be helpful to plot the a priori cdfs as well, so as to see how they were constrained a posteriori.

P31-L495 - The 'trial and error' a priori distribution estimations bothers me a bit when claiming objectivity. Given the limited number of observations, the a priori distribution is expected to affect the output.

P31-L500 'It was possible to identify...one (DCM)'. This should not be surprising. It is in general easier to fit a model with more parameters than one with fewer. However, the more impressive claim would be that the parameters are identifiable as well. I refer to the work of Werner et al. (2005, doi:10.1016/j.jhydrol.2005.03.012), to illustrate that challenge.

P32-L528 'Thus, our result... resistance dominated'. I do not see how this follows from your results nor from the previous sentence.

P33 Conclusions. The authors conclude the article with 8 claims.

Claim 1. The authors claim it is possible to identify the parameters of physics-based models, even if those models have many parameters. This is an unlikely claim (given the number of data points and the number of model parameters), but may follow from a confused definition of 'parameter identifiability' versus 'model identifiability'. If the parameters are 'identifiable', I would expect narrow a-posteriori distributions compared to the a-priori distributions. Figures 14-15 do not show a-priori distributions. Although it is difficult to judge whether the parameter distributions are meaningfully narrow, it seems only C is well defined. If so, it would be interesting to reflect on figure 1.

The second claim, like the first, seems to only apply to model identifiability, not parameter identifiability. The authors might spend some words on how they perceive the model to be used - does it matter if the models are physics-based? Or any (data-based) model that fits the rating curve applicable?

Claim 3: Perhaps the authors could explain how uncertainty relating to parameter equifinality can be distinguished from other uncertainty.

Claim 4, first sentence: Would not a better explanation be that the model is insufficient in some way, and that model parameters differ to account for this? Second sentence: I don't understand this in relation to the third claim.

**Technical corrections**

P2-L22 use plural for solutions
P2-L53 Use of the indefinite article would improve sentence
p3-L57 discussed by Yen p3-L58 For instance P3-L77 model's P3-L77-78 'The better model...uncertainty' please revise P3 - L86-87 'as it was assumed...their distributions' I don't understand this sentence P4-L92 check references

P4-L109 'It is out...the available methods'. Methods for what?

P5 L133 'fit measures' what are fit measures?

P8-L211 I count 8; which are the other two?

P14-L316 'It was was done'. In general, it is advised to explicitly state what you are referring to. In this case, I'm not sure what the authors mean by 'it'.

P20-L101: 'the probabilistic term'. Which term is this?

P31-L506: 'horological' (I assume 'hydrological', although I can accept the term if the model was setup around October 31st :) )

Table 1: I suggest to merge Tables 1-2 into one table.

Table 1 caption: It is not clear from the caption whether this is a priori or a posteriori distributiosn

Figure 4: The vertical axes of the upper figures are not equal

Figure 4 caption I don't understand what the authors intend by presenting 'exemplary' rating curves. Aren't these results?

Figure 8 Please explain 'n' here as well. Each figure should be independently understood

figure 15 caption: 'confidence intervals and median of a probabilistic solution'. I'm a bit thrown of by the indefinite article.. which solution are the authors referring to?

---

## Author Comment (AC1) · 26 Mar 2020

adam\_kiczko@sggw.pl

Received and published: 26 March 2020

**1 Major comments**

**Referee #1's major comments:**

The paper addresses an important and highly relevant topic and is of great importance for the professional community to improve the capabilities to accurately model the complex flow in vegetated compound channels. However, I identified various issues that should be addressed and clarified by the authors. In general, the paper has been written with great care, but there are many passages that should be improved stylistically
and where the language should be improved (see my detailed comments below; I recommend that the language is cross-checked by a native speaker). I also found that the terminology should be better introduced and defined to help the reader to better understand the complex content of the manuscript. For example, it became not really clear to me what is meant by, for example, identification data points, verification data points, computation points, observation points etc. Overall, the manuscript focuses on a statistical analysis of different approaches regarding uncertainties associated with input parameters. This is highly relevant, but the manuscript is mostly written from a statistical (probabilistic) point of view. However, in practice the tested approaches are typically used by hydraulic or environmental engineers, and it would be nice to outline the chosen approach more generally at the beginning of the manuscript, so that the significance of the results becomes clearer for the target-audience. In this context, I am not an expert in statistics and this fact triggered many questions (see below). In my opinion, this manuscript could make a real impact if it would be written in a way that practitioners will better understand what has been done. Also, more specific statements related to the parameter variation would be helpful. I acknowledge that this is partly addressed in the discussion, but it becomes not really obvious from the preceding sections. To summarize, *this is an interesting manuscript*. However, the presentation of the material should be improved. As I have many specific comments, I am afraid that I have no other choice than to recommend returning the manuscript to the authors for major revisions. I hope the authors will find my comments useful.

**Response:** We are pleased to hear that the reviewer acknowledges the relevance and importance of our work, and are very grateful to the reviewer for his comments. Especially, we would like to make this article interesting for practitioners, and we acknowledge the reviewer's many remarks on the clarity of this study. We agree that the study would benefit from extending the focus from statistics to practical implications/significance, and the manuscript will be revised accordingly, paying attention to introducing the concepts related to the uncertainty more clearly already in the Introduction. If the manuscript is considered for revision, we would like to address comments

by providing better explanations of the study scope, definition of the terminology used, and a clearer presentation of the applied methodology of uncertainty analysis. We will have the language checked.

We also agree with the reviewer's suggestion, that more specific statements should be given in the case of a priori parameter variations. This issue was also mentioned by other reviewers, and we are sure, that we should provide a detail explanation of our approach in that matter. First, we would like to introduce term of uninformative parameter bands for the a priori distribution. We think it is the only way to compare different methods, as we assume that the modeler has no prior knowledge on channel resistance. Moreover, that parameter ranges should be large enough to ensure that the highest probability region of the probabilistic solution is not affected by them. We would like to include in the methodology section a subsection, where these issues would be explained.

**2 Specific comments**

We provided answers along with reviewers specific comments:

1. Title: I am not sure that I understand what is meant by "identifiability" – I guess a "statistical" meaning is meant, but that becomes not clear.

**Response:** With "identifiability" we refer to the possibility of finding a distribution of model parameters that explains its uncertainty and follows physical constraints set on the parameter values, assuming that the modeler's knowledge is restricted only to water level and discharge measurements. Brief explanations can be given immediately in the abstract. We will consider changing the title of the manuscript to something more generally understandable.

2. P2, L22: What is meant by stability?

**Response:** Morphological stability of the channel system and this way the sentence should be clarified.

3. P2, L25: I find this a bit confusing, as there is some redundancy with the sentence before. This could be formulated better.

**Response:** We will revise lines 22-25 as: “Such nature-based solution (NBS) allow combining the technical needs, e.g. [FB02?]ow conveyance and stability, and the environmental requirements, e.g. improved water quality and biodiversity (Rowinski et al., 2018), but require reliable predictions on the discharge capacity. Predictions using the conventionally applied methods (e.g. Posey, 1967) can . . .”.

4. P2, L27: What exactly is meant by “regions”? I assume the different channel parts are meant (but one could also think about different geographical regions).

**Response:** The reviewer is right, the term “parts of channel with highly different flow resistance” will be more appropriate and will be used in the revised manuscript.

5. P2, L33: Is this approach really simple? I would delete the latter word.

**Response:** Comparing to Pasche and similar methods DCM is very simple, but we understand that this appears before introducing these methods. The word should be removed.

6. P2, L36: Please check language.

“Despite the development of more advanced methods, providing often much more detailed and physically based description of channel flow resistance, DCM is till this day found in the majority of practical models for flood hazard assessments, design of hydraulic structures or water management.”

**Response:** We will revise the sentences so that the text will flow better: “DCM is presently the basis for the majority of practical models for flood hazard assessments, design of hydraulic structures and water management.”

Printer-friendly version

Discussion paper

**7. P2, L41: Which relationship is meant here?**

"It should be noted, that this relationship can be amplified with inadequacy of a flow model, as mentioned by Yen (1999)"

**Response:** The sentence should be rewritten: "It should be noted, that such dependence on flow can be amplified with inadequacy of a flow model (Yen 1999)."

**8. P2, L42 & P2. L43: Check language**

"A number of studies were devoted developing a more process-based description of channel flows (Yen, 2002).

**Response:** The 2 sentences will be revised to: "A number of studies were devoted to developing a process-based description of either the channel flow processes or the interactions with elements obstructing the flow, such as vegetation (Yen, 2002)."

**9. P2, L44: Why "should"?**

**Response:** Instead of "should", we will write: "... the most sophisticated model of the channel capacity can be attributed to Shiono and Knight (1991),"

**10. P2, L47: The approach was published in 1991, and the only reference for its successful use is from 2020? I doubt that it took 30 years that it was successfully used...**

**Response:** The reviewer is right that we should have included here other references, as well, e.g.:

- Abril, J.B. i Knight, D.W. (2004). Stage-discharge prediction for rivers in flood applying a depth-averaged model. *Journal of Hydraulic Research*, 42 (6), 616-629.;

- Babaeyan-Koopaei, K., Ervine, D.A., Carling, P.A. i Cao, Z. (2002). Velocity and Turbulence Measurements for Two Overbank Flow Events in River Severn. *Journal of Hydraulic Engineering*, ASCE, 128 (10), 891-900.
- Kordi, H., Amini, R., Zahiri, A. i Kordi, E. (2015). Improved Shiono and Knight method for overflow modeling. *Journal of Hydrologic Engineering*, 20 (12), 04015041.
- Sharifi, S., Sterling, M. i Knight, D.W. (2011). Can the application of a multi-objective evolutionary algorithm improve conveyance estimation? *Water and Environment Journal*, 25 (2), 230-240.
- Shiono, K. i Rameshwaran, P. (2015). Mathematical modelling of bed shear stress and depth averaged velocity for emergent vegetation on floodplain in compound channel. In: E-proceedings of the 36th IAHR World Congress, 28.
- Tang, X. i Knight, D.W. (2008a). A general model of lateral depth-averaged velocity distributions for open channel [FB02?]ows. *Advances in Water Resources*, 31, 846–857.
- Tang, X. i Knight, D.W. (2008b). Lateral depth-averaged velocity distribution and bed shear in rectangular compound channels. *Journal of Hydraulic Engineering*, 134 (9), 1337–1342.
- Tang, X., Sterling, M., i Knight, D. W. (2010). A general analytical model for lateral velocity distributions in vegetated channels. *River Flow 2010*, 469-475.
- Zhang, J., Zhong, Y. i Huai, W. (2018). Transverse distribution of streamwise velocity in open-channel flow with artificial emergent vegetation. *Ecological Engineering*, 110, 78-86. doi: 10.1016/j.ecoleng.2017.10.010

**11. P2, L48: Please check language**

**Response:** The term “physically-based” should be changed to “physics-based”.

12. P2, L49: Why referring specifically to Pasche, and not to Pasche & Rouvé?

**Response:** The reason is that, the methodology was developed by Pasche and this is the only English article presenting its basis. The reviewer is however right and to avoid confusions we shall include also references to original Pasche works in German.

13. P2, L53: Please check language

**Response:** Sentence should sound: “A simplified version of the method was proposed by Mertens (1989).”

14. P2, L55: Check how reference is included into the text.

**Response:** Thank you for pointing this out.

15. P3, L57: I doubt that all of the cited approaches have been developed to parametrize the two-layer approach. Some of them deal more with the parameterization of vegetation properties.

**Response:** A straight-forward two-layer flow description was proposed by Luhar and Nepf (2013) while Västilä and Järvelä (2018) generalized the approach, incorporating a parameterization of flexible foliated vegetation (Västilä and Järvelä 2014; Jalonen and Järvelä 2015).

16. P3, L58: Please check the sentence (“. . . for in. . .”).

**Response:** “for” should be removed.

17. P3, L60: Please improve “Methods like Pasche and Rouvé (1985). . .” Pasche and Rouvé are the names of the authors who developed the method (a similar comment can be given regarding the reference to Västilä and Järvelä in the same line).

**Response:** We will revise the sentence (and similar sentences) to: “Methods such as those developed by Pasche and Rouvé (1985). . .”

18. P3, L62: Is this really true? Remote sensing methods have significantly developed, and dependent on regions and countries, such information may be available. . .

**Response:** There are studies where remote sensing techniques have been used to estimate vegetation properties for hydraulic modeling. However, such studies present rather the state-of-the-art in research, while in many practical assignments the vegetation data is mostly insufficient for these approaches. We understand, that this point is confusing, and we will specify that this statement applies to the practice and will provide a comment on remote sensing techniques.

19. P3, L65: In my opinion, this is not an argument for simpler methods. Such a non-physical based black-box will not help to better understand the problems at hand and requires manual and arbitrary calibration.

**Response:** We will clarify our message and revise the preceding and following sentences to: “With these practical limitations, the use of a roughness coef[FB01?]cient lumping all effects, such as the Manning coef[FB01?]cient, can provide a reasonable prediction (i.e. Marcinkowski et al., 2018, 2019). However, the common practical approach of adjusting the roughness coef[FB01?]cents to fit the model to observations may lead to the coefficients to be used beyond their physical interpretation, as discussed by Yen (1999) in the response to Khatibi et al. (1997).”

20. P3, L67: Why disused? What is meant by this?

**Response:** We apologize for the spelling mistake. “Disused” should read “discussed”, i.e., we refer here to the exchange of discussion papers between Yen and Khatibi et al. We found it very relevant to presented study.

21. P3, L68: Fr typically defines the Froude number – but I guess here the word “For” is meant? I stop here giving particular comments on the language, as I already have provided many such comments showing the need to improve the paper.
**Response:** Apologies for misspelling, should be “For”. We will carefully check the language in the revised manuscript.

22. P3, L69: I do not understand this statement – “when bathymetric data do not account for the true complexity of the river geometry” – what is meant by “true complexity of the river geometry”?

**Response:** We meant here the situation where assumption of the linear evolution of geometric features between two cross-section is not maintained. So i.e. when in reality between two cross-sections there are some irregularities in channel geometry, which are not shown by the cross-sectional geometry data. We will address reviewer remark by revising our sentence to: “when cross-sectional geometry data do not account for the true complexity and irregularity of the river geometry between the cross-sections”.

23. P3, L70: This example is not really related to the topic of the paper. . .

**Response:** We agree that it is not related to the vegetative flow resistance, but in our opinion this perfectly links with the problem of using formulas with parameters beyond their physical meaning.

24. P3, L72-75: This is difficult to understand – please improve.

**Response:** We will clarify this sentence.

25. P3, L77: Which answer? In other words, what is the question?

**Response:** We agree, the sentence should be rephrased as follows: “This leads to an old dilemma, where a simple model with limited number of parameters is compared with a complex one with more parameters (Kuczera and Mroczkowski, 1998), which can be addressed through the models’ predictive uncertainty.”

26. P3, L81: “. . .comparing to the Manning. . .” – this part of the sentence remains unclear to me.

Printer-friendly version

Discussion paper

**Response:** It can be added for the clarity, “comparing to the Manning”.

27. P3, L86: I do not understand what is meant by “parameterized in a sense of their distributions”

**Response:** The sentence should be rephrased. The chosen sources of uncertainty were described using probabilistic distributions, reflecting modelers' expectations on possible variability of each input.

28. P4, L93: Improve stylistically ( . . . Järvelä Järvelä. . . )

**Response:** We apologize for the misspelling.

29. P4, L95: Which study? This one?

**Response:** Should be “In this study”.

30. P4, L99: “The overall goal of this paper is to compare the uncertainty, parameter identifiability and physical interpretation of the parameters of discharge capacity methods characterized with different levels of parameterization”. This sentence is very difficult to understand.

**Response:** The sentence can be rewritten as follows: “The overall goal of this paper is to compare the uncertainty, model identifiability and physical interpretation of the parameters of chosen discharge capacity methods. The challenge arises from different number of parameters in each model.”.

31. P5, Figure 1: Please improve the caption and the description of the figure; I find it difficult to understand (note also that not all parameters have been defined)

**Response:** The caption will be improved.

32. P5, L120 – 125: Please improve – this could be explained more clearly in my opinion.
**Response:** Following also other reviewers remarks on the clarity, we agree that the section 2.1 should provide much better explanations of our approach.

33. P9, L242: “. . .are plant species. . .”? I am not sure that I understand what is meant.

**Response:** Should be rephrased “are factors specific for plant species or plant type”

34. P9, L247: This depends on the level of submergence – otherwise 20% of the discharge may be neglected. . .

**Response:** The reviewer is correct that with the simplified two-layer model (STLM), up to 20% of the discharge is neglected, depending on the density and cross-sectional blockage of vegetation. This share of discharge results from back-calculating from the original simplified approach proposed by Luhar & Nepf (2013). We will add this information to the text.

35. P9, L251: This depends - for the typically used rigid cylinder analogy,  $B_x$  will basically be constant.

**Response:** We would like to remind that  $B_x$  represents the bulk-level cross-sectional scale distribution of vegetation. Thus,  $B_x$  of any type of vegetation will depend on the changes in the wetted cross-sectional area resulting from changes in water level. The effect will be particularly large when the water level rises above the height of floodplain vegetation, in which case the  $B_x$  typically starts to decrease.

Note, that in general case,  $B_x$  value is also affected by the cross-section geometry.

36. P9, L253: What exactly is  $|_l$  and  $|_r$ ? In this context, a sketch would be helpful.

**Response:** If the manuscript is considered for the revision, we will include here a sketch illustrating the way how these parameters are applied.

37. P10, L264: The model of Luhar and Nepf was already mentioned before - it may be a good idea to restructure the manuscript and to present this approach earlier? Also, this is not the “original formula” (which should be stated more clearly), as the hydraulic radius is used while Luhar and Nepf used the water depth.

**Response:** We agree this is already modified model, however the modified Luhar-Nepf model was introduced with eq. 6-7.

38. P10, L269: Check writing style (...“formula 9...”, “...three parameter one...”) etc).

**Response:** Thank you for pointing this out, we aim to be more specific. Lines 269-271 should read: “Eq. (9) has a convenient form to be easily applied in practical cases, where usually the Manning equation is used. In the present study, this approach is called the Practical Two-Layer Model (PTLM) as it requires less parameters influenced by vegetation.”

39. P10, L280: Remove the full stop after “experiments”

**Response:** Thank you.

40. P10, L290: Strictly speaking, uniform conditions are impossible to achieve by this setup - I would prefer if the terminology “quasi uniform” is used. The flume slope (or the slope range) should also be given - I could not find this information in Koziol (2010) as this paper seems to be in Polish language.

**Response:** Reviewer is right, we will provide additional comment on the flow uniformity and references, where this issue for the WULS-SGGW flume was discussed. Also the information on the channel slope is missing and should be provided. For the the WULS-SGGW flume the slope was:  $s = 0.0005$ .

41. P10, L291: How were water levels recorded and what was the spacing between the measurements?

**Response:** We used a pressure gauge, that allowed us to measure differences in depths between downstream and upstream sections of the flume at the distance

of 4.8 and 12 m from the inflow to the channel (the length of the channel was 16 m). The setup of the experiments in the WULS-SGGW flume is given in the recent articles of Kubrak et al. 2019 and less recent of Kozioł (2013). References and explanations should be included in the article text.

- Kubrak, E., Kubrak, J., Kuśmierczuk, K., Kozioł, A., Kiczko, A., & Rowiński, P. M. (2019). Influence of stream interactions on the carrying capacity of two-stage channels. *Journal of Hydraulic Engineering*, 145(4), 06019003.
- Kubrak, E., Kubrak, J., Kozioł, A., Kiczko, A., & Krukowski, M. (2019). Apparent Friction Coefficient Used for Flow Calculation in Straight Compound Channels. *Water*, 11(4), 745.
- Kozioł, A. P. (2013). Three-dimensional turbulence intensity in a compound channel. *Journal of Hydraulic Engineering*, 139(8), 852-864.

42. P11, L300: What kind of vegetation?

**Response:** We will add the information that it was mainly grassed vegetation, consisting of different species with both stems and foliage.

43. P13, L320: What is constrain 5? Is equation 5 meant?

**Response:** Yes, we meant constraint given with the eq. 5. The sentence should be rephrased.

44. P13, L328/330: Please use another notation for the number of observation points before,  $n$  was used to define the Manning coefficient, and this is confusing (see also my comment below; L336). In this context, what exactly is meant by observation point? It could also be the number of points Also, it should be mentioned that only floodplain flows were investigated (for convenience of the reader).

**Response:** We agree with the reviewer, we will use another symbol for the number of observations. Also we will stress that calculations were performed only

Printer-friendly version

Discussion paper

for flows higher than bankfull condition. By the observations we meant the measured water level  $H$  and flow rate  $Q$ . Each “observation” consisted of a single pair of these values. We agree that this should be explained clearer.

45. P13, L334: Not necessarily in some (or many) cases there exist data for high flows that can be used for calibration.

**Response:** We think this is a terminology issue and agree that the term “low” flow is inaccurate. In both cases we consider floodplain flows, but of low vs higher floodplain water depth, and we will rephrase the sentence accordingly.

46. P13, L336: I am partly confused here,  $n$  is the number of observation points which could also be the number of readings taken for the water depth measurement. Please be more specific.

**Response:** As in previous response, we would like to clarify the terminology concerning the number of observations.

47. P13, L346: Please improve cross-references - e.g., here it should be Figure 4a. Figure 4d-f: Why are the lowest two points characterized by the same discharge?

**Response:** These are two independent water level/discharge measurement and differences in water level value comes from measurement uncertainty.

48. P15, L349: I am not sure that I can follow - this could be explained better (what is exactly is meant by computation points).

**Response:** As in previous response, we would like to clarify the terminology concerning the number of observations.

49. P16, Table 2 (also Table 1): Could you comment on the used parameter bands in the text (why were these bands chosen?)?

**Response:** This issue was also raised also by another reviewer. We used “uninformative” ranges for parameters, although within physically interpretable bands.

Explanations given in the text are insufficient and we would like to add explanations in the methodology section, explaining our approach.

50. P16, L353: What exactly are identification data points?

**Response:** It should be “observations used for identification”.

51. P16, L355: But the Pasche model results are not included in Fig. 5? This is confusing.

**Response:** We apologize that the reference to the figure 6 is missing (and the the wrong case number was given)

52. P16, L356: (5a, 6a) - does this refer to the Figures? Please improve throughout the manuscript.

**Response:** The reviewer is right that we should cross-check all references throughout the manuscript

53. P17, Figure 5: I am not sure that I understand what is meant by “Ratio”.

**Response:** Probably the word “share” would be better.

54. P17, L365: This definition could have been given earlier.

**Response:** We agree, the definition should be given and explained in the methodology section.

55. P17, L369: Here, equation is used, at other places formula is used and at some places numbers are just given. Please improve - this is confusing. Also, the used model could be specified more precisely (the same applies to “other unspecified models”).

**Response:** We are sorry for causing such unnecessary confusion and will be consistent.

Printer-friendly version

Discussion paper

56. P17., L370: Now n is the ensemble count - this is confusing.

**Response:** Should be number of observations.

57. P18, L374: I am not a statistical expert (although I have some knowledge regarding statistics), but this is a bit confusing....

**Response:** We understand it is unclear. In other words, we meant, that with that model it was impossible to reproduce a rating curve that was able to explain much more than a single observation point.

58. P18, L378: Why is 1 an extreme value?

**Response:** The sentence is unclear, it should be explained that, for that model only for some sets it was possible to find a solution that was able to explain several observation points.

59. P18, L380: This fact could be explained in some more detail when outlining the approach. I am getting a bit lost here...

**Response:** Our idea to address the reviewer remarks on the clarity of presentation is to add a section in the methodology, supplemented with a schema, explaining our approach.

60. P18, L383: Vegetation characteristics of Ritobacken have not been defined; what is meant by a “flexible approach”?

**Response:** The vegetation in the Ritobacken should be considered as flexible. The vegetation characteristics will be included in the description of the Ritobacken case study.

61. P19, L396: I have trouble understanding this - a more general outline of the procedure would be helpful (this should be provided earlier, not here in the presentation of the results).

P26, L416: I am not sure that I understand what is meant here. I stop giving more comments here on chapter 3 as I have problems to understand what exactly was done - the procedure could be outlined in some more detail

P31, L492-498: It would be good to explain all this earlier in some more detail.

**Response:** We thank the reviewer for helping us to understand that we need to add a detailed explanation of our approach in methodology section.

62. P31, L508: Isn't this rather obvious? By the way, what about errors in the measurements - how would they affect this analysis?

**Response:** It is true, but with this statement, we would like to show that we were able to reproduce this effect using our approach. In the case of measurement uncertainty: we should mention in the methodology section, that we analyze the total uncertainty - of the model and also measurement. With our approach it is impossible to distinguish these two sources. The remark should be addressed with a detailed explanation of the adopted uncertainty analysis.

63. P33, L578: Numerical experiments are mentioned - but I doubt that detailed numerical simulations were carried out (no statements are given in the manuscript); this again shows the need to formulate statements more precisely.

**Response:** The remark links to the clarity in presenting our approach. We hope, we would be able to explain it better with additional subsection of the methodology.

---

## Author Comment (AC2) · 26 Mar 2020

Dear Reviewer, We are grateful for remarks, we hope they will allow us to improve the manuscript text. Bellow, we provide answers to reviewer comments:

1. L7-8 I think this sentence is not clear. Does this mean that the accuracy (rather than uncertainty) does not have to be considered?

   **Response**: The sentence should be rephrased: "We developed a new probabilistic approach for comparing six models of channel discharge capacity in respect of their uncertainty. The model with the lowest estimated uncertainty, that explains

differences between computed and observed values, should be considered as the most favorable."

2. L18 (general remark on the Introduction section) The paragraphs and sentences need to be rearranged and reorganized to improve the flow. In the current form of the introduction section, ideas are scattered, and similar statements and descriptions are found at different places.

   **Response**: If the manuscript is considered for the revision, we will improve the clarity of this section.

3. L120-121 The beauty of using a process-based method is that we may be able to measure or observe the values of its parameters (rather than calibrating them). The concept sounds like the authors treat the process-based methods as conceptual or black-box models. I agree with the idea that all models will become conceptual or black-box at certain spatial and temporal scale. Please discuss the theoretical background and implication of this method somewhere in the manuscript to more clearly contextualize this study.

   **Response**: This issue is also raised by the third reviewer. With our study, rather than advocating for identifying in the inverse manner all possible parameters, we would like to discuss "what if these physical parameters are identified". So, we agree that we should stress the advantage of physic-based methods and provide a better explanation of our view. The reviewer comment on the effect of spatial and temporal scales on physical meaning of models, is exactly the way how we consider application of these methods in practice. If the article is considered for revision, we would like to add a developed discussion on these issues.

4. Table 1: Please elaborate how these boundaries were determined (Tables 1 and 2).

   **Response:** We adopted the uninformative a prior parameter distributions, however maintaining physical variability ranges of parameters. In an adopted pro-

cedure, the width of a priori parameter ranges were wide enough to reflect the total model uncertainty. This was obtained by trial-and-error with the objective to ensure that the high probability region of the final solution is enclosed within these bands. If applicable, we will provide a much more detailed description of this procedure.

5. Table 1: The sizes of samples are different depending on the models (and the numbers of their parameters), but I could not find any consistency. Please elaborate how the sizes were determined.

   **Response:** The reviewer is right, we estimated the number of Monte Carlo simulations, by analyzing the convergence of the water depth mean. Such an explanation should be given in the revised article.

6. L375 I think the use of a more advanced sampling technique such as the Latin Hypercube sampling can help cover the extremes.

   **Response**: The reviewer is right, hopefully we already used Latin Hypercube sampling (uniform). The information should be given in the text.

7. L376-377 This is also a function of the size of the Monte Carlo samples and the parameter value boundaries determined in Tables 1 and 2.

   **Response:** Our goal in designing a prior parameter ranges, was to ensure, that the solution is independent of a priori parameter distributions. This was a reason for uninformative parameter ranges and relatively wide ranges: to ensure that a high probability region is enclosed within the whole sample. More detailed explanations on that matter should be included in the revised manuscript.

8. **Figures**: Figures and tables should be located right after paragraphs that first mentioned them in the manuscript. I found many of the paragraphs are not followed by the corresponding figures and tables.

   **Response:** We will work on the manuscript layout.

9. L539-541: It should be useful if the authors can discuss the hydraulic background and implications of this finding.

**Response:** This applies to the case with dense and flexible vegetation. According to our results, this well agrees with the assumptions of GTLM model, which very well in terms of accuracy, explains the rating curve. Also, the PTLM had good performance and only of two layer approaches, the STLM, where vegetation flow is neglected, had a poor accuracy. We think, that the comment in the manuscript can be developed.

---

## Author Comment (AC3) · 26 Mar 2020

**1 Major comments**

First, we would like to express our gratitude for the reviewer's efforts and constructive remarks. We are grateful for the reviewer's view on the relevance and need of the conducted validation of the vegetation models at such smaller scale channels. The reviewer raises several important issues that we would like to address in our response with more detailed comment. We identified the following crucial points from the general and specific comments:

[Figure]

1. The drawbacks of the uncertainty analysis and the interpretation of obtained confidence intervals.

2. Definition of the identifiability, "parameter identifiability" versus "model identifiability".

3. A priori parameter distributions, ranges should be informative or uninformative?

4. Can we still consider a model as physic-based, if all essential parameters are being identified with very general assumptions on their variability ranges?

5. Insufficient evidence for findings presented in conclusions.

6. Insufficient literature review and positioning the study in the present state-of-the-art.

7. Presentation of the results for higher discharges

8. Data publishing.

At the end of our response, we also provided more brief answers on remaining reviewers specific comments.

**Ad. 1. The drawbacks of the uncertainty analysis and the interpretation of obtained confidence intervals.**

As the reviewer indicated, we used a term of the informal uncertainty estimation in the respect of applied GLUE methodology. We agree, that this might suggest that results of uncertainty are not reproducible, making comparison of models unreliable. We were aware of this problem and although GLUE method is in its general form informal and based on the modeler subjective assumptions on the model uncertainty, we took advantage of the approach presented by Romanowicz et al. (1996) and Romaowicz and

Beven (2002), equivalent to Bayes identification with a simple error model of a normally distributed white noise. The observation equation is then:

$$H(Q, \theta) = \hat{H} + \epsilon$$

where $\epsilon$ stands for the noise, $H(Q, \theta)$ modeled water levels for given parameters $\theta$ and $\hat{H}$ observations. As steady state models are considered, the use of the white noise model is justified, as autocorrelation is not present. The noise is 0-mean with unknown variation. In common GLUE approaches, the variation is chosen subjectively by a choice of likelihood function and so-called "behavioral" parameters. In our study, we formalized this step using constraint on number of observations falling within confidence intervals (Eq. 5). Along with eq. 3 this allows to identify $\kappa$ factor, which multiplied by $\sigma^2$ (variation of model residual for scaling) provides a variation of the noise. All together, assessing the widths of the a prior parameter distribution by "trial and error", to ensure that they are wide enough to explain model uncertainty and the Eq. 5-kind constraint, makes the approach similar to the adaptive Monte Carlo sampling demonstrated i.e. by Blasone et al. (2008). We have not applied automation in determination of a priori parameter ranges, although the concept is the same.

To reduce the effect of the a priori distributions (note, always present with finite bands) we used a relatively wide a priori parameter ranges. The goal was to ensure that a high probability region is enclosed within Monte Carlo sample. This was tested by setting such ranges, although keeping their physical interpretation, that widths of confidence intervals were sensitive for $\kappa$, above values found by minimization task given by eqs. 3-5. So, it was always possible finding wider confidence intervals and the a posteriori 95% quantiles of computed water levels were noticeably narrower than the spread of MC sample. This ensures that within region of interest (explaining the model uncertainty) the solution becomes insensitive to the spread of a priori distribution.

With a given error model, as the reviewer mentioned in his comments, applied analysis accounts only for the total uncertainty. It is not possible distinguishing other sources

like measurement error. We think however, that providing additional term e.g. for measurement errors would make the comparison of discharge models difficult, as we can expect different estimations of measurement uncertainty for different methods but for the same data sets. For sure this leads to the overestimation of parameter variability ranges, but it applies to all methods in similar way.

We hope above explanation will satisfy the reviewer, as our approach to the uncertainty estimation is based on formal assumptions and allows for comparison of different models. We thank the reviewer for pointing out that such description is missing in the manuscript and should be given in the revised version.

**Ad 2. Definition of the identifiability, "parameter identifiability" versus "model identifiability"**

We thank the reviewer for raising the issue that the "parameter identifiability" was in some parts of the text erroneously used when "model identifiability" should have been used. The reviewer suggested, that the term identifiability should be clearly defined. In the previous version of the manuscript we gave a remark on our understanding of this term (line 350, sec 3.2.: "The model identifiability is understood here as the ability to determine the parameter a posteriori distribution that explains the model uncertainty in relation to observations. This is satisfied by meeting the constraint given in Equation 5"), although we agree that this definition should be developed. Applying the probabilistic identification problem we consider that the model is identifiable, if it is possible to find a posteriori parameter distribution that explains the total model uncertainty in the respect of observations. The criterion for identifiability is the constraint given by Eq. 5 – 95% confidence intervals should enclose not less than 95% of observations.

The other issue, that reviewer pointed out, is that if we consider model identifiability or parameter identifiability. So, the question is about the aim of the analysis: obtaining a model with a good prediction skills or estimate parameter values that agree with measured values. Our concept for the article was however different, although the reviewers'

comments have helped us to realize that we did not present our concept well. With the inverse identification problem, the goal is always to identify the model and it was the same in our study. Having identified the model, in the second stage, it is possible to analyze, if obtained parameters agree with their true values. Our results (for Pasche and GTLM) indicate, that for many parameters (but not for all) the median of the solution noticeably differs from measured values. The obvious problem is of course, if a model that well explains the rating curve, but its parameters might be different from their real physical values, should be used for water level predictions? At this point, probably not emphasized in the text, instead of advocating for our approach, we would like to present a discussion on that issues, as in our opinion the model identifiability is not the written assumption in many studies. With our article we would like to provide cons but also pros for the identification of models with a strong physical interpretation. Because, although parameters might be different, which sparks an impression of black-box modeling (more comments on that issue in the section on black-box modeling), differences are usually interpretable. The shift in a given parameter is compensated by others, i.e. the large stem diameter comes along with too large spacing of plants for the Pasche method. In our opinion, the ability for such interpretation might be considered as an advantage of the more physics-based models over simpler ones, as if modeler is aware of parameter interactions and can decide, if e.g. given before discrepancies in vegetation characteristics are important in analyzed case. Moreover, having easily interpretable parameters values, in contrary to e.g. Manning coefficient, it might be possible to recognize their unrealistic values, resulting from other model errors like improper representation of a geometry: tree trunks with diameters over several meters are much more evident than Manning roughness coefficients ranging for the floodplain at 0.1-0.2 $m^{-1/3}s$. However, we do not to show that it is possible to identify model parameters but show what is the effect of model identification, also in terms of estimated parameters.

If the manuscript is considered for the revision, we would like to improve the text by providing a clear definition of the identifiability and explain that our aim is the model identification, the output of which are also parameter values. We would also provide a

discussion on the use of models, the parameters of which might be different from the real ones.

**Ad. 3. A priori parameter distributions, ranges should be informative or uninformative?**

We have not introduced terms of informative and informative a prior parameter distribution. Our idea was to formulate identification problem the same way that it is done in most practical case studies of flood flow modeling, where usually there is very limited data on vegetation properties in a river channel. So our a prior distributions are uninformative and provide a wide region, where the solution for a general case, different channels, might be found. Moreover, using informative parameter bands would introduce a subjectivity to the study and it would be impossible to compare different methods. I.e. how to apply similar constraints on stem diameters in Pasche method and Manning roughness coefficients in DCM?

The reviewer's idea to take advantage of known parameter values is however attractive. It might be interesting to investigate how parameter identifiability and uncertainty estimates are affected when a prior parameter variation is reduced with additional knowledge. We think however, that this could be the scope of another study.

We would like to address the reviewer's remark on the type of the a priori distribution by directly defining that we use uninformative a prior distribution and explain more clearly our concept at this point.

**Ad. 4. Can we still consider a model as physic based, if all essential parameters are being identified with very general assumptions on their variability ranges?**

The reviewer raises an important issue: is it still a process based model, and not a black-box, if most of its physical variables are identified through an inverse problem (calibration)? The process-based methods are indeed functions with large number of parameters and when their physical interpretation is neglected, they indeed might take

a form of formulas rather explaining the data, than providing an insight into the process itself. This would make the task similar to the problem of estimating the rating curve, as in studies mentioned by the reviewer. To maintain the physical interpretation of models it is necessary to ensure that parameter values are restricted by physical constraints. In our study we used non-informative a prior parameter distribution, although within physically possible ranges of a given parameter. We analyzed, if these physical constraints allow eliminating an inappropriate method and in the case of the two-layer and Pasche approaches we succeeded: GTLM had very poor prediction skills when applied to rigid-unsubmerged vegetation, while Pasche, Mertens methods were unidentifiable for flexible submerged vegetation cases.

However, physical constraints on parameters values might be insufficient, as it is possible to identify values much different from real, measured ones. So, i.e. does the Pasche model maintain its physical consistency, when used with much larger spacing and larger stem diameters, than those measured from flume experiments? As we already mentioned in the comment on the identifiability, these are issues, which we wanted to demonstrate with the study (parameter interpretation section), rather than answer directly. Please note, that possible conclusions apply as well to Pasche or GTLM, like to the Manning formula. The difference is that in the first case the interpretation is obvious, whiles large values of Manning coefficients are common in practice. So maybe this is an advantage of process based approaches, where parameters are easily interpretable? They can be identified and modeler can validate if obtained values follows his/her exceptions.

On the other hand, the use of the inverse problem to determine parameters values, is not uncommon approach even for "very" process-based approaches, like i.e. Shino-Knight model, where it is necessary to identify turbulence parameters (like turbulent viscosity and secondary/advection flow term) and it is known that outcomes are affected by high equifinality (Knight et al. 2007). In our opinion, such problem will apply to all process-based methods, when applied in general practice task. Therefore,

we would like to avoid impression that we suggest using the physic-based models as black-box ones, but our aim was to investigate the usually unstated assumption on the identifiability of such models.

The discussion of these issues was missing in previous version of the manuscript, but we would like to include it in the revision.

**Ad. 5. Insufficient evidence for findings presented in conclusions.**

The reviewer indicated that conclusion are not well-supported with the manuscript text. In the case of conclusion points (claims) 1 and 2, the issue will be clarified by specifying that the primary goal is the model identification (please see the note on identifiability), as the reviewer suggested.

We agree with the reviewers remarks on conclusion points 3-4. With our approach we are unable to separate sources of uncertainty, including the equifinality. The equifinality is present, and it can be seen in parameter distributions, where wide regions in parameter space can be considered highly probable (high values of likelihood measure). In the case of uncertainty and the number of observation it is not a matter of equifinality but the ill-posed inverse problem (insufficient number of observations). The conclusion points 3-4 should be revised as follows:

3. The uncertainty related to the **ill-posed inverse problem** is noticeable only when a small number of observations is used in parameter identification.

4. The parameters obtained through the identification differ from their measured physical values, which results from the parameter equifinality. The equifinality does not, however, affect the uncertainty of a model.

The way, how this effect can be traced should be explained. It can be done by interpreting obtained average confidence widths as a function of the number of observation points used for identification (Appendix A, Figures A1-A25). Wide confidence intervals, and their spread for the small observation number n=1 (following the reviewers remarks

the symbol should be changed) should be attributed to the ill-posed inverse problem. Additional data points allow to narrow confidence intervals and reduce their spread, among different observation sets. In the manuscript text we provided for each method the number of observations, at which the width of confidence interval stabilizes. Additional observations affect the solutions but not widths of confidence intervals. Note, that our analysis at this point is rather descriptive. We understand that additional observations (of water levels) do not noticeably affect the estimates of uncertainty bands, so the effect of ill-posed inverse problem becomes negligible.

The reviewer remarks will be addressed in the text by clarifying our understanding of the model identification and in the case of "parameter equifinality" correcting conclusions as presented and providing a discussion of the ill-posed inverse problem.

**Ad. 6. Insufficient literature review and positioning the study in the present state-of-the-art.**

We agree with the reviewer, that the literature review can be improved and references to studies such as these provided by the reviewer as an example, should be included in the manuscript. However, in our opinion the research problem is different than in given examples. In our case we address the parameter identification problem with variables having specific physical meaning (i.e. Manning roughness coefficient or vegetation height, density). The point was not only to obtain an efficient estimator of the water level-flow dependency, but investigate, if it can be obtained using physically interpretable models and then if interpretability is maintained in terms of parameters determined through the inverse problem. Examples of such approaches might be found in hydrology, as we hoped we presented in the manuscript, but not in hydraulics. The only exception might be a study of Berends (2019), analyzing inverse identification of Delft3D model, which we found after submitting our manuscript. This is the most similar study, although focused on the single but distributed model, and we will cite that publication.

If applicable, in the revised version of the manuscript the description of how we consider our study novel should be improved. Also the review of the state-of-arts should be improved and positioning of our study in the respect of rating-curve fitting should be discussed.

**Ad. 7. Presentation of the results for higher discharges**

The reviewer indicates that for the scope of research, it would be interesting to present the results, when the model is verified for higher discharges. We think it is a very good idea and if applicable we would like to add an additional section to the results, where i.e. discharge curves obtained with models identified using lower flows are analyzed in respect of the accuracy in predicting higher flows.

**Ad. 8 Data publishing.**

The reviewer suggests, that the revised version of the manuscript should include data we used in computations. We agree at this point, and we are ready to include our data sets.

**2   Responses to specific comments**

1. P1-L10: 'quasi-Bayes'. The only time this term is mentioned is in the abstract. Perhaps a definition could be included in the manuscript.

   **Response**: The term "quasi" should be removed, as it applies to common form of the GLUE.

2. P3-L58 the authors use 'process-based' models and 'physically-based' models seemingly interchangeably. I recommend choosing either'process-based'or'physics-based' (not 'physically-based', which is admittedly used throughout literature)

**Response**: Difficult choice, we think that process-based suits better and this term will be used in the revised manuscript.

3. P3-L58 I do not share the authors' broad assertion that physics-based approaches are unpopular in practical applications.

   **Response**: It should be specified, that this statement applies to flood hazard assessments, where other methods than the DCM are very rare.

4. P3-L62-64 The authors skip over many other possibilities by jumping from the detailed parameters of the Vastila and Jarvela models to the Manning coefficient. A (physics based) model with fewer parameters would be an option. Numerous studies can be found in literature where two-dimensional models use spatially distributed information on vegetation, often based on remote sensing techniques.

   **Response**: The reviewer is right, we should mention methods that allow to estimate the vegetative roughness based on e.g. treating vegetation as rigid cylinders (with drag coefficient and frontal area/density derived from remote sensing). Although we would like to indicate that these approaches present rather the state-of-art than the present practice in several countries, including Finland and Poland where the authors come from, where the lumped approach to resistance parameters prevails. We will highlight in the introduction that we are focusing on 1D models for practical applications.

5. P3-L72 This is a very valid point, and in my view the most important objective of this study

   **Response**: Thank you for acknowledging this point which we agree is the main objective of the work. If the article is considered for revision, we would like to emphasize this issue throughout the manuscript and present this earlier in the Introduction.

6. P3-L74 "Any method can ... a parameter calibration". I reject this statement. Do the authors like cake only if all its ingredients can be individually tasted?

   **Response**: We had such impression analyzing the problem based on our experience on flood hazard assessments. We understand that we should not generalize, so we consider softing this statement, by i.e.: "Usually a method is widely applied in practice if all its parameters can be identified as the solution to the inverse problem – a parameter calibration."

7. P3-L77 'predictive uncertainty' is a technical term used differently by different authors. The authors should define their use of the term.

   **Response**: We agree and will define "the predictive uncertainty as the estimated total uncertainty of the modeled variable".

8. P3-L79 'As one of the first works, this paper' What paper do the authors refer to? (If they mean their manuscript, see my general remark on novelty).

   **Response**: We meant our manuscript, in this point we still consider our approach novel, with comments given in general responses.

   "As one of the first works, this paper evaluates the uncertainty of chosen 1D state-of-art methods for predicting the influence of complex vegetation on the discharge capacity (understood as the dependency between water level and discharge) in compound channels where vegetative flow resistance dominates".

9. P3-L82 "most of the previous studies": the literature review by the authors cites dated literature. To give confidence in this statement, the authors might provide a review of more recent literature.

   **Response**: The literature review should be updated with e.g. studies on discharge curve fitting. Please see also our general comment on the literature review and position of our study.

10. P3-L89-90 I don't think this is a valid contrast. Morphologic and hydraulic modelling are very different challenges and many existing vegetation models are not suited for morphological modelling.

    **Response**: We disagree at this point. Warmink et al. (2013) analyzed the hydraulic resistance, parametrized with bed morphological features. This makes the study similar to the parametrization of resistance using vegetation characteristics. To avoid future confusions, we would like to develop this comment on Warmink et al. (2013) article.

11. P4-L95 The authors compare 'explicit' and 'implicit' uncertainty analysis. I'm not sure this terminology is commonplace, and in any case requires explanation.

    **Response**: The reviewer is right, we will develop the explanations concerning these terms.

12. P4-L116-L120 I'm unsure why the authors use the terms 'minor values' (text), 'minor parameters' (figure), or 'conservative approach'. To me this sounds derogative, as if to discredit this approach in favour of their proposed alternative, although I readily assume the authors do not intend this. For instance, I do not see why surface roughness is in any way 'minor'. In fact, if this parameter is used for calibration it is very likely the most sensitive parameter, so by all accounts should be labelled 'major'. Nor do I see why the first approach should be labelled 'conservative' (what is conserved? Do the authors mean 'traditional'?). Second (and this point was raised by another referee as well), the 'conservative approach' is surely preferable - if reasonable estimates of the uncalibrated parameters are available - over treating vegetation models as black boxes.

    **Response**: We thank the reviewer for raising these points and want to highlight that it was not our intention to discredit approaches where measured values of vegetation are available, but to analyze if the the use of physic-based approaches provide reliable estimates for small channels without a prior knowledge on vegetation properties. We will clarify this scope of the manuscript more clearly when describing the goal of the paper at the end of Introduction section. Following the reviewer's suggestion, we will use the term "traditional approach" when referring to the way how it is usually done (Fig. 1a), as the opposition to the new (proposed) approach (Fig. 1b). Further, we will highlight that "traditional" approach is preferable when vegetation data is available. In the case black-box issue, please refer to our broad answer on that issue.

In the case of "minor values", we meant those parameters of process-based methods that have less importance on the conveyance estimation of compound channels compared to vegetation characteristics; herein, please note that (P4, L100) states "This work focuses on one-dimensional methods for compound channels with a significant share of the flow resistance generated by vegetation.". We will express this issue more clearly in the revised manuscript. We agree that the term "minor" is not precise, as these parameters are nevertheless significant. So we will replace the word "minor" by "parameters other than vegetation properties". We agree that the term "minor" may have sounded derogative as we did not express ourselves clearly, although this was not our intention.

13. P5 L134-135: For its merits in popularizing uncertainty analysis, the GLUE method is (in)famous for the liberal use of the likelihood measure, which does not agree with Bayes' theorem. The authors choose to use a so-called 'informal' likelihood measure with a scaling factor that controls the uncertainty. The authors then force the model uncertainty to include at least the right amount of data points through equation 5. This approach, inspired by Bayes theorem, has the known disadvantage that predictive and model uncertainty are lumped (the authors approximate the total uncertainty), that parameter uncertainty tends to be overestimated, and that the choice for a likelihood measure is arbitrary (i.e. not following from the error model, as is the case in a proper Bayesian approach). A defense by the authors on their choice for an informal over a formal approach

would be appreciated for readers unfamiliar with this distinction. Also, given that the authors use an informal method, I'm interested whether a behavourial threshold was used(the authors mention the need for this, but not whether it was used. I assume none was used given the scaling factore.)

**Response**: The response on the uncertainty analysis was given before in general comments. In the case of behavioral threshold, the reviewer is right, we have not used it, because of the applied likelihood function. Such comment will be added to the manuscript after defining the likelihood function.

14. P7-L194 Please elaborate which resistance (all,only the bed,only the imaginary wall?)

   **Response**: The information in the manuscript is not precise: bed, imaginary wall and also vegetation stems.

15. P11-L297 Is the Ritobacken Brook free [FB02?]owing? Can uniform [FB02?]ow be reasonably assumed? P13 In general, I suggest adding the first section of chapter 3 to the method section, as new methodology is introduced here.

   **Response**: The Ritobacken Brook is free flowing in that there are no hydraulic structures affecting the flow at the investigated discharges and water levels. No changes were observed in the rating curve at the downstream culvert (downstream of which there is a forested section with steep slope). At very high discharges, the culvert at the downstream end of the study reach will start to dam the flow, but no such high flows were recorded in the present data. The flow at Ritobacken is gradually varied (we will replace the term "non-uniform flow" by "gradually varied"), and therefore we used the energy slope instead of bed slope(L312). In the case of the first section of the chapter 3, we agree on moving lines 315-324 to methodology, of course keeping the tables in results.

16. P13-L315 By'trial and error' choosing the sampling size of Monte Carlo, do the

authors mean increasing the sample until convergence is observed? What convergence criteria is used? Which sampling method is used?

**Response**: We thank the reviewer for pointing out that this information is missing in the article. For the convergence we used two criteria: mean of computed water levels and good fit (in deterministic manner) of the rating curves determined for each combination of observations. For the sampling we used Latin Hypercube method. The information will be included in the text.

17. P13-L315 "In a similar way"; are a priori distributions chosen by trial and error? In principle a priori distributions are either informative, based on prior knowledge, or uninformative. Here an uninformative uniform prior is chosen, but I have to learn this from the captions in table 1. It would be helpful if this is explicitly added to the text as well. I would also appreciate a brief exposition of the choice for a noninformative prior. Given the models are physics-based, and the authors have a pretty good estimation of their likely values, it seems more logical to use informed priors.

**Response**: Please, see general comments: Ad. 3.

18. P17-L365 Here the authors define "model identifiability" as (I paraphrase): "it is identifiable if it is fittable". The authors then admit that their approach would allow even poor models to [fit well, while "the only limitation could be the physical meaning of the parameters". It is unclear whether the authors did indeed let themselves be restrained by the physical meaning of those parameters. One may remark here that minor changes to their chosen approach (i.e. a formal Bayesian approach and informed priors) would be expected to alleviate some of these problems.

**Response**: This issue was addressed in general comments.

19. At this point it is also good to remark on a different, perhaps more fundamental point. The authors go into depth into 'model identifiability' but the reader was led

to believe that 'parameter identifiability' was the objective of the study. Yet most results and almost all figures focus on the question 'will the model fit' - which is not a very interesting point to stress given the objective of the study. The only figures that support 'parameter identifiability' are 14 and 15, but those are currently insufficient to support the claims made in the conclusions; it would be helpful to plot the a priori cdfs as well, so as to see how they were constrained a posteriori.

**Response**: This issue of model/parameter identiafiablity was addressed in general comments. The prior distributions are uniform, so the way how they are constrained with likelihood function can be presented using dot-plots (parameter value vs likelihood measure) or like in the case of Fig. 14-15. We prefer the second option, as the figure is much more readable. As the reviewer found Fig 14-15 interesting, we think that such plots can be presented for other methods as well.

20. P31-L495 - The 'trial and error' a priori distribution estimations bothers me a bit when claiming objectivity. Given the limited number of observations, the a priori distribution is expected to affect the output.

**Response**: Explanation given in a broad comment on the uncertainty estimation.

21. *P31-L500 'It was possible to identify...one (DCM)'. This should not be surprising. It is in general easier to fit a model with more parameters than one with fewer. However, the more impressive claim would be that the parameters are identifiable as well. I refer to the work of Werner et al. (2005, doi:10.1016/j.jhydrol.2005.03.012), to illustrate that challenge.*

**Response**: This links with the problem statement: model identification vs parameter identification. As we mentioned in general comments, we would like to present the model identification and then analyze it outcomes, also in terms of parameter identification. The reviewer's idea, given with the Werner et al. (2005) is very interesting, as it could be analyzed in terms of uncertainty estimation with

increasing knowledge on parameter variability. However, we are afraid, that it would be hard to address this issue in a single article.

22. P32-L528 'Thus, our result... resistance dominated'. I do not see how this follows from your results nor from the previous sentence.

**Response**: We agree that this information was not directly presented. The Ritobacken case was also monitored with the absence of vegetation as cited in L28: "particularly in small to medium-sized channels where up to 90 per cent of the [FB02?]ow resistance can be caused by plants (e.g. Västilä et al., 2016)." We will add this result from the field to the appropriate place in the revised manuscript. The reviewer is right, we have not performed studies without vegetation so the claim is unsupported with the results. The results show, that the choice was important for analyzed cases and in this way, the sentence should be rephrased.

23. P33 Conclusions. The authors conclude the article with 8 claims.

Claim 1. The authors claim it is possible to identify the parameters of physics-based models, even if those models have many parameters. This is an unlikely claim (given the number of data points and the number of model parameters), but may follow from a confused definition of 'parameter identifiability' versus 'model identifiability'. If the parameters are 'identifiable', I would expect narrow a-posteriori distributions compared to the a-priori distributions. Figures 14-15 do not show a-priori distributions. Although it is difficult to judge whether the parameter distributions are meaningfully narrow, it seems only C is well defined. If so, it would be interesting to re[FB02?]ect on figure 1. The second claim, like the first, seems to only apply to model identifiability, not parameter identifiability. The authors might spend some words on how they perceive the model to be used-does it matter if the models are physics-based? Or any(data-based) model that fits the rating curve applicable?

Claim 3: Perhaps the authors could explain how uncertainty relating to parameter

equifinality can be distinguished from other uncertainty.

Claim 4, first sentence: Would not a better explanation be that the model is insufficient in some way, and that model parameters differ to account for this? Second sentence: I don't understand this in relation to the third claim.

**Response**: we have addressed this remarks in general comments (Ad. 5).

**Literature**

- Berends, K. D. (2019) Human intervention in rivers, quantifying the uncertainty of hydraulic model predictions, PhD thesis, University of Twente, Netherlands.

- Blasone, R., Vrugt, J., Madsen, H., Rosbjerg, D., Robinson, B., and Zyvoloski, G. (2008). "Generalized likelihood uncertainty estimation (GLUE) using adaptive Markov chain Monte Carlo sampling." Adv. Water Resour., 31(4), 630–648.

- Knight, D. W., Omran, M., and Tang, X.: Modeling depth-averaged velocity and boundary shear in trapezoidal channels with secondary flows, Journal of Hydraulic Engineering, 133, 39–47, 2007.

- Romanowicz RJ, Beven KJ, Tawn J. Bayesian calibration of flood inundation models. In: Anderson MG, Walling DE, editors. Floodplain processes. Chichester: Wiley; 1996. p. 333–60.

- Romanowicz, R. J., & Beven, K. J. (2006). Comments on generalised likelihood uncertainty estimation. Reliability Engineering & System Safety, 91(10-11), 1315-1321.

---

## Author Response (AR1)

May 12, 2020

Hydrology and Earth System Sciences (HESS)

Dear Editor,

In the revision of the manuscript entitled: "Predicting discharge capacity of vegetated compound channels: uncertainty and identifiability of 1D process-based models" we included all reviewers remarks, according to the previously posted responses. Please find the manuscript with marked changes attached.

Following reviewers remarks, we have rewritten the introduction section to make it more clear. We also improved the literature review to provide a better positioning of our study in the state-of-art. We hope, that the revised methodology section provides a better explanation of the concept of our study. We also developed the analysis of results and discussion of outcomes. The conclusion points should be now better grounded.

In addition to changes, we indicated in our responses to reviewers, we introduced following, important modifications:

1. We found a mistake in the code for the PTLM model. Instead of using a hydraulic radius in the term $\frac{h}{R}$ of Equation 12, we used as in the original formulation, the water depth $\frac{h}{H}$. In the result it was necessary to recompute all PTLM cases.

2. As we indicated in our response, we use an uninformative parameter ranges for *a priori* parameter distributions, but within physical bands. It appeared, that we used wrong ranges for several parameters:

   - $C^*$, used in the GTLM, STLM and PTLM, instead od $0.08 - 1$ it is now $0.01 - 0.2$;

   - $A_l/A_b$ and $A_l/A_b$, used in the GTLM, instead of $0 - 3.2$ and $0 - 0.16$ it is now $0 - 30$ for both;

   - $C_D a$ for PTLM, instead of $0.01 - 0.4$ it is now $0.01 - 100$

3. New calculations were performed with larger number of Monte Carlo simulations, as we also performed more detailed investigations of the solution convergence.

Because of new simulations and also reviewers remarks, all figures were recomputed. Please, note that due to technical limitations of Latexdiff tool, changes in figures were not presented in the document with registered changes.

Yours sincerely

Authors

[revised manuscript text omitted]

---

## Author Response (AR2)

Dear Editor,

We are very grateful for your and reviewers' contribution in improving our article. It is clear to us that the comments of the reviewers have allowed us to significantly improve the manuscript. Thank you for your effort. Bellow, please find our responses to reviewers' comments.

In acknowledges, we have included additional funding source of the Polish National Centre for Research and Development, which provided us with funds for some numerical analyzes, we used in the present study. We forgot to mention this contribution earlier, and we would like to add a note the end of acknowledgements (the submitted version includes this note). Additionally, we also included projects, in which data for Ritobecken (published in the manuscript) was acquired. The new version of acknowledges :

"The research was partly supported by National Science Centre (Poland), Program Miniatura 1, project no. 2017/01/X/ST10/00987, Maa- ja vesitekniikan tuki ry (Grant No 33271), Maj and Tor Nessling Foundation (Grant No 201800045), and by the National Centre for Research and Development (Grant No 347837/11/NCBR/2017, ``Technical innovations and system of monitoring, forecasting and planning of irrigation and drainage for precise water management on the scale of drainage/irrigation system"). We acknowledge Academy of Finland (Grant No 133113), Maa- ja vesitekniikan tuki ry and the Finnish Ministry of Agriculture and Forestry for funding the collection of the original field data."

**Reviewer 1:**

The reviewer probably did not find our point-by-point responses posted in the authors' response:

https://www.hydrol-earth-syst-sci-discuss.net/hess-2019-635/#discussion

We are sorry for the situation, because reviewers comments were very valuable.

**Reviewer 2:**

*My only real remaining concern that exceeds technicalities, is that I would have liked to have those questions more directly answered in the conclusions. For example, the last sentence of the abstract and question number four are not answered in the conclusions. One might imagine a reader, made enthusiastic by these 4 questions, readily skipping to the conclusion for the answer, only to find 8 conclusions to different questions. However, I would leave it to the authors to consider this advice.*

*Response*

We agree with the reviewer. We have added the following sentence to the point 5 of the conclusions: "Therefore, the results showed that it is possible to choose an appropriate model, without a prior knowledge of vegetation properties in the channel, by comparing obtained uncertainty widths." The last sentence of the point 6 was developed into a separate point (now 7 of 9):

"7. In most cases, the Manning-based DCM had also satisfactory performance, but results suggests it

had poorer capabilities for extrapolation to high floodplain flows when calibrated with only low floodplain flows, in comparison to process-based models."

*Specific & technical comments:*

*Line numbers refer to 'hess-2019-635-manuscript-version3.pdf'*

*The authors introduce numerical convergence in line 611. I would like the authors to make explicit what they mean by this convergence. The authors state multiple times that the Pasche method has a complex numerical nature - which may lead to instabilities, yet did not before address a lack of convergence. Perhaps a small note in the methodology may address this.*

*Response*

We rephrased the use of the term convergence in the sentence in line 610: "The differences in the case of these particular parameters comes from the more complex structure of the Pasche model, restricting values of az, due to lack of a numerical convergence." We meant here, the Pasche model convergence. We have rephrased the sentence as follows: "The differences in the case of these particular parameters comes from the more complex  structure of the Pasche model, restricting values of a_z, due to lack of a numerical convergence for its implicit formulas."

We also added the note about the convergence in the methodology section, line 289: "Equations describing these dependencies have an implicit form that requires iterative methods for solving, so that the Pasche method has a very complex numerical solution and it may be affected by a lack of convergence for infeasible parameter sets."

*L 78: 'The obtained in the field values' check sentence*
It was: "The obtained in the field values characterizing vegetation, have to be attributed to a spatial unit, representing usually a vegetation class." Changed to:  "Values characterizing vegetation, obtained in the field, have to be attributed to a spatial unit,  usually representing a vegetation class."

*L 79: "representing usually a vegetation class", switch 'usually' and 'representing'*
OK
*L80-81: make explicit what you mean by 'it' ('generalisation of...?')*
"On the one hand, together with the nonlinear form of the vegetation resistance models it introduces significant uncertainty." Changed to: "On the one hand, together with the nonlinear form of the vegetation resistance models such a generalization introduces significant uncertainty. "

*L82: is 'should' the correct word here? Perhaps 'instead of representing physical quantities, they now reflect...'. I think the ideal is that our equations are somewhat related to physical quantities, but indeed, the waters get mudied.*
Was: "On the other, it weakens  the link between measured values and model parameters, which instead of representing physical quantities, should reflect their lumped hydraulic effect." Now: "On the other, it weakens  the link between measured values and model parameters, which reflect the lumped hydraulic effect instead of representing physical quantities."

*L724: 'the best' - I'm not sure what the authors mean by 'the best uncertainty estimates'. Narrower?*
"The results agree with Dalledonne et al., (2019), who obtained the best uncertainty estimates for the more complex models. " Changed to: "The results agree with Dalledonne et al. (2019), who obtained the narrowest uncertainty estimates for the more complex models."

*L795: "unreal" Perhaps 'unrealistic?'*

Changed to unrealistic.

**Reviewer 2:**

*The authors resubmitted a carefully revised manuscript in which my comments have been adequately addressed. However, the language should still be further polished and I provide some examples below. Overall, the manuscript can, in my opinion, be accepted.*

*L61: This clearly depends on the point of view and the complexity of the problem. One could also say that a drawback of most existing methods to determine flow resistance of vegetation is that these methods oversimplify the problem so that they can be easily applied.*
"An important drawback of vegetation models for hydraulic resistance, from the practical point of view, is that they require much more data than traditional methods."
Reviewer is of course right, although taking a modeler perspective, the number of parameters is a key factor, why process-based methods are not considered. We will specify this view by adding word modeler in brackets after word practical:
"An important drawback of vegetation models for hydraulic resistance, from the practical (modeler's)  point of view, is that they require much more data than traditional methods."

*L78/79: Please check this sentence. It does not make sense.*

The sentence:
"The obtained in the field values characterizing vegetation, have to be attributed to a spatial unit, representing usually a vegetation class."
Was changed to: "Values characterizing vegetation, obtained in the field, have to be attributed to a spatial unit,  **usually representing** a vegetation class."

*L83: What exactly is meant with the structure of the flow model? 1D, 2D, or 3D models?*
We used the term in more general meaning, including governing equations, assumptions of flow dynamics, etc.
At the end of the sentence:
"Such quantities are of course immeasurable and depend on the structure of the flow model."
We will add:
", adopted governing equations, simplifications of the flow dynamics".

*L85: Please check this sentence.*

"So, to treat them similarly to  Manning coefficients, which are usually determined in this way, by adjusting their values, to obtain an agreement between  computed and observed e.g. water levels, stream velocities or flow rates -- by solving the inverse problem through calibration ...:"
Rephrased to:
"So, to treat them similarly to  Manning coefficients, which are usually obtained by the model **calibration**, **where their values are adjusted**, to ensure an agreement between  computed and observed e.g. water levels, stream velocities or flow rates -- by solving the inverse problem ...:"

*L128/138: Check for redundancy.*
We have rephrased the first paragraph of the methodology, was:
"This section provides an overall description of the applied methodology. The analysis is performed with process-based approaches for vegetation roughness, including Pasche (Pasche, 1984) and Mertens (1989) models for rigid emergent vegetation(section 2.2.2) and flexible vegetation models based on the two-layer assumption of Luhar and Nepf (2013), generalized by140Västilä and Järvelä (2018, sections 2.2.3-2.2.4). Computations were performed for steady state conditions, by applying vegetation roughness model for water levels in a channel cross section"

Now:
"This section provides an overall description of the applied methodology. In the subsection 2.2.2 Pasche (Pasche, 1984) and Mertens  (1989)  models  for  rigid  emergent  vegetation  were presented.  Flexible  vegetation  models  based  on  the  two-layer assumption of Luhar and Nepf (2013), generalized by Västilä and Järvelä (2018) were provided in subsections 2.2.3-2.2.4. Computations were performed for steady state conditions, by applying vegetation roughness models **to find** water levels in a channel cross section."

*L159: I am not sure that I understand. E.g., what is meant by model calibration for mixed flows?*
"For calibration the points of  rating curves were used, the effect of different possible combinations of observations in identification task was also investigated, e.g. model was calibrated for a set of five lower flows, but also for a set of five higher and mixed ones."
We rephrased the term mixed flows to: "all intermediate sets".

*L166: I am a bit confused here – at this line it is stated that discharge Q is given and at L140 it was stated that vegetation roughness model was applied for water levels…. (and hence not for a given discharge; see also L178).*
We were solving for water levels and to make it clear, we changed to sentence in Line 141, which sounds now:
"Computations were performed for steady state conditions, by applying vegetation roughness models **to find** water levels in a channel cross section."

*L176: Check format.*
We put the citations in the brackets.

*L250: Please specify this comment.*

Was: (see comment of Mantovan and Todini, 2006), we changed to: see the comment on the purpose of the Bayesian identification of Mantovan and Todini, 2006).

*L251: Complicated sentence.*
Agree, was: "Parameter variability is used to describe the uncertainty, specifically with the Equation 2 the error $\zeta$."
Rephrased to: "Parameter variability is used to describe the uncertainty, specifically the error $\zeta$ defined with the Equation 2."

*L314: Something is not correct in this sentence.*
"Physically, the drag coefficients for bed and the vegetation zone interface may take separate values"
We rephrased it to:
"Physically, there might be different values of drag coefficients for bed and the interface of the vegetation zone."

*L417: "real measured values! – what does "real" refer to?*
Thank you, we removed word "real" and change it to "measured".

**L492 – Figure 10 to Figure 14: Is it possible to adjust the color code so that the each Model is always characterized by the same color?**
Good point, we updated the figure colors.

**L704: Check sentence - the phrase "extrapolation was not successful in Autumn 2011" does not make sense to me.**
We rephrased the sentence:

[revised manuscript text omitted]

---

## Author Response (AR3)

July 20, 2020

Hydrology and Earth System Sciences (HESS)

Dear Editor,

Thank you again for your effort, we have corrected both sentences, as you suggested. We apologize for these differences in the letter and manuscript. The article was proofread, and we forgot to update the quotes in the letter.

Yours sincerely

Authors